# ReForm-Eval: Evaluating Large Vision Language Models via Unified Re-Formulation of Task-Oriented Benchmarks

## Abstract

Recent years have witnessed remarkable progress in the development of large vision-language models (LVLMs). Benefiting from the strong language backbones and efficient cross-modal alignment strategies, LVLMs exhibit surprising capabilities to perceive visual signals and perform visually grounded reasoning. However, the capabilities of LVLMs have not been comprehensively and quantitatively evaluated. Most existing multi-modal benchmarks require task-oriented input-output formats, posing great challenges to automatically assess the free-form text output of LVLMs. To effectively leverage the annotations available in existing benchmarks and reduce the manual effort required for constructing new benchmarks, we propose to re-formulate existing benchmarks into unified LVLM-compatible formats. Through systematic data collection and reformulation, we present the ReForm-Eval benchmark, offering substantial data for evaluating various capabilities of LVLMs. Based on ReForm-Eval, we conduct extensive experiments, thoroughly analyze the strengths and weaknesses of existing LVLMs, and identify the underlying factors. Our benchmark and evaluation framework will be open-sourced as a cornerstone for advancing the development of LVLMs.

## 1 Introduction

With the trend led by ChatGPT (OpenAI, 2023a), LLMs (Large Language Models) (OpenAI, 2023b; Touvron et al., 2023a; Chiang et al., 2023) have ushered in revolutionary advancements in Natural Language Processing (NLP). Inspired by these efforts, researchers attempt to extend the success of LLMs to the realm of vision language. By equipping LLM with visual encoders and aligning multi-modal representations through generative pre-training, large vision language models (LVLMs) (Li et al., 2023b; Liu et al., 2023b; Zhu et al., 2023; Ye et al., 2023) possess the capability to comprehend visual information and engage in multi-modal conversations with users.

However, the reliability of such LVLMs remains a mystery. On the one hand, these models demonstrate surprising abilities like OCR (Liu et al., 2023d), meme understanding (Zhu et al., 2023), and visual commonsense reasoning (Li et al., 2023b). On the other hand, LVLMs suffer from fundamental issues, such as object hallucination (Li et al., 2023d). Meanwhile, due to the lack of suitable benchmarks, there is a shortage of quantitative analysis and comparison of LVLMs.

The main reason for this situation is the structural gap between existing task-oriented multi-modal benchmarks and LVLMs. Most existing benchmarks are designed for specific tasks and demand highly structured input-output formats (Lin et al., 2014). For instance, VQA v2 (Goyal et al., 2017) requires concise answers, typically in the form of single words or short phrases. Previously evaluated vision-language pre-trained models (Chen et al., 2020; Zhang et al., 2021) need to be fine-tuned and learn task-specific parameters to fit the structures of such benchmarks. On the contrary, LVLMs are flexible and tend to provide detailed responses, even for yes-or-no questions. As depicted in the flowchart in the upper part of Figure 1, such gap poses the greatest obstacle to accurate automated evaluation, particularly when assessing the desired zero-shot capabilities.

To bridge the structure gap, we explore ways of re-formulating existing benchmarks into unified formats that are compatible with LVLMs. Referring to Figure 1, we adapt the evaluation process to the unified form shown in the lower part. Multi-modal benchmark datasets are re-formulated as

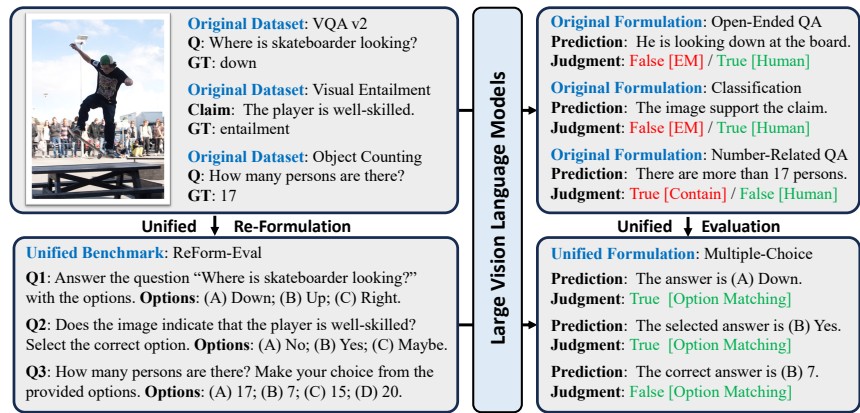

Figure 1: Illustration of the unified re-formulation of existing benchmarks into multiple-choice problems. The text within square brackets indicates the evaluation methods, with red and green denoting incorrect and correct judgment, respectively. "EM" is short for exact match.

multiple-choice problems or specialized text generation problems. Datasets for tasks with specific text generation requirements, like OCR and image captioning, are re-formulated as specialized text generation problems. Other datasets are restructured into multiple-choice problems.

The unified formulation enables universal and comprehensive evaluation. For each formulation, we design a consistent and reliable evaluation method. As mentioned in (Fu et al., 2023), current LVLMs may struggle to follow multiple-choice instructions, we propose both black-box and white-box approaches to assist: (1) Guiding LVLMs to output in desired formats through in-context-learning; (2) Directly calculating the generation probability for options and selecting the one with the highest value. Considering the sensitivity of LVLMs to the input prompts (Zeng et al., 2023), we design an instability-aware evaluation strategy and introduce a metric to characterize such instability.

Based on the re-formulation framework, we present our unified multi-modal benchmark, ReForm-Eval. For a comprehensive evaluation, we re-formulate 61 benchmark datasets based on existing data resources, the evaluation dimensions range from basic visual perception to high-level visual reasoning and dialog. Compared with recent LVLM benchmarks that require manual annotation (Fu et al., 2023; Liu et al., 2023c), ReForm-Eval fully utilizes publicly open resources and provides significantly more data, almost 100 times the size of MMBench. Meanwhile, unlike LVLM-ehub (Xu et al., 2023), which requires designing complex and dataset-specific evaluation strategies, ReForm-Eval offers greater scalability and a more universally applicable and efficient evaluation approach.

Based on ReForm-Eval, we conduct a comprehensive evaluation of 16 open-source LVLMs across various capability dimensions. We hope ReForm-Eval and the associated findings can constitute a valuable augmentation to the ongoing efforts in LVLM research and development.

## 2 RELATED WORKS

### 2.1 LARGE VISION LANGUAGE MODELS

Inspired by the advancements of LLMs and the multi-modal understanding abilities demonstrated by GPT-4 (OpenAI, 2023b), developing open-source LVLMs currently dominates the multi-modal research. Visual signals encoded by visual encoders (Radford et al., 2021) are incorporated in LLMs through linear projection (Tsimpoukelli et al., 2021), Q-former (Li et al., 2023b), or cross-attention layers (Alayrac et al., 2022). To enable multi-modal instruct tuning, MiniGPT4 (Zhu et al., 2023) bootstraps high-quality data by refining the previous output, LLaVA (Liu et al., 2023b) proposes to employ GPT-4 to generate image-involved dialogs while other works construct instruct tuning data from existing vision-language benchmarks (Xu et al., 2022; Dai et al., 2023; Li et al., 2023c).

To seamlessly adapt LLMs for multi-modal scenarios, many efforts are paid including designing strategies for parameter freezing (Ye et al., 2023), introducing light-weight trainable modules into the backbone (Gong et al., 2023; Gao et al., 2023), incorporating continuous output (Peng et al., 2023; Chen et al., 2023), and enhancing the visual representations (Zeng et al., 2023; Hu et al., 2023; Li et al., 2023a). Benefiting from the aligned representations from ImageBind (Girdhar et al., 2023), LVLMs can be further extended to more modalities (Han et al., 2023; Su et al., 2023).

However, the capabilities of existing LVLMs are mainly demonstrated by qualitative examples (Zhu et al., 2023; Su et al., 2023; Gong et al., 2023). To our knowledge, few benchmarks are suitable for evaluating the capabilities of LVLMs, hindering quantitative analysis and comparison of LVLMs.

## 2.2 MULTI-MODAL BENCHMARKS

**Task-Oriented Benchmarks** Most existing multi-modal benchmarks can not be directly utilized to evaluate LVLMs since they are designed for specific tasks and rely on structured input-output formats for evaluation. VQA v2 (Goyal et al., 2017) requires concise answers, retrieval benchmarks (Lin et al., 2014; Young et al., 2014) demand dense scores for all image-text pairs, VCR (Zellers et al., 2019) provides coordinates to refer visual object in the question, and bounding box output is necessary for RefCOCO (Kazemzadeh et al., 2014). This characteristic makes it challenging to utilize such benchmarks to evaluate the free-form text outputs of LVLMs unless complex post-processing and evaluation methods are designed specifically (Xu et al., 2023; Yin et al., 2023).

**Benchmarks for LVLMs** To facilitate reliable and efficient automated evaluation of LVLMs, efforts are paid to construct LVLM-compatible benchmarks, such as yes-or-no problems in MME (Fu et al., 2023) and multiple-choice problems in MMBench (Liu et al., 2023c). A portion of the benchmarks are designed to assess specific capabilities (Liu et al., 2023d; Wang et al., 2023) or diagnose particular issues (Li et al., 2023d; Zhao et al., 2023), while others aim for comprehensive evaluation (Fu et al., 2023; Liu et al., 2023c). However, limited manual annotation (around 100 samples per dimension in MME and MMBench) could potentially introduce evaluation bias into the results.

## 3 REFORM-EVAL BENCHMARK

In this section, we describe how to construct ReForm-Eval by re-formulating existing task-oriented multi-modal benchmarks. Section 3.1 introduces the general framework of re-formulation. Section 3.2 summarizes the capability dimensions assessed in ReForm-Eval and corresponding datasets. Section 3.3 illustrates the methods and strategies used to evaluate LVLMs based on ReForm-Eval.

### 3.1 UNIFIED RE-FORMULATION FRAMEWORK

Existing LVLMs primarily adopt LLMs as backbones and use free-form text to interact with users. This paradigm makes the output more flexible and aligned with human needs. However, the gap between these models and existing highly structured benchmarks poses challenges for evaluation. In order to effectively reuse the annotations in existing benchmarks, these benchmarks need to be re-formulated into appropriate formats. Motivated by benchmarks for LLMs (Hendrycks et al., 2020; Srivastava et al., 2022; Huang et al., 2023), ReForm-Eval considers two formats that are compatible with LVLMs, namely multiple-choice problems and text-generation problems.

Multiple-choice problem is the primary format in ReForm-Eval. By providing options for the questions, models are guided to produce responses in a constrained format. The key in multiple-choice problem construction is how to prepare meaningful negative options. Generally, for close-vocabulary classification tasks, we build relationships between categories based on which hard negative options are selected. For open-ended tasks, based on the question and the correct answer, negative options can be obtained with the help of task-specific strategies or LLMs like ChatGPT.

For OCR and image captioning that involves text generation, corresponding benchmarks are formulated as text-generation problems tailored to various scenarios. We curate the input prompts to describe the tasks and requirements. For OCR tasks, responses should contain the target tokens in the image. For description tasks, models should provide concise depictions of the visual content.

### 3.2 EVALUATION DIMENSIONS

To address the wide range of questions posed by users, LVLMs need to possess diverse capabilities. For a comprehensive evaluation, we curate 61 benchmark datasets from existing resources, summarizing the assessed capabilities into 2 major categories and 8 sub-categories which are illustrated in Figure 2. To avoid information overload, details about the re-formulation procedures and dataset statistics are provided in Appendix A.

### 3.2.1 VISUAL PERCEPTION TASKS

**Coarse-Grained Perception (CG)**  Coarse-grained perception is the ability to recognize the overall layout and main objects at the image level. We evaluate this capability through ***image classification*** using Flowers102 (Nilsback & Zisserman, 2008), CIFAR10 (Krizhevsky et al., 2009), ImageNet-1K (Deng et al., 2009), Pets37 (Parkhi et al., 2012), and MEDIC (Alam et al., 2023) benchmarks, and ***scene recognition*** using TDIUC (Kafle & Kanan, 2017) and VizWiz (Gurari et al., 2018) benchmarks. The samples are re-formulated as multiple-choice questions.

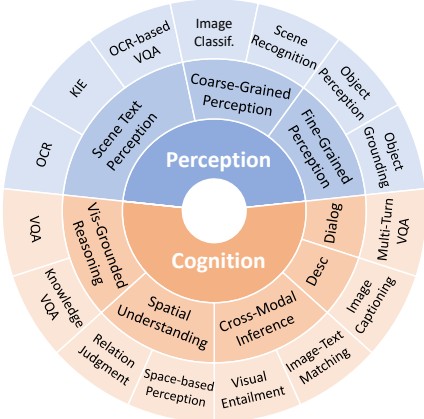

Figure 2: Assessed capability dimensions and tasks in ReForm-Eval. "Desc" and "Classif" are respectively short for description and classification.

**Fine-Grained Perception (FG)**  Fine-grained perception requires detailed sensing at the object level. We set up the ***object perception*** task (using TDIUC (Kafle & Kanan, 2017) and MSCOCO (Lin et al., 2014) benchmarks) and the ***object grounding*** task (using MSCOCO (Lin et al., 2014) and Ref-COCO (Yu et al., 2016) benchmarks) for evaluation. Object perception measures how well a LVLM can identify local semantics, while object grounding assesses the ability to localize fine-grained objects. All tasks are formulated as multiple-choice questions.

**Scene Text Perception (STP)**  Scene text perception enables LVLMs to identify, understand, and perform inference based on text in images. This evaluation is conducted through ***optical character recognition*** (OCR) using 6 benchmarks (including CUTE80 (Risnumawan et al., 2014), IC15 (Karatzas et al., 2015), IIIT5K (Mishra et al., 2012), COCO-Text (Mishra et al., 2012), WordArt (Xie et al., 2022), TextOCR (Singh et al., 2021)), ***key information extraction*** (KIE) using 3 benchmarks (including SROIE (Huang et al., 2019), POIE (Kuang et al., 2023) and FUNSD (Jaume et al., 2019)) and ***OCR-based VQA*** using 3 benchmarks (including TextVQA (Singh et al., 2019), DocVQA (Mathew et al., 2021) and OCR-VQA (Mishra et al., 2019)). We consider STP as a specialized text-generation problem that requires output to contain exactly matched words.

### 3.2.2 VISUAL COGNITION TASKS

**Visually Grounded Reasoning (VGR)**  A reliable LVLM is supposed to perform reasoning based on multi-modal contextual information. In order to assess such capability, we adopt the commonly applied ***visual question answering*** (VQA) task and its variant, ***knowledge-based visual question answer*** (K-VQA), which further requires models to utilize internally stored knowledge. For vanilla VQA, we adopt VQA v2 (Goyal et al., 2017), GQA (Hudson & Manning, 2019), and Whoops (Bitton-Guetta et al., 2023). As for KVQA, we consider 6 benchamrks including OK-VQA (Marino et al., 2019), ScienceQA (Lu et al., 2022), VizWiz (Gurari et al., 2018), Vi-QuAE (Lerner et al., 2022), A-OKVQA (Schwenk et al., 2022) and ImageNetVC (Xia et al., 2023). The aforementioned benchmarks are re-formulated into multiple-choice questions.

**Spatial Understanding (Spatial)**  Spatial understanding is the key to the real-life application of LVLMs on robots. This task requires a comprehensive understanding of both the object-object and object-observer relationship so as to make reasonable behaviors. We access such capability through ***spatial relation judgment*** (SRJ) using VSR (Liu et al., 2023a) and MP3D-Spatial, a benchmark designed for embodied tasks in real-world environments, constructed from Matterport3D (Chang et al., 2017). Additionally, we employ ***Space-Based Reasoning*** (SBR) through the CLEVR (Johnson et al., 2017) benchmark. The SRJ task aims to accurately identify spatial relationships, forming a concept of where the ego is in space. The SBP task entails complex reasoning ability based on the understanding of spatial relationships. All samples are re-formulated as multiple-choice questions.

**Cross-Modal Inference (CMI)**  A thorough comprehension of both modalities is required to perform cross-modal inference on the relationship between images and texts. We consider two tasks: ***image-text matching*** (ITM) requires models to measure the cross-modal similarities and ***visual***

*entailment* (VE) demands models to check whether the information is entailed across modalities. MSCOCO (Lin et al., 2014), WikiHow (Koupaee & Wang, 2018), Winoground (Thrush et al., 2022) are adopted for ITM while VE considers SNLI-VE (Xie et al., 2019) and MOCHEG (Yao et al., 2023). Both tasks are re-formulated as multiple-choice questions.

**Visual Description (Desc)**    Visual description is an inherent capability of LVLMs as generative models. We adopt the ***image captioning*** task on MSCOCO (Lin et al., 2014), TextCaps (Sidorov et al., 2020), NoCaps (Agrawal et al., 2019), and Flickr30K (Young et al., 2014) for evaluation. These datasets are formulated as text-generation problems with the requirement of concise outputs.

**Multi-Turn Dialogue (Dialog)**    Existing benchmarks primarily focus on single-turn conversation. ReForm-Eval evaluates the performance of LVLMs in multi-turn dialogues. We consider the ***multi-turn VQA*** task using VisDial (Das et al., 2017) and VQA-MT, the latter is constructed by reorganizing questions in VQA v2. Both benchamrks are formulated as multiple-choice questions.

## 3.3   EVALUATION STRATEGY

### 3.3.1   EVALUATION METHODS AND METRICS

With the unified problem formulation, the performance of LVLMs can be universally evaluated. For specialized text-generation problems, the evaluation method depends on the scenario. For visual description, we follow Li et al. (2023b) to use CIDEr (Vedantam et al., 2015) as the evaluation metric. Since the adopted datasets mainly provide concise references, we craft the prompt to require concise responses and restrict the maximum number of tokens a model can generate. As for STP, input prompts are well-designed to instruct models to identify the scene texts. The evaluation metric is word-level accuracy: the proportion of ground-truth words that appear complete in the output.

Considering multiple-choice problems, the model performance is assessed using accuracy. We label the answer options with markers like "(A)" and then determine correctness by checking the markers in the output of models. The challenge with this approach is that current LVLMs may not always adhere well to multiple-choice instructions, i.e. the output may not include the required marker.

To assist in the evaluation of multiple-choice problems, ReForm-Eval provides both a black-box method and a white-box method. The black-box method provides in-context samples to guide LVLMs to generate responses in desired formats. Here is an example of the input prompt:

> $X_{\text{system-message}}$
> Human: Can you see the image? Options: (A) Yes; (B) No; (C) Not Sure; (D) Maybe.
> Assistant: The answer is (A) Yes.
> Human: $X_{\text{question}}$ Options: $X_{\text{options}}$
> Assistant: The answer is

where $X_{\text{SystemMessage}}$ is the system message required by most LVLMs, $X_{\text{question}}$ and $X_{\text{options}}$ are respectively the question and the answer options described in text, the text in red is the in-context sample provided to the model. Notice that the in-context sample provides no information about the image. The effectiveness of the black-box strategy is demonstrated in Section 4.3.3.

The white-box approach is based on the inherent attribute of current LVLMs as generative models. Given the visual context $v$, the question $q$, and $N$ answer options $C = \{c^i\}_{i=1}^N$, the answer prediction can be determined by the generation likelihood predicted by the evaluated model:

$$\hat{c} = \arg\max_{c^i \in C} P_\theta(c^i|v, q) = \arg\max_{c^i \in C} \sum_{t=1}^{t_c} P_\theta(c^i_t|v, q, c^i_{<t}) \tag{1}$$

where $P_\theta(c^i_t|v, q, c^i_{<t})$ is parameterized by the causal-LLM-based LVLMs and $\{c^i_1, ..., c^i_{t_c}\}$ is the tokenized sequence of $c^i$. For multiple-choice problem assessment, we provide both the black-box generation evaluation results and the white-box likelihood evaluation results.

### 3.3.2   INSTABILITY-AWARE EVALUATION

As demonstrated in previous work (Xu et al., 2022; Zeng et al., 2023), LLM-based models are sensitive to the different but equivalent instructions. In ReForm-Eval, instability-aware evaluation

| Model | Generation Evaluation | | | | | | | | | Likelihood Evaluation | | | | | | |
| | Perception | | | Cognition | | | | | $\bar{R}$ | Perception | | Cognition | | | | $\bar{R}$ |
| | CG | FG | STP | Spatial | VGR | Dialog | CMI | Desc | | CG | FG | Spatial | VGR | Dialog | CMI | |
| BLIP-2$_F$ | 69.4 | 76.6 | 38.1 | 43.2 | 73.3 | **61.8** | 66.9 | **74.3** | 2 | 60.7 | 74.4 | 51.1 | 69.8 | 62.6 | 58.9 | 4 |
| InstructBLIP$_F$ | **71.2** | **78.1** | **41.2** | **46.1** | **73.9** | 60.6 | **71.4** | 43.8 | 2 | 60.4 | 75.6 | 51.2 | 71.0 | 67.2 | 55.5 | 4 |
| InstructBLIP$_V$ | 69.1 | 70.8 | 40.7 | 44.4 | 63.0 | 48.6 | 53.8 | 27.3 | 4 | 58.5 | 77.8 | 52.3 | **73.5** | **68.7** | 55.4 | 3 |
| LLaVA$_V$ | 28.7 | 34.4 | 18.4 | 28.7 | 44.0 | 35.6 | 47.3 | 36.8 | 11 | 61.0 | 70.3 | 42.4 | 58.9 | 52.3 | 48.0 | 8 |
| LLaVA$_{L_2}$ | 48.3 | 59.8 | 21.5 | 41.2 | 59.7 | 46.3 | 49.9 | 39.5 | 6 | 49.9 | 65.6 | 47.4 | 56.7 | 48.6 | 49.7 | 11 |
| MiniGPT4 | 46.2 | 53.2 | 33.0 | 34.6 | 45.6 | 39.5 | 45.4 | 47.5 | 7 | 54.9 | 70.6 | 49.2 | 57.3 | 54.1 | 50.9 | 8 |
| mPLUG-Owl | 42.0 | 37.2 | 39.8 | 26.8 | 37.5 | 35.2 | 40.4 | 44.7 | 11 | 57.9 | 66.1 | 48.6 | 54.3 | 45.5 | 49.8 | 10 |
| PandaGPT | 28.2 | 34.6 | 4.5 | 33.3 | 41.9 | 34.1 | 36.6 | 1.6 | 14 | 42.3 | 47.4 | 39.4 | 43.3 | 41.5 | 37.0 | 16 |
| IB-LLM | 29.2 | 32.7 | 8.2 | 35.6 | 36.7 | 35.3 | 36.6 | 27.6 | 13 | 49.6 | 54.4 | 46.1 | 50.3 | 39.5 | 45.6 | 15 |
| LA-V2 | 33.2 | 30.8 | 24.2 | 23.8 | 36.3 | 35.4 | 41.1 | 36.0 | 13 | 42.7 | 61.4 | 48.6 | 54.1 | 43.4 | 49.9 | 12 |
| mmGPT | 30.4 | 30.3 | 16.7 | 26.9 | 33.0 | 31.8 | 38.2 | 27.7 | 14 | 52.6 | 62.4 | 47.2 | 56.2 | 43.1 | 44.1 | 13 |
| Shikra | 47.2 | 47.5 | 8.3 | 33.3 | 41.2 | 35.2 | 44.5 | 31.8 | 11 | 60.9 | 66.8 | 45.5 | 58.5 | 59.5 | **59.3** | 7 |
| Lynx | 59.5 | 62.6 | 18.6 | 40.2 | 58.4 | 47.0 | 53.0 | 60.7 | 5 | **66.1** | 76.2 | **53.9** | 69.9 | 60.0 | 57.4 | 3 |
| Cheetor$_V$ | 52.0 | 50.3 | 25.9 | 30.6 | 49.9 | 40.3 | 47.4 | 61.6 | 8 | 56.1 | 69.0 | 48.4 | 58.7 | 57.6 | 50.6 | 8 |
| Cheetor$_{L_2}$ | 46.5 | 51.4 | 18.8 | 34.5 | 54.4 | 40.6 | 44.0 | 43.9 | 8 | 61.6 | 56.1 | 48.7 | 57.5 | 46.8 | 47.2 | 11 |
| BLIVA | 41.7 | 43.4 | 40.8 | 33.3 | 42.4 | 39.8 | 45.2 | 52.5 | 8 | 64.9 | **78.2** | 51.7 | 72.9 | 68.1 | 53.7 | **2** |

Table 1: General evaluation results of LVLMs across different capability dimensions. "CG", "FG", "CMI", and "Desc" are respectively short for coarse-grained perception, fine-grained perception, cross-modal inference, and description. "$\bar{R}$" represents the average rank across dimensions.

is thus introduced. For each task, multiple (more than five) instruction templates are manually designed. Each sample is tested multiple times with different templates and shuffled options if it is a multiple-choice question. The final result is based on the average of the multiple tests.

To directly characterize the instability of models, we further introduce a metric. For a multiple-choice problem with answer options $C = \{c^i\}_{i=1}^N$, the empirical prediction distribution of a model can be calculated from the $M$ tests as $p_i = \frac{1}{M} \sum_{j=1}^M \mathbb{1}(\hat{c}_j = c^i)$ where $\hat{c}_j$ is the prediction of the $j$-th test. Then the instability is measured by the entropy of the prediction distribution: $e = -\sum_{i=1}^N p_i \log(p_i)$. Larger $e$ indicates higher uncertainty in the predictions for that sample. For text-generation tasks, instability is not accessible as the prediction distribution is not directly measurable.

## 4 EXPERIMENTS

### 4.1 IMPLEMENTATION DETAILS

Based on ReForm-Eval, we evaluate 16 models with around 7B parameters that are trained with 13 different methods, including BLIP-2 (Li et al., 2023b), InstructBLIP (Dai et al., 2023), LLaVA (Liu et al., 2023b), MiniGPT4 (Zhu et al., 2023), mPLUG-Owl (Ye et al., 2023), PandaGPT (Su et al., 2023), ImageBind-LLM (IB-LLM) (Han et al., 2023), LLaMA-Adapter V2 (LA-V2) (Gao et al., 2023), multimodal-GPT (mmGPT) (Gong et al., 2023), Shikra (Chen et al., 2023), Lynx (Zeng et al., 2023), Cheetor (Li et al., 2023a), BLIVA (Hu et al., 2023). Details of the methods are introduced in Appendix B.2. All experiments are conducted in the same software and hardware environment to ensure fairness. For specific parameter settings, please refer to Appendix B.1.

**Notations** For models with multiple variants based on different backbones, we use subscripts to denote the backbone used: $F$, $V$, $L$, and $L_2$ represent FlanT5, Vicuna, LLaMA, and LLaMA2, respectively. For multiple-choice problems, "Generation Evaluation" and "Likelihood Evaluation" are respectively based on the black-box and white-box strategies. For each task under different strategies, the best result is marked in **bold** while the runner-up is underlined.

### 4.2 GENERAL PERFORMANCE

Table 1 presents the comprehensive performance of each model across dimensions, from which several insights can be gleaned. (1) BLIP-2 and InstructBLIP continue to hold the top-2 positions in most dimensions, but in some individual dimensions, Lynx, BLIVA, and Shikra also take the lead. (2) It's worth noting that the effectiveness of models like BLIVA and Lynx only becomes apparent when using likelihood evaluation. We suspect this is attributed to the instruction-following ability of models, please refer to Section 4.3.4 for a detailed analysis. (3) Compared to models based on CLIP visual encoders, PandaGPT and IB-LLM, which are based on the ImageBind encoder, exhibit relatively poorer performance in image-text tasks. Meanwhile, most top-performing models utilize Vicuna and FlanT5 as the backbone. Further analysis is available in Section 4.3.1 regarding the

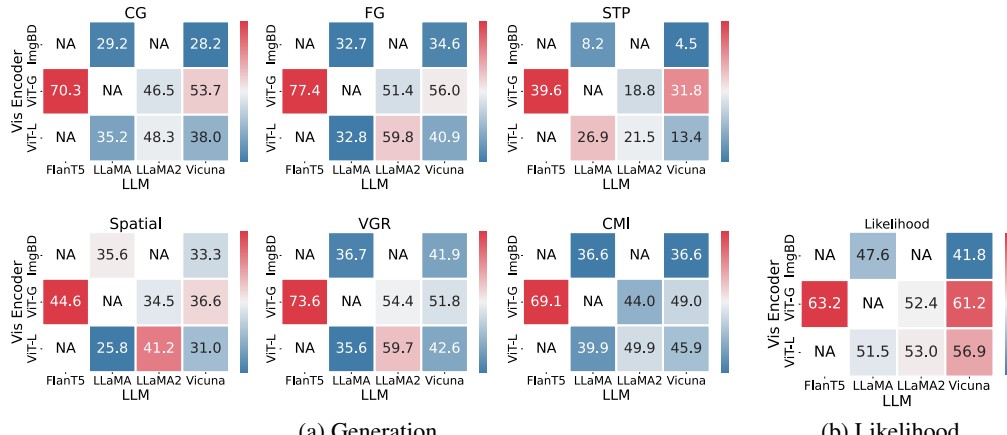

(a) Generation      (b) Likelihood

Figure 3: The influence of different language and visual backbones. For generation evaluation, we average the results of various models based on the backbone used. To better visualize the results, we selected heatmaps across six dimensions (dialog and desc are omitted). For likelihood evaluation, we further compute the average score across dimensions since the performance trend is consistent. Note that "ImgBD" is short for ImageBind in this figure.

| Visual Backbone | | ImageBind | | ViT-G | | ViT-L | | |
|---|---|---|---|---|---|---|---|---|
| Connection Arch | | BindNet+Gate | Linear | Perceiver | Q-Former | Adapter | Linear | Perceiver |
| **Generation** | Perception | 23.4 | 22.4 | 46.9 | 50.4 | 29.4 | 34.9 | 32.7 |
| | Cognition | 34.3 | 29.5 | 51.9 | 49.3 | 34.5 | 41.0 | 34.2 |
| **Likelihood** | Perception | 31.0 | 31.4 | 61.1 | 58.6 | 32.0 | 44.3 | 35.0 |
| | Cognition | 36.0 | 36.5 | 49.7 | 49.1 | 34.2 | 42.3 | 33.7 |

Table 2: Average evaluation performance categorized by connection modules (see Table 7 for more details) and visual backbones under generation and likelihood strategy.

impact of model architecture and backbones. (4) Apart from the architecture, a common characteristic among BLIP-2, InstructBLIP, Lynx, and BLIVA is the use of relatively high-quality data during pre-training. For data-related analysis, please refer to Section 4.3.2.

## 4.3 COMPREHENSIVE ANALYSIS

### 4.3.1 EXPLORE THE MODEL ARCHITECTURE

**Model Backbone** To gain a better insight into the backbone influence, we group models based on the backbone, as illustrated in Figure 3. For language backbones, Vicuna-based models outperform LLaMA-based models, whereas LLaMA2 and Vicuna excel in different dimensions. Under likelihood evaluation, Vicuna consistently performs better. FlanT5 seems the best, as the related models are BLIP-2 and InstructBLIP. Regarding visual backbones, ViT-G (from EVA-CLIP (Sun et al., 2023)) generally outperforms ViT-L (from CLIP (Radford et al., 2021)), which in turn outperforms ImageBind. Furthermore, LLaMA2 tends to favor smaller visual encoders like ViT-L, while Vicuna performs better when paired with larger visual encoders like ViT-G.

**Connection Module** We further analyze the effect of connection modules in Table 2. ImageBind appears to perform subpar regardless of the choice of connection module. For larger visual backbones like ViT-G, both Perceiver and Q-Former show decent performance. For smaller visual backbones (ViT-L), Linear connection module is consistently better.

In summary, **language backbones are supposed to possess strong instruction-following capabilities. As for visual backbones, it's advisable to choose ViT-G and carefully select a connection module compatible with the corresponding visual backbone.** Besides, different model architectures result in varying parameter quantities. We discuss the impact in Appendix C.3.

### 4.3.2 EXPLORE THE DATASET

**High-Quality Pre-training Dataset** MSCOCO (Lin et al., 2014) is a typical high-quality human-annotated dataset that is commonly used during pre-training. To quantitatively assess its impact,

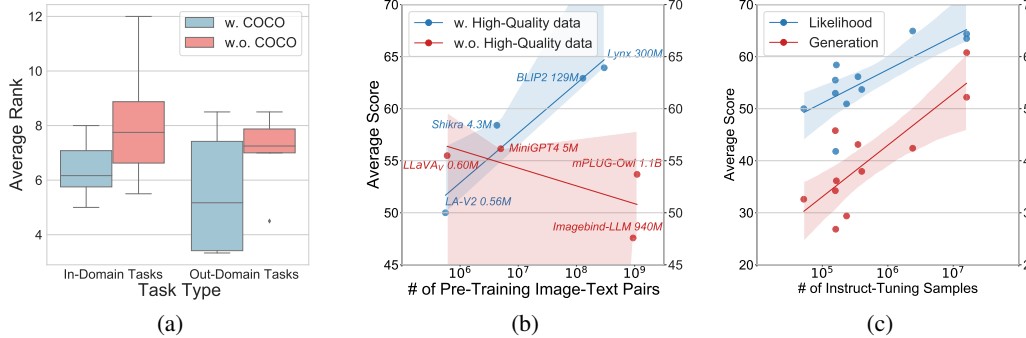

(a)                         (b)                         (c)

Figure 4: The influence of datasets in the pre-training and instruct-tuning stages. (a) compares the average rank of models pre-trained with and without the MSCOCO dataset. (b) shows the relationship between the scale of pre-training data and the average performance score of models grouped by data quality. (c) shows the relations between the number of instruct-tuning samples and the average score. The shaded area represents the 95% confidence interval.

| Backbone | LLaMA-7B | | Vicuna-7B | | Vicuna-7B+ | | FlanT5-xl | | Vicuna-7B+LoRA |
|---|---|---|---|---|---|---|---|---|---|
| Model | LA-V2 | mPLUG-Owl | MiniGPT4 | Cheetor | Shikra | LLaVA | BLIP-2 | InstructBLIP | PandaGPT |
| Hit Rate | 85.14 | 62.86 | 100 | 99.97 | 65.42 | 85.32 | 100 | 99.99 | 99.41 |
| Hit Rate+ | 100 | 100 | 100 | 100 | 100 | 100 | 100 | 100 | 99.97 |

Table 3: Instruction-following ability of LVLMs in multiple-choice problems. "Vicuna-7B+" indicates the LLM backbone is fine-tuned. "Hit Rate" and "Hit Rate+" represent the format hit rate without and with in-context samples, respectively.

we compare the average performance between models pre-trained with and without MSCOCO. As shown in Figure 4 (a), MSCOCO not only helps with in-domain tasks but also enhances generalization results on out-domain tasks. Therefore, to effectively align cross-modal representations during pre-training, it is crucial to include such high-quality pre-training data.

**Scaling Up Pre-Training Dataset** To scale up the LVLM training, it is necessary to utilize image-text pairs crawled from the web. Figure 4 (b) compares two groups of models: the red-marked group uses data filtered based on rules or CLIP, such as CC (Sharma et al., 2018) and LAION (Schuhmann et al., 2021), while the blue-mark utilizes relatively high-quality data including aforementioned annotated data and synthetic captions from BLIP (Li et al., 2022). Results show that it is more effective to scale up utilizing synthetic data, resulting in a desired increasing curve. We believe the reason behind this is that synthetic captions are cleaner and more associated with images. While the diversity of data may be impaired, the generalizable backbones mitigate the negative impact.

**Instruct-Tuning Dataset** We also explore the impact of the number of instruct-tuning samples. The fitted curve in Figure 4 (c) demonstrates that increasing the number of instruct-tuning samples leads to improved performance of LVLMs.

In general, **the quality of pre-training data and the scale of instruct-tuning samples are crucial factors for improving LVLMs**. Appendix C.4 provides the complete data used in this section.

### 4.3.3 EFFECT OF IN-CONTEXT SAMPLE

To demonstrate the effectiveness of the black-box evaluation strategy introduced in Section 3.3.1. We assess LVLMs' ability to follow multiple-choice instructions under different strategies. The experiments are conducted in the re-formulated VQA v2, a response is considered as hitting the format if it includes the option mark like "(A)". Some results are listed in Table 3. It is obvious that the ability is tightly related to the backbone. LVLMs based on raw LLaMA inherit the weak instruction-following ability of the backbone. At the same time, fine-tuning the full backbone results in catastrophic forgetting of the capability, while LoRA-based fine-tuning does not. However, **in-context samples can effectively provide format information and guide LVLMs to respond in the desired format**, facilitating automated evaluation. The complete results are in Table 22.

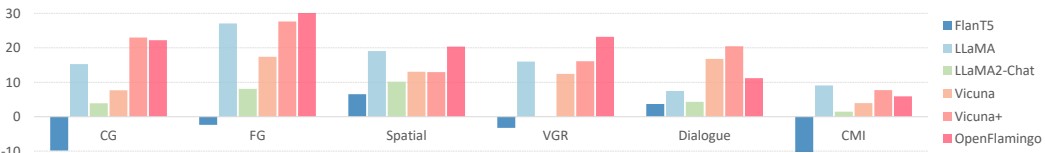

Figure 5: Performance gap of models under different evaluation strategies, grouped and averaged based on the language backbone. The vertical axis indicates how much the likelihood evaluation surpasses the generation evaluation, truncated for simplicity. "+" indicates fine-tuned backbones.

### 4.3.4 GENERATION V.S. LIKELIHOOD EVALUATION

For generation evaluation, the results reflect the coupling of the multi-modal understanding capability and the instruction-following capability. Meanwhile, likelihood evaluation directly probes the generative models and relaxes the requirement for instruction following.

As shown in Figure 5, likelihood evaluation yields better results than generation evaluation in most cases, even when LVLMs are guided through in-context learning. This indicates that **most LVLMs have limited instruction-following capability, further hindering downstream performance**. We believe the primary factor behind this is the LLM backbone, as models based on FlanT5 and LLama2-Chat have the least performance gap between likelihood and generation evaluation in all the dimensions, FlanT5-based models even perform better using generation evaluation in CG, FG, VGR, and CMI. To address the issue, LVLMs should leverage stronger backbones or introduce sufficiently diverse data for instruct tuning, as done in FlanT5. Besides, the comparison between Vicuna and Vicuna+ demonstrates that **multi-modal instruct tuning the backbone currently can not improve the instruction-following capability of LVLMs**.

### 4.3.5 BEHIND THE INSTABILITY

To investigate the source of instability, we conduct experiments on ScienceQA by applying three types of perturbations separately to LVLMs, including random instructions, shuffling option orders, and random option marks (uppercase, lowercase, or numeric).

| Instability Source | Generation | Likelihood |
|---|---|---|
| Instruction | 0.1607 | 0.0492 |
| Option Order | 0.5523 | NA |
| Option Mark | 0.3295 | NA |

Table 4: Average instability by three types of random perturbations across all models.

As illustrated in Table 4, shuffling the option order results in the highest instability, highlighting a misunderstanding of the option contents. Similar to MM-Bench (Liu et al., 2023c), we observe that most models exhibit some degree of preference for specific options (refer to Appendix C.6 for more details). Our in-depth finding is that option preference reduces the instability from random instructions and random option marks, but increases the instability from random option orders. The randomness of instruction has the least effect, suggesting that LVLMs can reasonably comprehend the carefully crafted instructions. With likelihood evaluation, the instability is significantly lower because it is a white-box method that directly probes generative models without the need for random sampling during generation. These phenomenons are common to all models, the complete results are in Appendix C.5. In summary, **current LVLMs are unstable and sensitive to subtle changes in the prompt, especially during black-box evaluations**.

## 5 CONCLUSION

In this paper, we propose to re-formulate task-oriented multi-modal benchmarks to evaluate LVLMs. By systematically collecting and efficiently re-formulating 61 benchmarks into unified formats that are compatible with LVLMs, we construct a benchmark, ReForm-Eval, which covers 8 capability dimensions. Compared with recently constructed benchmarks for LVLMs, ReForm-Eval provides more data without the need for manual annotation. Additionally, we design dependable automated evaluation methods based on the unified formats, ensuring an impartial assessment of different LVLMs. Leveraging ReForm-Eval, we conduct an exhaustive evaluation of various LVLMs and delve into the factors influencing their performance. Generally, ReForm-Eval serves as a reliable tool for quantitative analysis of LVLMs, aiding in the research and development of LVLMs.

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

## A  BENCHMARK CONSTRUCTION

In this section, we introduce the collected datasets and the corresponding re-formulation procedures in detail. The statistics of the re-formulated datasets are provided in Table 5 and Table 6.

### A.1  COARSE-GRAINED PERCEPTION

For the Flowers102 dataset, we employ the complete validation set for evaluation purposes. However, for CIFAR10, ImageNet-1K, Pets37, and VizWiz, we perform random subsampling of 10%. Concerning the TDIUC dataset, given that certain models in their training phase utilized a portion of the TDIUC dataset originating from the Visual Genome, we initially exclude this subset of data to prevent potential data leakage. Subsequently, we apply a shuffling operation to the entire TDIUC dataset and perform equidistant sampling, resulting in the selection of 2.5% of the sport_recognition data (TDIUC$_{sport}$) and 1% of the scene_recognition data (TDIUC$_{scene}$). In the case of MEDIC(Alam et al., 2023), we sample an equal number of samples from each label to balance the answer distribution.

For Flowers102 and Pets37, we randomly select three incorrect class labels, in addition to the correct label, from their original set of categories to form multiple-choice question options. For the TDIUC, we aggregate all answers for the same task to create an answer pool, and then utilize the same approach above to construct four answer options for multiple-choice questions.

For ImageNet-1K, we calculate similarities within its own set of 1000 categories using WordNet and selected the four options with the highest similarity to the correct class as choices (the highest one must be the right answer, and we need to get them out of order).

For CIFAR10, we initially employ WordNet to identify synonyms of the answers that are semantically related but not synonymous. These synonyms are then ranked based on their similarity. Subsequently, we manually adjust some of the less common candidate options. Finally, we likewise select the top four options with the highest similarity as all choices.

As for VizWiz, we re-formulate it into two benchmarks: VizWiz$_2$ as a binary classification task to determine whether there is any quality issue with the image. VizWiz$_4$ as a 4-choice question, requiring the model to determine the exact reason for the quality issue. We sort the issues related to image quality based on the number of votes in the annotations, the top one is considered the true label while the second to fourth options serve as negative choices.

For MEDIC (Alam et al., 2023), it is re-formulated to MEDIC$_{dts}$, a benchmark for disaster type selection (dts), we directly use all seven classification labels as choice options.

### A.2  FINE-GRAINED PERCEPTION

For TDIUC (Kafle & Kanan, 2017), we initially exclude the subset sourced from Visual Genome (Krishna et al., 2017) to prevent evaluation on the training data. Then, we shuffle the entire dataset and conducted an equidistant sampling strategy for task sample balance. Specifically, we sample 1% of the data for color (TDIUC$_{color}$), detection(TDIUC$_{detection}$), and counting tasks (TDIUC$_{counting}$), and 2.5% for position tasks. As for the utility task (TDIUC$_{utility}$), we retain and utilized all 171 data samples. For answer options, we uniformly count all answers within the data and randomly selected three options other than the correct answer to form all four choices.

Regarding RefCOCO (Yu et al., 2016), we re-formulate the referring expression selection (RefCOCO$_{res}$) task, in which the LVLMs are supposed to select the correct referring expression from the options based on the image region in the bounding box. We sample an equal number of samples from each object category, in order to balance the correct referring expression categories appearing in the questions. As for negative options in each question, we sample the negative referring expression from a distinct subcategory within the same category as the positive sample.

For MSCOCO (Lin et al., 2014), we re-formulate four tasks: object counting (counting), multiple class identification (MSCOCO$_{mci}$), grounded object identification (MSCOCO$_{goi}$) and missing object selection (MSCOCO$_{mos}$) for object-level evaluation. The multiple class identification task aims to evaluate the LVLM's ability of object classification. Further, the grounded object identification and missing object selection tasks concentrate on object perception within a specified region of interest.

| Task Name | Dataset Name | Data Source | Datset Split | # of Images | # of Samples |
|---|---|---|---|---|---|
| Coarse-grained Perception | Flowers102 | Flowers102 | val | 818 | 818 |
| | CIFAR10 | CIFAR10 | test | 10000 | 10000 |
| | ImageNet-1K | ImageNet-1K | val | 50000 | 50000 |
| | Pets37 | Pets37 | test | 3669 | 3669 |
| | VizWiz$_2$ | VizWiz | val | 4049 | 4049 |
| | VizWiz$_4$ | VizWiz | val | 2167 | 2167 |
| | TDIUC$_{sport}$ | TDIUC | val | 6001 | 8696 |
| | TDIUC$_{scene}$ | TDIUC | val | 9219 | 21320 |
| | MEDIC$_{dts}$ | MEDIC | test | 15688 | 15688 |
| Fine-grained Perception | MSCOCO$_{mci}$ | MSCOCO | val2017 | 2323 | 3600 |
| | MSCOCO$_{goi}$ | MSCOCO | val2017 | 2404 | 3600 |
| | MSCOCO$_{mos}$ | MSCOCO | val2017 | 2479 | 2479 |
| | TDIUC$_{color}$ | TDIUC | val | 18808 | 38267 |
| | TDIUC$_{utility}$ | TDIUC | val | 162 | 171 |
| | TDIUC$_{postiion}$ | TDIUC | val | 7131 | 9247 |
| | TDIUC$_{detection}$ | TDIUC | val | 21845 | 29122 |
| | TDIUC$_{counting}$ | TDIUC | val | 26166 | 41991 |
| | RefCOCO$_{res}$ | RefCOCO | val | 9397 | 34540 |
| | MSCOCO$_{count}$ | MSCOCO | val2014 | 513 | 513 |
| Scene Text Perception | CUTE80 | CUTE80 | all | 288 | 288 |
| | IC15 | IC15 | test | 1811 | 1811 |
| | IIIT5K | IIIT5K | test | 3000 | 3000 |
| | COCO-Text | COCO-Text | val | 9896 | 9896 |
| | WordArt | WordArt | test | 1511 | 1511 |
| | TextOCR | TextOCR | val | 3000 | 3000 |
| | Grounded IC15 | IC15 | val | 221 | 849 |
| | Grounded COCO-Text | COCO-Text | val | 1574 | 3000 |
| | Grounded TextOCR | TextOCR | val | 254 | 3000 |
| | FUNSD | FUNSD | test | 47 | 588 |
| | POIE | POIE | test | 750 | 6321 |
| | SROIE | SROIE | test | 347 | 1388 |
| | TextVQA | TextVQA | val | 3023 | 4508 |
| | DocVQA | DocVQA | val | 1286 | 5312 |
| | OCR-VQA | OCR-VQA | test | 3768 | 3944 |

Table 5: Dataset statistics of visual perception tasks in ReForm-Eval.

The former allows models to assess which object exists within the given bounding box of the image, while the latter asks models to judge which object disappears within all the given bounding boxes of the image.

For the multiple class identification and grounded object identification tasks, we randomly sample 300 object annotations from each super-category in the valid split to ensure balance. This results in a total of 3600 evaluation data samples for each task. For the mos task, we filter out the objects with the height and width of their bounding boxes smaller than 50 and finally get 2479 samples. As for options generation, we employ a hierarchical strategy. For the multiple class identification task, we begin by randomly selecting the object class from within the super-category of the target object. If there are insufficient options, we broaden our selection to all object categories. In tasks related to region, our initial step is to randomly choose object categories present in the image but do not meet the requirement specified in the question. In cases where this is not possible, we follow the sampling procedure used in the multiple-class identification task. The examples of these grounded fine-grained tasks as shown in Table 6. The counting task has the same setting as the counting task in the TDIUC dataset.

## A.3 SCENE TEXT PERCEPTION

For OCR, we use 6 original OCR benchmarks (including CUTE80 (Risnumawan et al., 2014), IC15 (Karatzas et al., 2015), IIIT5K (Mishra et al., 2012), COCO-Text (Mishra et al., 2012), Wor-dArt (Xie et al., 2022) and TextOCR (Singh et al., 2021)) as the evaluation tasks. Current OCR

| Task Name | Dataset Name | Data Source | Datset Split | # of Images | # of Samples |
|---|---|---|---|---|---|
| Spatial Understanding | CLEVR | CLEVR | val | 5726 | 6900 |
| | VSR | VSR | test | 1074 | 1811 |
| | MP3D-Spatial | MP3D | - | 3341 | 4735 |
| Cross-Modal Inference | COCO$_{itm}$ | MSCOCO caption | val2017 | 5000 | 25014 |
| | COCO$_{its}$ | MSCOCO caption | val2017 | 5000 | 25014 |
| | WikiHow | WikiHow | val | 32194 | 32194 |
| | Winoground | Winoground | all | 800 | 800 |
| | SNLI-VE | SNLI-VE | test | 1000 | 17901 |
| | MOCHEG | MOCHEG | test | 1452 | 3385 |
| Visually Grounded Reasoning | VQA v2 | VQA v2 | val2014 | 15638 | 21441 |
| | GQA | GQA | testdev | 398 | 12578 |
| | Whoops | Whoops | all | 498 | 3362 |
| | OK-VQA | OK-VQA | val | 5032 | 5045 |
| | ScienceQA | ScienceQA | test | 2017 | 2017 |
| | VizWiz | VizWiz | val | 4319 | 4319 |
| | ViQuAE | ViQuAE | test | 1105 | 1257 |
| | K-ViQuAE | ViQuAE | test | 1094 | 1245 |
| | A-OKVQA | A-OKVQA | val | 1122 | 1145 |
| | A-OKVQRA | A-OKVQA | val | 1122 | 1145 |
| | A-OKVQAR | A-OKVQA | val | 1122 | 1145 |
| | ImageNetVC | ImageNetVC | all | 3916 | 4076 |
| Multi-Turn Dialogue | VQA-MT | VQA v2 | val2014 | 1073 | 1073 |
| | VisDial | VisDial | val2018 | 2064 | 2064 |
| Visual Description | COCO | MSCOCO caption | val2017 | 5000 | 5000 |
| | TextCaps | TextCaps | val | 3166 | 3166 |
| | NoCaps | NoCaps | val | 4500 | 4500 |
| | Flickr30K | Flickr30K | test | 1000 | 1000 |

Table 6: Dataset statistics of visual cognition tasks in ReForm-Eval.

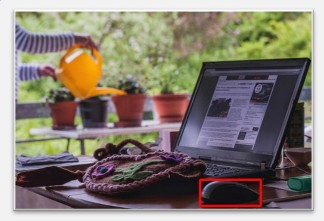

**Question:** Which object does the red bounding box of this image contain among the following options?
**Options**:
(A) person;           (B) potted plant;
(C) mouse;            (D) tv.
**Answer:** (C)

**Grounded Object Identification**

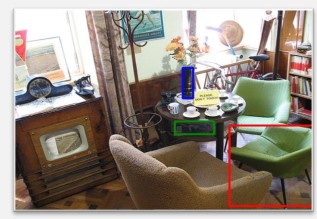

**Question:** Which object from the options that does not exist in bounding boxes of the image?
**Options**:
(A) book;             (B) bicycle;
(C) vase;             (D) chair.
**Answer:** (B)

**Missing Object Selection**

Figure 6: Examples of grounded fine-grained tasks.

benchmarks utilize cropped images containing only target text as visual input sources (Xu et al., 2023; Liu et al., 2023d). To further assess text identification in complex visual contexts, we propose grounded OCR tasks (including gIC15, gCOCO-Text, and gTextOCR). Specifically, we filter out the bounding boxes containing target texts larger than 40x40 for better evaluation. The image, along with the bounding box annotations and the corresponding instruction, will be fed into the model for evaluation, which is similar to the grounded fine-grained tasks (i.e. MSCOCO$_{goi}$).

For KIE, we utilize the test splits of 3 benchmarks (including SROIE (Huang et al., 2019), POIE (Kuang et al., 2023) and FUNSD (Jaume et al., 2019)) as the evaluation tasks.

And for OCR-based VQA, we use 3 benchmarks (including TextVQA (Singh et al., 2019), DocVQA (Mathew et al., 2021) and OCR-VQA (Mishra et al., 2019)) as the evaluation tasks. We filter out the question-answer pairs that need to be inferred based on the scene texts.

## A.4 VISUALLY GROUNDED REASONING

For VQAv2 (Goyal et al., 2017), we sample 10% for reformulation owing to the extremely large population. Besides, since ViQuAE (Lerner et al., 2022) provides relevant knowledge information for each question, we additionally construct K-ViQuAE with knowledge as context, which assesses models' reasoning ability hierarchically with ViQuAE (Lerner et al., 2022). For ScienceQA (Lu et al., 2022), only 2017 questions of all the 4241 test set are paired with an image, which are selected in our benchmark. Besides, original A-OKVQA (Schwenk et al., 2022) gives rationales for answering each question, therefore we construct A-OKVQRA and A-OKVQAR for hierarchical evaluation.

For VQAv2 (Goyal et al., 2017), GQA (Hudson & Manning, 2019), OK-VQA (Marino et al., 2019), VizWiz (Gurari et al., 2018), ViQuAE (Lerner et al., 2022) and Whoops (Bitton-Guetta et al., 2023), ChatGPT is employed to generate appropriate negative options, and the prompt template for querying is:

> You are a multiple-choice generator. Given a question and an answer, you need to generate three additional incorrect options while ensuring their plausibility and confusion.
> Question: {question}
> Answer: {correct answer}

Note that for yes or no questions, the negative option is directly derived as no or yes, and ChatGPT is not employed.

While ImageNetVC (Xia et al., 2023) randomly selects 3 candidate options from the correspondent answer set with the commonsense type of each question. As for ScienceQA (Lu et al., 2022) and A-OKVQA (Schwenk et al., 2022), they adopt their original options because of the original single-choice formulation.

As for A-OKVQAR, the prompt template for querying ChatGPT to generate negative rationales is:

> You are a multiple-choice generator. Given a question and an answer, along with a rationale for that answer, you need to generate 3 counterfactual rationales. These counterfactual rationales should be contextually relevant while also being sufficiently distinct from the correct rationale.
> Question: {question}
> Answer: {correct answer}
> Rationale: {rationale}

## A.5 SPATIAL UNDERSTANDING

For CLEVR (Johnson et al., 2017), we filter out the question types that do not involve spatial relations and randomly select 300 samples from each question type related to spatial relations. For different question types, we randomly select false options from their corresponding answer sets. In cases where some question types have insufficient options, we add 'Not sure' and 'Unknown' as false options to maintain the number of four options.

For VSR (Liu et al., 2023a), the original dataset comprises captions that describe true or false spatial relations among objects in the corresponding images. We select image-caption pairs from the test split where the spatial descriptions are right and use them for our evaluation tasks. The false options are generated by randomly sampling different spatial relations from the test split.

MP3D (Chang et al., 2017) also known as Matterport3D, comprises a large-scale collection of RGB-D images captured from nearly 10,800 real indoor viewpoints with 50,811 object instance annotations. Based on this dataset, we extract two types of spatial relations to re-formulate our benchmark MP3D-Spatial: object-object level relations (left, right, above, under, on top of, and next to) and object-observer level relations (far and close). For spatial relations 'on top of' and 'next to,' we use

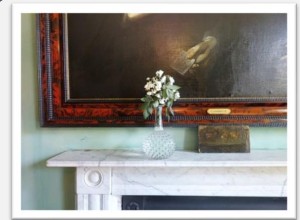

**Question:** Describe the spatial connection between vase and mantel within the image.
**Options**:
(A) The vase is inside the mantel;
(B) The vase is right of the mantel;
(C) The vase is next to the mantel;
(D) The vase is on the top of the mantel.
**Answer:** (D)

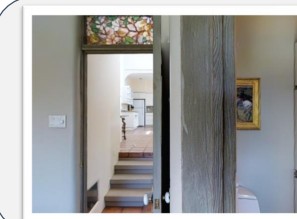

**Question:** In the image, point out the object that has the greatest distance from you.
**Options:**
(A) picture;
(B) refrigerator;
(C) stairs;
(D) unknown.
**Answer:** (B)

Figure 7: Examples of spatial relation judgment in MP3D-Spatial.

1,135 annotated samples for our task. For other relations, we utilize both bounding box information and depth to determine the spatial relationships and extract 600 samples for each type of relation. As for false options, we randomly select non-matching spatial relations to serve as the incorrect options for our reformulated task. The examples of the re-formulated data are shown in Figure 7

### A.6 Cross-Modal Inference

In this section, we consider two kinds of tasks, including image text matching and visual entailment.

For MSCOCO (Lin et al., 2014), we re-formulate two tasks, including COCO image text matching (COCO$_{itm}$) and COCO image text selection (COCO$_{its}$). The matching task requests LVLMs to determine whether the given image and text are matched. The selection task instructs LVLMs to select the best-matched caption for the given image. We randomly sample image and text pairs as positive samples. For each image, we first find the negative images that have distinct but similar object categories. For each negative image, we find the most similar captions with the positive caption according to the object appearing in the sentence.

WikiHow (Koupaee & Wang, 2018) provides introductions to common skills. Within each skill, there are multiple crucial tasks, and each task is composed of several steps. We re-formulate the Wikihow image text selection task, in which given the task name, the LVLMs are supposed to choose the matching step description. We randomly sample visual and textual descriptions of the steps to form multiple-choice questions. To mine the hard negative options, we try to randomly take three samples from the same task as the positive sample. If the negative samples from the task level are insufficient, we then select some from the skill level and dataset level in turn.

For Winoground (Thrush et al., 2022), we re-formulate a caption selection task, which requires models to choose the correct caption. Since Winoground has captions for each image pair that have exactly the same words but in a different order, the options are the captions for every image pair.

For SNLI-VE (Xie et al., 2019), we re-formulate the visual entailment task, in which the LVLMs are required to determine whether the text can be inferred based on the image clues and should give answer of uncertainty when the evidence is insufficient. The options of multiple-choice question comprise "yes", "not sure" and "no". To balance the correct answer distribution, for each image, we sample an equal number of samples from each label.

For MOCHEG (Yao et al., 2023), we re-formulate the visual and textual entailment task, in which the LVLMs are supposed to determine whether the claim can be infered based on the visual and textual evidence and judge out whether the evidences are insufficient. The options consist of "supported", "refuted" and "not enough information".

### A.7 Visual Description

We re-formulate the image captioning task from four dataset including MSCOCO (Lin et al., 2014), TextCaps (Sidorov et al., 2020), NoCaps (Agrawal et al., 2019) and Flickr30K (Young et al., 2014).

In this task, the LVLMs are expected to generate a brief description for given image. Among these dataset, TextCaps additionally examines the optical character recognition capability of LVLMs by requesting models to pointing out the text in the image. We randomly sample these datasets for evaluation.

### A.8 MULTI-TURN DIALOGUE

To mimic a naive setup, we construct VQA-MT (VQA Multi-Turn) by considering multiple questions for the same image and gathering them into a multi-turn conversation. For VQA-MT, different images are accompanied by different amounts of questions in the re-formulated VQA v2 (Goyal et al., 2017), only the images with more than 2 questions are kept. For images with more than 10 questions, only the first 10 questions are kept. All questions for the same image are arranged into a dialogue without inter-round dependencies. In the filtered dataset, there are 1073 image-dialogue pairs. The negative options are directly adopted from the re-formulated VQA v2.

As for VisDial (Das et al., 2017), there is a 10-turn QA dialogue for each image. the original datasets provide 100 options for each question while. The prompt template for querying GPT-3.5 to generate negative options is:

> I will provide a question with the correct answer, please give me 3 incorrect options to help me get a single-choice question.
> Question: {question}
> Answer: {correct answer}

Different from the original VisDial to perform offline dialogue (the history contains correct answers), we perform online dialogue (the history contains the previous output of the models). To further investigate whether the performance of LVLMs changes with an increasing number of dialogue turns, we calculate the correlation coefficient between the accuracy and the number of dialogue turns.

## B EVALUATION DETAILS

### B.1 IMPLEMENTATION DETAILS

Our benchmark and the evaluation framework are PyTorch-based. All experiments are conducted on 8 Tesla V100 GPUs. During the evaluation, half precision is used to accelerate the process.

To ensure fair comparisons between LVLMs, we try our best to keep the parameter setting aligned with the demo code provided by the original codebase. However, we limit the maximum number of tokens a model can generate for all LVLMs. It is set to 30 for most questions except the image-caption task where it is set to the upper quartile (the 75th percentile) of the reference caption length in the corresponding datasets. all input texts are formulated into conversations as required by different models, using the same system messages, roles, and separators. As for the image input, we only consider single-image inputs, we use the same preprocess method mentioned in the original paper except for Lynx, which utilizes $420 \times 420$ input resolution and we still use $224 \times 224$ for a fair comparison.

It is worth noting that ReForm-Eval comprises a total of over 500,000 evaluation instances across over 300,000 images, and considering the need for multiple tests for each instance, this results in significant computational cost. To this end, we further construct a subset by sub-sampling 10% data from the whole ReForm-Eval. All experiments conducted in this paper are based on the subset. We will open-source both the subset we use and the complete data for the research community.

### B.2 MODELS

In this section, we introduce the evaluated LVLMs in detail. For each method, we identify the version assessed in this paper if multiple variants are provided by the method. Additionally, we summarize the architecture of LVLMs in Table 7 and the datasets they use in Table 8.

| Model | Model Architecture | | | | | |
| --- | --- | --- | --- | --- | --- | --- |
| | Vis Encoder | LLM | Connection Module | #oP | #oTP | #oVT |
| BLIP-2 | ViT-G/14 | FlanT5-XL | Q-Former | 3.94B | 106.7M | 32 |
| InstructBLIP$_F$ | ViT-G/14 | FlanT5-XL | Q-Former | 4.02B | 187.2M | 32 |
| InstructBLIP$_V$ | ViT-G/14 | Vicuna-7B | Q-Former | 7.92B | 188.8M | 32 |
| LLaVA$_V$ | ViT-L/14 | Vicuna-7B | Linear | 7.05B | 6.74B | 256 |
| LLaVA$_{L_2}$ | ViT-L/14 | LLaMA2-7B | Linear | 7.05B | 6.74B | 256 |
| MiniGPT4 | ViT-G/14 | Vicuna-7B | Q-Former+Linear | 7.83B | 3.1M | 32 |
| mPLUG-Owl | ViT-L/14 | LLaMA-7B | Perceiver | 7.12B | 384.6M | 65 |
| PandaGPT | ImageBind | Vicuna-7B+LoRA | Linear | 7.98B | 37.8M | 1 |
| IB-LLM | ImageBind | LLaMA-7B+LoRA+BT | BindNet+Gate | 8.61B | 649.7M | 1 |
| LA-V2 | ViT-L/14 | LLaMA-7B+BT | Linear+Adapter+Gate | 7.14B | 63.1M | 10 |
| mmGPT | ViT-L/14 | LLaMA-7B+LoRA | Perceiver+Gate | 8.37B | 23.5M | 64 |
| Shikra | ViT-L/14 | Vicuna-7B | Linear | 6.74B | 6.44B | 256 |
| Lynx | ViT-G/14 | Vicuna-7B+Adapter | Perceiver | 8.41B | 688.4M | 64 |
| Cheetor$_V$ | ViT-G/14 | Vicuna-7B | Query+Linear+Q-Former | 7.84B | 6.3M | 32 |
| Cheetor$_{L_2}$ | ViT-G/14 | LLaMA2-Chat | Query+Linear+Q-Former | 7.84B | 6.3M | 32 |
| BLIVA | ViT-G/14 | Vicuna-7B | Q-Former+Linear | 7.92B | 194.6M | 32 |

PS: Underlined represents a trainable component. "BT" represents bias-tuning . "BindNet" represents bind network.

Table 7: Model architecture of different LVLMs. "#oP", "#oTP", and "#oVT" are number of total parameters, number of trainable parameters, and number of visual tokens, respectively.

**BLIP-2** BLIP-2 (Li et al., 2023b) is pre-trained in two stages: the representation learning stage and the generative learning stage, where the image encoder and the LLM are frozen and only a lightweight Q-Former is trained for bridging the modality gap. "blip2-pretrain-flant5xl" is evaluated in our experiment.

**InstructBLIP** InstructBLIP (Dai et al., 2023) further extends BLIP-2 with task-oriented instruct tuning, pre-trained with Vicuna using the same procedure as BLIP-2. Additionally, an instruction-aware Q-Former module is proposed in InsturctBLIP, which takes in the instruction text tokens as additional input to the Q-Former. During instruction tuning, only parameters of Q-Former are fine-tuned based on pre-trained checkpoints, while keeping both the image encoder and the LLM frozen. We take "blip2-instruct-vicuna7b" and "blip2-instruct-flant5xl" as evaluation versions.

**MiniGPT-4** MiniGPT4 (Zhu et al., 2023) adds a trainable single projection layer based on BLIP-2 and also adopts a two-stage training approach, where the first stage is pre-training the model on large aligned image-text pairs and the second stage is instruction tuning with a smaller but high-quality image-text dataset with a designed conversational template. During training, the image encoder, the LLM, and the Q-Former are all frozen. "pretrained-minigpt4-7b" is used in our setup.

**LLaVA** LLaVA (Liu et al., 2023b) employs a linear layer to convert visual features into the language embedding space, with a pre-training and instruction tuning stage. During pre-training, both the visual encoder and LLM weights were frozen. Then, keeping only the visual encoder weights frozen, the weights of the projection layer and LLM in LLaVA are updated with generated instruction data. In our experiment, "liuhaotian/LLaVA-7b-delta-v0" and "liuhaotian/llava-llama-2-7b-chat-lightning-lora-preview" are used for evaluation.

**mPLUG-Owl** mPLUG-Owl (Ye et al., 2023) proposes a novel training paradigm with a two-stage fashion. During pre-training, mPLUG-Owl incorporates a trainable visual encoder and a visual abstractor, while maintaining the LLM frozen. In the stage of instruction tuning, language-only and multi-modal instruction data are used to fine-tune a LoRA module on the LLM. "MAGAer13/mplug-owl-llama-7b" is used in our experiment, but LoRA is not implemented in this version.

**ImageBind-LLM (IB-LLM)** ImageBind-LLM (Han et al., 2023) adopts a two-stage training pipeline. In the pre-training stage, a learnable bind network and a gating factor are updated. The bind network transforms image features, while the gating factor weights the transformed image features to determine the magnitude of injecting visual semantics and the result is added to each word token for each layer. In the instruction tuning stage, a mixture of language instruction data and visual

| Model | Pre-training Data | | Instruction Data | |
|---|---|---|---|---|
| | Dataset | Size | Dataset | Size |
| BLIP-2 | MSCOCO+VG+CC3M+ CC12M+SBU+L400M | 129M | - | - |
| InstructBLIP | Following BLIP-2 | 129M | COCO Caption+ WebCapFilt+ TextCaps+VQAv2+ OKVQA+A-OKVQA+ OCR-VQA+LLaVA | 16M |
| LLaVA | CC3M | 595K | LLaVA | 158K |
| MiniGPT4 | CC3M+SBU+L400M | 5M | CC3M formatted by ChatGPT | 3.5K |
| mPLUG-Owl | L400M+COYO+ CC3M+MSCOCO | 104B tokens | Alpaca+Vicuna+ Baize+LLaVA | 102K+90K+ 50K+158K |
| PandaGPT | - | - | MiniGPT4+LLaVA | 160K |
| IB-LLM | CC3M+SBU+L2B+ COYO+MMC4 | 940M | Alpaca+GPT4LLM+ ShareGPT+ MiniGPT4+LLaVA | NA |
| LA-V2 | COCO Caption | 567K | GPT4LLM | 52K |
| mmGPT | - | - | Dolly15K+GPT4LLM+ LLaVA+MiniGPT4+ A-OKVQA+ COCO Caption+ OCR VQA formatted by ChatGPT | 15K+52K+ 158K+3.5K 5K+0.5K+0.5K |
| Shikra | LLaVA-Pretraining+ Flickr30K+ VG+RefCOCO+ VQAv2+PointQA+ Visual7W+VCR | $\sim$ 4.3M | LLaVA+Shikra-RD | 158K+5.9K |
| Lynx | BlipCapFilt+ CC12M+CC3M+SBU+ more 26 datasets (listed in Table 9 in Lynx) | 14B tokens | text-only+image-text+ video-text data (all 32 datasets are listed in Table 9 in Lynx) | 3B tokens |
| Cheetor | pre-training image-text data | 5M | CAGIT+image-text data | 64K+700K |
| BLIVA | L400M+CC3M+ SBU | 558K | Following InstructBLIP | $\sim$ 2.4M |

Table 8: Pre-training data and instruction tuning data used by different LVLMs. In Lynx, the training data size is represented by "tokens" and these tokens are considered as consumed samples rather than data volume, which is also the case for the pre-training data in mPLUG-Owl. "L400M" and "L2B" refer to LAION-400M and LAION-2B, repectively. GPT4LLM is a GPT4 version of Alpaca with higher quality. "$\sim$" denotes that the value is approximated by multiplying the batch size with the training steps of one epoch. "NA" denotes that the specific size is not mentioned in the corresponding paper.

instruction data is used to update partial parameters in LLaMA by LoRA and bias-norm tuning. We utilize "Cxxs/ImageBind-LLM/7B" for evaluation and call it "IB-LLM".

**LLaMA-Adapter V2** In LLaMA-Adapter V2 (Gao et al., 2023), a joint training paradigm is proposed, where only the visual projection layers and early zero-initialized attention with gating are pre-trained using image-text data, while the late adaptation prompts with zero gating, the unfrozen norm, newly added bias, and scale factors are implemented for learning from the language-only instruction data. "LLaMA-Adapter-V2-BIAS-7B" is applied for evaluation.

**Multimodal-GPT (mmGPT)**  Multimodal-GPT (Gong et al., 2023) is fine-tuned from Open-Flamingo, where the whole open-flamingo model is frozen and the LoRA module is added and updated to the self-attention, cross-attention, and FFN part in the LLM, using language-only and multimodal instruction data. "mmgpt-lora-v0-release" is used for evaluation. To simplify, we refer to it as "mmGPT".

**PandaGPT**  PandaGPT (Su et al., 2023) utilizes a one-stage training method using a combination of 160k image-language instruction-following data from MiniGPT-4 and LLaVA, where only two components are trained: a linear projection matrix connecting the visual representation generated by ImageBind to Vicuna, and additional LoRA weights applied to Vicuna attention modules. "openllmplayground/pandagpt-7b-max-len-1024" is evaluated as our implemented version.

**Shikra**  Shikra (Chen et al., 2023) consists of a vision encoder, an alignment layer, and an LLM. This model is trained in two stages, where both the fully connected layer and the entire LLM are trained and the visual encoder is frozen. We select "shikras/shikra-7b-delta-v1" for our evaluation in this experiment.

**Cheetor**  Cheetor (Li et al., 2023a) is initialized from BLIP-2 and pre-trains Q-Former that matches Vicuna and LLaMA2. A lightweight CLORI module is introduced that leverages the sophisticated reasoning ability of LLMs to control the Q-Former to conditionally extract specific visual features, and further re-inject them into the LLM. During training, only a set of query embeddings and two linear projection layers need to be updated. "cheetah-llama2-7b" and "cheetah-vicuna-7b" are specifically used for assessment.

**Lynx**  Lynx (Zeng et al., 2023) leverages the method of pre-training and instruction tuning, where lightweight trainable adapters are added after some layers in the frozen LLMs and a resampler mechanism is adapted to reduce the vision token sequence, using learnable latent queries. During pre-training, the newly added layers are trained to build connections of different modalities. "finetune-lynx" is opted for evaluation.

**BLIVA**  BLIVA (Hu et al., 2023) is initialized from a pre-trained InstructBLIP and merges learned query embeddings output by the Q-Former with projected encoded patch embeddings. Demonstrating a two-stage training paradigm, the patch embeddings projection layer is pre-trained and both the Q-Former and the project layer are fine-tuned by instruction tuning data. "mlpc-lab/BLIVA-Vicuna" is employed under evaluation in our experiment.

## C  COMPLEMENTARY RESULTS AND ANALYSIS

### C.1  PER-DIMENSION RESULTS AND ANALYSIS

In this section, we will provide the complete results and corresponding analysis of all capability dimensions.

#### C.1.1  RESULTS ON COARSE-GRAINED PERCEPTION

Tabel 9 and Table 10 provide results of coarse-grained perception tasks, including image classification and scene recognition. For generation evaluation, BLIP-2 and InstructBLIP take the lead in most tasks except the image quality assessment in VizWiz. We speculate that this is because the training data primarily consists of text describing the visual content, with very little description of the image quality, which may result in models not understanding the image quality. However, this challenge is also faced by most LVLMs, some of them act worse than random guesses.

For likelihood evaluation, the situation is slightly different. Most models perform better across different datasets while BLIP-2 and InstructBLIP struggle in Flowers102, ImageNet-1K, Pets37, and MEDIC. We hypothesize this is due to the fact that the category names of these datasets are not common, the calculated likelihood may not be as effective as those for common tokens. As mentioned in Section C.4, the diversity of the pre-training data used by BLIP-2 and InstructBLIP is limited, while mPLUG-Owl utilizes diverse pre-training data. This explains why mPLUG-Owl

| Model | Image Classification | | | | | | | | | | | |
| --- | --- | --- | --- | --- | --- | --- | --- | --- | --- | --- | --- | --- |
| | Flowers102 | | CIFAR10 | | ImageNet-1K | | Pets37 | | VizWiz$_4$ | | VizWiz$_2$ | |
| | Acc | Instby | Acc | Instby | Acc | Instby | Acc | Instby | Acc | Instby | Acc | Instby |
| Generation Evaluation | | | | | | | | | | | | |
| BLIP-2$_F$ | 75.57 | 0.07 | **87.32** | 0.03 | 73.07 | 0.11 | 75.19 | 0.06 | 24.44 | 0.25 | 46.68 | 0.00 |
| InstructBLIP$_F$ | **77.75** | 0.04 | 83.72 | 0.02 | **77.06** | 0.07 | **83.28** | 0.02 | 27.59 | 0.15 | 47.43 | 0.01 |
| InstructBLIP$_V$ | 72.81 | 0.06 | 86.28 | 0.03 | 71.78 | 0.06 | 80.77 | 0.06 | **48.00** | 0.05 | 49.37 | 0.41 |
| LLaVA$_V$ | 35.92 | 0.45 | 14.38 | 0.69 | 19.48 | 0.60 | 25.90 | 0.20 | 14.91 | 0.93 | 45.20 | 0.43 |
| LLaVA$_{L_2}$ | 50.07 | 0.23 | 41.42 | 0.21 | 50.74 | 0.15 | 41.42 | 0.21 | 27.31 | 0.28 | 46.44 | 0.02 |
| MiniGPT4 | 43.89 | 0.20 | 43.06 | 0.44 | 48.85 | 0.31 | 43.06 | 0.44 | 31.30 | 0.39 | 45.89 | 0.27 |
| mPLUG-Owl | 37.56 | 0.52 | 37.30 | 0.29 | 37.54 | 0.30 | 42.19 | 0.41 | 31.85 | 0.75 | 46.09 | 0.49 |
| PandaGPT | 34.43 | 0.61 | 26.32 | 0.06 | 27.63 | 0.05 | 29.07 | 0.02 | 31.11 | 0.31 | 29.75 | 0.84 |
| IB-LLM | 26.99 | 0.13 | 23.42 | 0.00 | 22.45 | 0.12 | 24.59 | 0.48 | 26.02 | 0.29 | 49.90 | 0.24 |
| LA-V2 | 25.89 | 0.44 | 25.96 | 0.42 | 18.08 | 0.64 | 29.07 | 0.02 | 33.05 | 0.29 | 54.21 | 0.03 |
| mmGPT | 26.21 | 0.64 | 25.92 | 0.27 | 25.60 | 0.25 | 27.48 | 0.25 | 28.24 | 0.69 | 50.59 | 0.32 |
| Shikra | 41.54 | 0.11 | 50.72 | 0.11 | 47.99 | 0.10 | 42.62 | 0.11 | 21.48 | 0.21 | 47.72 | 0.13 |
| Lynx | 56.19 | 0.30 | 77.20 | 0.11 | 57.82 | 0.26 | 60.98 | 0.32 | 26.30 | 0.19 | **54.60** | 0.01 |
| Cheetor$_V$ | 55.26 | 0.27 | 59.12 | 0.11 | 46.51 | 0.25 | 44.86 | 0.28 | 33.98 | 0.23 | 49.90 | 0.16 |
| Cheetor$_{L_2}$ | 37.80 | 0.23 | 70.82 | 0.15 | 43.64 | 0.15 | 36.39 | 0.21 | 31.11 | 0.24 | 46.88 | 0.06 |
| BLIVA | 30.71 | 0.22 | 37.52 | 0.21 | 36.68 | 0.20 | 35.57 | 0.25 | 32.78 | 0.19 | 48.71 | 0.20 |
| Likelihood Evaluation | | | | | | | | | | | | |
| BLIP-2$_F$ | 56.31 | 0.05 | 89.40 | 0.03 | 51.40 | 0.07 | 54.10 | 0.07 | 12.78 | 0.01 | 48.12 | 0.04 |
| InstructBLIP$_F$ | 55.23 | 0.04 | 81.62 | 0.02 | 53.32 | 0.06 | 56.34 | 0.05 | 12.96 | 0.03 | 49.45 | 0.05 |
| InstructBLIP$_V$ | 50.44 | 0.04 | 88.40 | 0.03 | 44.04 | 0.06 | 51.20 | 0.06 | 14.91 | 0.02 | 51.09 | 0.08 |
| LLaVA$_V$ | 48.78 | 0.04 | 92.56 | 0.01 | 51.16 | 0.05 | 47.98 | 0.05 | 14.35 | 0.00 | 54.21 | 0.06 |
| LLaVA$_{L_2}$ | 48.51 | 0.04 | 48.52 | 0.06 | 42.12 | 0.06 | 48.52 | 0.06 | 13.33 | 0.02 | 48.91 | 0.02 |
| MiniGPT4 | 52.27 | 0.03 | 55.41 | 0.04 | 52.03 | 0.05 | 55.41 | 0.04 | **36.02** | 0.03 | 46.88 | 0.01 |
| mPLUG-Owl | 59.98 | 0.02 | 88.66 | 0.02 | 51.86 | 0.03 | 75.08 | 0.02 | 20.09 | 0.05 | 46.53 | 0.00 |
| PandaGPT | 48.78 | 0.06 | 76.86 | 0.06 | 43.89 | 0.08 | 24.21 | 0.11 | 27.03 | 0.11 | 48.02 | 0.00 |
| IB-LLM | 48.66 | 0.03 | 83.8 | 0.02 | 43.18 | 0.05 | 48.31 | 0.04 | 14.63 | 0.03 | 46.53 | 0.00 |
| LA-V2 | 30.83 | 0.09 | 62.14 | 0.08 | 24.49 | 0.00 | 47.27 | 0.08 | 12.96 | 0.04 | 46.53 | 0.00 |
| mmGPT | 41.78 | 0.04 | 93.34 | 0.04 | 45.02 | 0.08 | 41.53 | 0.06 | 13.70 | 0.05 | 46.53 | 0.00 |
| Shikra | 50.86 | 0.01 | 89.70 | 0.02 | 47.99 | 0.10 | 53.77 | 0.04 | 28.70 | 0.03 | **57.48** | 0.04 |
| Lynx | **67.73** | 0.03 | 87.86 | 0.03 | 56.47 | 0.07 | **75.85** | 0.04 | 20.09 | 0.07 | 47.23 | 0.01 |
| Cheetor$_V$ | 49.36 | 0.05 | 87.30 | 0.03 | 47.88 | 0.09 | 43.06 | 0.11 | 22.22 | 0.10 | 50.30 | 0.05 |
| Cheetor$_{L_2}$ | 45.35 | 0.07 | **96.88** | 0.02 | 38.13 | 0.09 | 39.07 | 0.10 | 18.89 | 0.06 | 46.58 | 0.02 |
| BLIVA | 59.36 | 0.04 | 94.78 | 0.01 | **58.27** | 0.04 | 67.10 | 0.04 | 19.35 | 0.03 | 48.02 | 0.03 |

Table 9: Evaluation results on coarse-grained perception. "Acc" and "Instby" are short for accuracy and instability, respectively.

performs well on these tasks while BLIP-2 and InstructBLIP perform relatively poorly. However, BLIP-2 and InstructBLIP are able to distinguish these categories if provided in the question, enabling them to perform well under generation evaluation.

### C.1.2 RESULTS ON FINE-GRAINED PERCEPTION

Tabel 11 and Table 12 provide results of fine-grained perception tasks, including Object Perception and Object Grounding. For object perception, BLIP-2 and InstructBLIP dominate most tasks, especially when evaluated with the generation evaluation strategy. Under likelihood evaluation, the effectiveness of BLIVA is demonstrated, indicating that incorporating patch features helps the model to perceive fine-grained information. Additionally, Lynx is a strong competitor in object-perception tasks benefiting from its enhanced visual representations. Considering the results of MSCOCO$_{goi}$, most models are able to solve a part of the questions, indicating that LVLMs are able to understand the bounding boxes in images and the grounded questions. Bounding box labeling can be an optional method to provide locality information without the need for understanding continuous input. As for object grounding, the task is quite difficult for most models. Only BLIP-2 and InstructBLIP perform well under generation evaluation, but they still struggle under likelihood evaluation. We speculate that this is because there are only subtle differences between options in these questions, such as "the person on the left" and "the person on the right". In generation evaluation, all options are provided in the context, helping the models with strong instruct understanding abilities to distinguish between them. As for likelihood evaluation, options are provided to the model separately, models may not be able to distinguish them effectively.

| Model | Scene Recognition | | | | | | Avg. | |
|---|---|---|---|---|---|---|---|---|
| | $\text{TDIUC}_{\text{sport}}$ | | $\text{TDIUC}_{\text{scene}}$ | | $\text{MEDIC}_{\text{dts}}$ | | | |
| | Acc | Instability | Acc | Instability | Acc | Instability | Acc | Instability |
| **Generation Evaluation** | | | | | | | | |
| BLIP-2$_F$ | 93.75 | 0.12 | 88.66 | 0.04 | 60.29 | 0.25 | 69.44 | 0.10 |
| InstructBLIP$_F$ | **93.79** | 0.12 | **89.27** | 0.05 | **60.76** | 0.15 | **71.18** | 0.07 |
| InstructBLIP$_V$ | 90.62 | 0.20 | 69.78 | 0.09 | 52.00 | 0.13 | 69.06 | 0.12 |
| LLaVA$_V$ | 39.29 | 1.17 | 28.47 | 1.00 | 34.67 | 0.48 | 28.69 | 0.66 |
| LLaVA$_{L_2}$ | 74.13 | 0.51 | 67.29 | 0.16 | 36.00 | 0.28 | 48.31 | 0.23 |
| MiniGPT4 | 65.41 | 0.68 | 58.04 | 0.40 | 36.19 | 0.44 | 46.19 | 0.40 |
| mPLUG-Owl | 59.54 | 0.85 | 58.79 | 0.64 | 26.67 | 0.81 | 41.95 | 0.56 |
| PandaGPT | 22.39 | 1.35 | 38.04 | 0.86 | 14.95 | 0.66 | 28.19 | 0.53 |
| IB-LLM | 24.59 | 1.33 | 45.70 | 0.59 | 19.43 | 0.77 | 29.23 | 0.44 |
| LA-V2 | 42.94 | 1.11 | 48.69 | 0.54 | 20.76 | 0.79 | 33.18 | 0.48 |
| mmGPT | 28.81 | 1.29 | 45.23 | 0.64 | 15.25 | 0.87 | 30.37 | 0.58 |
| Shikra | 78.26 | 0.43 | 60.56 | 0.53 | 34.00 | 0.27 | 47.21 | 0.22 |
| Lynx | 85.69 | 0.29 | 69.07 | 0.15 | 47.52 | 0.26 | 59.49 | 0.21 |
| Cheetor$_V$ | 76.33 | 0.47 | 59.63 | 0.35 | 42.38 | 0.49 | 52.00 | 0.29 |
| Cheetor$_{L_2}$ | 53.58 | 0.92 | 68.6 | 0.15 | 29.71 | 0.29 | 46.50 | 0.27 |
| BLIVA | 65.87 | 0.68 | 56.73 | 0.41 | 30.95 | 0.41 | 41.72 | 0.31 |
| **Likelihood Evaluation** | | | | | | | | |
| BLIP-2$_F$ | 96.32 | 0.05 | 89.90 | 0.02 | 48.00 | 0.05 | 60.70 | 0.04 |
| InstructBLIP$_F$ | 96.9 | 0.05 | 91.21 | 0.01 | 46.10 | 0.09 | 60.35 | 0.04 |
| InstructBLIP$_V$ | **97.2** | 0.04 | 90.98 | 0.01 | 38.57 | 0.52 | 58.54 | 0.10 |
| LLaVA$_V$ | 92.67 | 0.11 | 89.53 | 0.08 | **57.62** | 0.20 | 60.98 | 0.07 |
| LLaVA$_{L_2}$ | 76.33 | 0.32 | 80.47 | 0.19 | 42.76 | 0.40 | 49.94 | 0.13 |
| MiniGPT4 | 87.52 | 0.19 | 68.88 | 0.12 | 39.62 | 0.37 | 54.89 | 0.10 |
| mPLUG-Owl | 80.91 | 0.28 | 64.02 | 0.27 | 34.19 | 0.28 | 57.92 | 0.11 |
| PandaGPT | 41.28 | 0.88 | 55.98 | 0.35 | 14.76 | 0.67 | 42.31 | 0.26 |
| IB-LLM | 78.53 | 0.35 | 55.61 | 0.33 | 26.95 | 0.25 | 49.58 | 0.12 |
| LA-V2 | 71.28 | 0.42 | 63.93 | 0.22 | 24.57 | 0.49 | 42.67 | 0.16 |
| mmGPT | 82.11 | 0.27 | 71.87 | 0.07 | 37.24 | 0.36 | 52.57 | 0.11 |
| Shikra | 92.11 | 0.12 | 85.14 | 0.06 | 42.19 | 0.39 | 60.88 | 0.09 |
| Lynx | 96.24 | 0.07 | **96.63** | 0.03 | 46.86 | 0.31 | **66.11** | 0.07 |
| Cheetor$_V$ | 88.81 | 0.18 | 77.29 | 0.08 | 38.48 | 0.36 | 56.08 | 0.12 |
| Cheetor$_{L_2}$ | 78.90 | 0.31 | 72.24 | 0.06 | 38.10 | 0.29 | 52.68 | 0.11 |
| BLIVA | 96.33 | 0.05 | 93.83 | 0.10 | 47.14 | 0.19 | 64.91 | 0.06 |

Table 10: Supplement of Table 9

### C.1.3 RESULTS ON SCENE TEXT PERCEPTION

Table 13 and Table 14 provide results of scene text perception, which consists of OCR, Grounded OCR, KIE and OCR-based VQA tasks. Since the scene text perception task requires the model output to contain the target tokens in the image, only generation evaluation is conducted. InstructBLIP consistently dominates the OCR task, while mPLUG-Owl and BLIVA take the lead on GroundOCR, KIE and OCR-based VQA tasks. mPLUG-Owl also performs well on OCR task. OCR-related tasks depend heavily on the image comprehension capability of visual backbones, hence we attribute the stable performance of mPLUG-Owl to its special pre-training procedure, as it is the only LVLM that trains the visual backbone during pre-training. Similarly, BLIVA utilizes additional encoded patch embeddings to improve the understanding of text within images. Therefore, enhancing the understanding and utilization of visual content is key to promoting improvements in scene text perception.

### C.1.4 RESULTS ON VISUALLY GROUNDED REASONING

Table 15 and Table 16 provide results of visually grounded reasoning, which consists of VQA and KVQA. For generation evaluation, BLIP-2$_F$ and InstructBLIP$_F$ achieve top-2 performance on nearly all the datasets for VGR tasks. While in likelihood evaluation, InstructBLIP$_V$, Lynx and BLIVA

| Model | Object Perception | | | | | | | | | |
|---|---|---|---|---|---|---|---|---|---|---|
| | TDIUC$_{color}$ | | TDIUC$_{utility}$ | | TDIUC$_{position}$ | | TDIUC$_{detection}$ | | TDIUC$_{counting}$ | |
| | Acc | Instability | Acc | Instability | Acc | Instability | Acc | Instability | Acc | Instability |
| **Generation Evaluation** | | | | | | | | | | |
| BLIP-2$_F$ | 73.90 | 0.45 | 92.40 | 0.09 | **91.55** | 0.19 | **99.38** | 0.02 | 71.00 | 0.54 |
| InstructBLIP$_F$ | **78.22** | 0.39 | **95.56** | 0.03 | 90.26 | 0.22 | 98.77 | 0.03 | **73.05** | 0.50 |
| InstructBLIP$_V$ | 69.19 | 0.58 | 89.01 | 0.09 | 81.29 | 0.42 | 97.26 | 0.07 | 61.76 | 0.77 |
| LLaVA$_V$ | 8.25 | 1.46 | 57.89 | 0.80 | 30.60 | 1.33 | 53.29 | 0.97 | 23.43 | 1.43 |
| LLaVA$_{L_2}$ | 57.02 | 0.83 | 85.85 | 0.20 | 68.70 | 0.69 | 91.78 | 0.20 | 43.14 | 1.10 |
| MiniGPT4 | 46.48 | 1.03 | 72.16 | 0.51 | 56.21 | 0.95 | 83.77 | 0.39 | 42.38 | 1.12 |
| mPLUG-Owl | 28.20 | 1.32 | 55.67 | 0.83 | 40.86 | 1.23 | 60.41 | 0.89 | 28.10 | 1.32 |
| PandaGPT | 30.18 | 1.27 | 57.19 | 0.79 | 27.59 | 1.38 | 61.99 | 0.85 | 26.52 | 1.33 |
| IB-LLM | 25.17 | 1.36 | 42.69 | 1.04 | 31.12 | 1.37 | 39.93 | 1.26 | 28.10 | 1.33 |
| LA-V2 | 26.48 | 1.33 | 40.58 | 1.08 | 29.66 | 1.40 | 43.90 | 1.19 | 25.10 | 1.35 |
| mmGPT | 27.10 | 1.32 | 38.71 | 1.11 | 26.29 | 1.43 | 38.63 | 1.27 | 25.33 | 1.35 |
| Shikra | 39.43 | 1.15 | 61.64 | 0.69 | 50.43 | 1.00 | 61.23 | 0.85 | 27.67 | 0.43 |
| Lynx | 68.93 | 0.61 | 84.91 | 0.19 | 64.31 | 0.76 | 91.85 | 0.2 | 33.95 | 1.22 |
| Cheetor$_V$ | 47.10 | 1.01 | 78.01 | 0.34 | 49.91 | 1.05 | 85.34 | 0.34 | 35.71 | 1.22 |
| Cheetor$_{L_2}$ | 54.57 | 0.84 | 81.52 | 0.29 | 57.76 | 0.90 | 84.72 | 0.37 | 31.29 | 1.26 |
| BLIVA | 44.28 | 1.06 | 61.75 | 0.68 | 45.09 | 1.14 | 63.49 | 0.82 | 41.95 | 1.13 |
| **Likelihood Evaluation** | | | | | | | | | | |
| BLIP-2$_F$ | 79.83 | 0.34 | 91.35 | 0.10 | 94.66 | 0.12 | 98.84 | 0.03 | 82.43 | 0.31 |
| InstructBLIP$_F$ | 90.46 | 0.16 | **93.80** | 0.07 | 94.91 | 0.11 | **99.45** | 0.01 | **82.95** | 0.29 |
| InstructBLIP$_V$ | **91.64** | 0.15 | 91.35 | 0.14 | **95.78** | 0.10 | 99.11 | 0.02 | 79.86 | 0.33 |
| LLaVA$_V$ | 70.44 | 0.48 | 83.27 | 0.12 | 77.59 | 0.46 | 97.05 | 0.06 | 62.23 | 0.59 |
| LLaVA$_{L_2}$ | 55.72 | 0.72 | 82.81 | 0.21 | 71.90 | 0.59 | 91.10 | 0.20 | 74.71 | 0.44 |
| MiniGPT4 | 69.92 | 0.49 | 86.90 | 0.13 | 76.12 | 0.49 | 96.37 | 0.09 | 73.19 | 0.46 |
| mPLUG-Owl | 57.75 | 0.68 | 90.41 | 0.10 | 75.69 | 0.52 | 93.22 | 0.15 | 71.00 | 0.49 |
| PandaGPT | 45.07 | 0.87 | 67.95 | 0.45 | 47.59 | 1.02 | 76.16 | 0.51 | 45.19 | 0.97 |
| IB-LLM | 36.14 | 0.92 | 82.22 | 0.25 | 63.53 | 0.73 | 81.16 | 0.39 | 23.95 | 1.09 |
| LA-V2 | 55.72 | 0.72 | 88.70 | 0.15 | 69.57 | 0.64 | 83.84 | 0.35 | 54.33 | 0.7 |
| mmGPT | 44.96 | 0.85 | 86.32 | 0.14 | 74.74 | 0.54 | 95.82 | 0.10 | 63.81 | 0.62 |
| Shikra | 58.07 | 0.67 | 86.67 | 0.16 | 73.10 | 0.55 | 88.22 | 0.25 | 74.24 | 0.43 |
| Lynx | 82.61 | 0.31 | 87.37 | 0.15 | 92.84 | 0.16 | 98.36 | 0.04 | 81.33 | 0.34 |
| Cheetor$_V$ | 72.74 | 0.46 | 82.69 | 0.16 | 80.60 | 0.42 | 98.08 | 0.05 | 70.90 | 0.49 |
| Cheetor$_{L_2}$ | 61.15 | 0.64 | 66.32 | 0.55 | 75.43 | 0.50 | 92.05 | 0.18 | 50.48 | 0.76 |
| BLIVA | 89.87 | 0.18 | 92.05 | 0.07 | 95.17 | 0.11 | 99.25 | 0.02 | 82.57 | 0.29 |

Table 11: Evaluation results on fine-grained perception.

also exhibit some outstanding performance. As discussed in Section 4.3.4, likelihood evaluation performances of many models are significantly better than generation.

We also conducted a hierarchical evaluation of LVLMs' external knowledge incorporation and reasoning abilities. Comparing the results of ViQuAE and K-ViQuAE, as well as A-OKVQA and A-OKVQRA, it is evident that, with the provision of external knowledge, the performance of most models has significantly improved.

In addition, it can be observed that, except for models based on FlanT5 and LLaMA2-Chat, the generation performance of other models on the A-OKVQAR dataset is significantly lower than the likelihood performance. This may be attributed to the fact that the rationale options are often longer, and for language models with weaker instruction-following capabilities (as mentioned in Section 4.3.4), memory lapses may occur during generation, thereby affecting performance.

### C.1.5 RESULTS ON SPATIAL UNDERSTANDING

Table 17 provides results of the spatial understanding capability dimension, which includes relation judgment and space-based perception tasks. For generation evaluation, BLIP-2 and InstructBLIP dominate in most tasks. For likelihood evaluation, although BLIP-2 and InstructBLIP still perform well, Lynx and BLIVA start to hold leadership positions. We attribute this to the fact that BLIP-2, InstructBLIP, Lynx and BLIVA uses relatively high-quality MSCOCO data. As the images in VSR dataset are retrieved from COCO 2017, models tend to have better performance on this dataset.

For the spatial relation judgment task, the performance on the MP3D-Spatial dataset is relatively poor. We believe the reason for this is that MP3D-Spatial is sampled from real navigation environments, which are inherently more complex. For space-based perception tasks, likelihood evalua-

| Model | Object Perception | | | | | | | | Object Grounding | | Avg. | |
|---|---|---|---|---|---|---|---|---|---|---|---|---|
| | MSCOCO$_{count}$ | | MSCOCO$_{mci}$ | | MSCOCO$_{goi}$ | | MSCOCO$_{mos}$ | | RefCOCO$_{res}$ | | | |
| | Acc | Instability | Acc | Instability | Acc | Instability | Acc | Instability | Acc | Instability | Acc | Instability |
| | | | | | Generation Evaluation | | | | | | | |
| BLIP-2$_F$ | **87.48** | 0.26 | 82.22 | 0.09 | 59.50 | 0.09 | 39.11 | 0.21 | 69.58 | 0.20 | 76.61 | 0.21 |
| InstructBLIP$_F$ | 87.25 | 0.26 | **84.72** | 0.08 | 63.05 | 0.11 | 37.65 | 0.28 | **72.75** | 0.12 | **78.13** | 0.20 |
| InstructBLIP$_V$ | 64.02 | 0.76 | 83.17 | 0.17 | **64.94** | 0.24 | 39.27 | 0.55 | 57.58 | 0.24 | 70.75 | 0.39 |
| LLaVA$_V$ | 18.60 | 1.48 | 44.88 | 0.91 | 29.56 | 1.14 | 34.66 | 0.94 | 42.41 | 0.38 | 34.36 | 1.08 |
| LLaVA$_{L_2}$ | 40.78 | 1.16 | 64.67 | 0.48 | 53.33 | 0.36 | 41.54 | 0.64 | 50.75 | 0.30 | 59.76 | 0.60 |
| MiniGPT4 | 35.95 | 1.24 | 58.50 | 0.66 | 53.06 | 0.61 | 41.94 | 0.68 | 41.00 | 0.29 | 53.15 | 0.75 |
| mPLUG-Owl | 28.07 | 1.33 | 31.00 | 1.02 | 30.94 | 1.04 | 36.60 | 0.91 | 32.08 | 0.78 | 37.19 | 1.07 |
| PandaGPT | 22.92 | 1.40 | 25.11 | 1.03 | 25.16 | 1.01 | 41.54 | 0.70 | 28.08 | 0.47 | 34.63 | 1.02 |
| IB-LLM | 28.54 | 1.32 | 32.06 | 0.90 | 30.67 | 0.85 | 40.24 | 0.89 | 28.75 | 0.62 | 32.73 | 1.09 |
| LA-V2 | 26.43 | 1.34 | 22.33 | 1.06 | 21.11 | 1.11 | 42.02 | 0.72 | 30.75 | 0.76 | 30.84 | 1.13 |
| mmGPT | 22.34 | 1.40 | 30.33 | 1.00 | 27.56 | 1.01 | 41.38 | 0.75 | 25.58 | 0.93 | 30.33 | 1.16 |
| Shikra | 29.04 | 1.31 | 70.50 | 0.44 | 54.61 | 0.57 | 28.91 | 0.57 | 51.75 | 0.22 | 47.52 | 0.72 |
| Lynx | 51.07 | 1.00 | 73.56 | 0.37 | 60.78 | 0.30 | **42.19** | 0.42 | 54.67 | 0.40 | 62.62 | 0.55 |
| Cheetor$_V$ | 29.63 | 1.29 | 49.67 | 0.78 | 45.11 | 0.72 | 38.78 | 0.70 | 43.75 | 0.52 | 50.30 | 0.80 |
| Cheetor$_{L_2}$ | 30.57 | 1.29 | 52.16 | 0.66 | 43.16 | 0.66 | 40.32 | 0.67 | 37.42 | 0.46 | 51.35 | 0.74 |
| BLIVA | 37.93 | 1.20 | 39.94 | 0.82 | 32.22 | 0.89 | 40.00 | 0.67 | 27.58 | 0.35 | 43.42 | 0.88 |
| | | | | | Likelihood Evaluation | | | | | | | |
| BLIP-2$_F$ | 77.31 | 0.41 | 81.77 | 0.02 | 60.39 | 0.02 | 39.27 | 0.02 | 38.58 | 0.15 | 74.44 | 0.15 |
| InstructBLIP$_F$ | 73.14 | 0.48 | 82.94 | 0.04 | 60.44 | 0.05 | 42.27 | 0.06 | 36.08 | 0.09 | 75.64 | 0.14 |
| InstructBLIP$_V$ | 89.98 | 0.17 | **84.72** | 0.02 | 64.94 | 0.02 | **43.89** | 0.04 | 37.17 | 0.11 | 77.84 | 0.11 |
| LLaVA$_V$ | 91.07 | 0.16 | 76.22 | 0.12 | 62.50 | 0.15 | 39.92 | 0.15 | 43.00 | 0.15 | 70.33 | 0.24 |
| LLaVA$_{L_2}$ | 89.01 | 0.20 | 64.44 | 0.17 | 47.44 | 0.12 | 40.24 | 0.13 | 38.33 | 0.09 | 65.57 | 0.29 |
| MiniGPT4 | 87.56 | 0.22 | 77.11 | 0.06 | 58.22 | 0.06 | 41.62 | 0.07 | 38.50 | 0.10 | 70.55 | 0.22 |
| mPLUG-Owl | 89.04 | 0.20 | 56.61 | 0.13 | 49.56 | 0.14 | 38.38 | 0.10 | 39.33 | 0.10 | 66.10 | 0.26 |
| PandaGPT | 66.04 | 0.60 | 32.22 | 0.29 | 26.94 | 0.31 | 42.41 | 0.28 | 24.50 | 0.13 | 47.38 | 0.54 |
| IB-LLM | 87.06 | 0.22 | 48.61 | 0.09 | 41.78 | 0.13 | 43.72 | 0.09 | 35.92 | 0.10 | 54.41 | 0.40 |
| LA-V2 | 87.45 | 0.22 | 49.83 | 0.08 | 46.83 | 0.16 | 41.21 | 0.09 | 36.67 | 0.11 | 61.42 | 0.32 |
| mmGPT | 69.12 | 0.56 | 61.56 | 0.11 | 52.19 | 0.37 | 43.08 | 0.10 | 32.33 | 0.08 | 62.39 | 0.35 |
| Shikra | 90.21 | 0.17 | 51.56 | 0.10 | 54.67 | 0.10 | 41.70 | 0.08 | **49.92** | 0.11 | 66.84 | 0.26 |
| Lynx | 88.97 | 0.18 | 80.22 | 0.08 | 65.28 | 0.09 | 42.02 | 0.12 | 43.42 | 0.09 | 76.24 | 0.16 |
| Cheetor$_V$ | 77.66 | 0.40 | 74.67 | 0.08 | 55.67 | 0.12 | 43.07 | 0.14 | 33.42 | 0.16 | 68.95 | 0.25 |
| Cheetor$_{L_2}$ | 80.16 | 0.36 | 67.88 | 0.09 | 51.56 | 0.11 | 39.92 | 0.12 | 32.00 | 0.13 | 61.70 | 0.34 |
| BLIVA | **94.04** | 0.10 | 84.00 | 0.03 | **66.72** | 0.04 | 42.35 | 0.06 | 35.75 | 0.13 | **78.18** | 0.10 |

Table 12: Supplement of Table 11

| Model | OCR | | | | | | Grounded OCR | | |
|---|---|---|---|---|---|---|---|---|---|
| | CUTE80 | IC15 | IIIT5K | COCO-Text | WordArt | TextOCR | gIC15 | gCOCO-Text | gTextOCR |
| | | | | Generation Evaluation | | | | | |
| BLIP-2$_F$ | 80.07 | 64.75 | 72.27 | 50.54 | 72.32 | 68.40 | 61.43 | 16.27 | 32.60 |
| InstructBLIP$_F$ | **84.17** | **75.36** | **83.40** | **56.66** | **75.76** | **74.27** | 55.24 | 17.33 | 32.60 |
| InstructBLIP$_V$ | 81.60 | 73.15 | 77.20 | 53.00 | 74.17 | 70.60 | 53.33 | 18.93 | 32.13 |
| LLaVA$_V$ | 26.11 | 23.65 | 25.73 | 13.00 | 34.44 | 36.00 | 40.24 | 17.33 | 30.00 |
| LLaVA$_{L_2}$ | 32.15 | 25.75 | 27.20 | 15.83 | 40.13 | 39.87 | 43.57 | 17.93 | 32.07 |
| MiniGPT4 | 71.39 | 58.90 | 71.67 | 40.36 | 72.45 | 60.27 | 44.05 | 11.00 | 28.60 |
| mPLUG-Owl | 73.68 | 60.77 | 74.67 | 46.39 | 73.25 | 64.27 | 64.52 | 20.27 | 34.67 |
| PandaGPT | 1.60 | 1.88 | 3.13 | 0.14 | 5.03 | 26.87 | 0.24 | 0.40 | 22.73 |
| IB-LLM | 11.94 | 8.95 | 8.60 | 2.41 | 11.66 | 29.73 | 6.19 | 3.33 | 23.80 |
| LA-V2 | 36.53 | 31.82 | 35.33 | 17.82 | 40.13 | 42.13 | 47.86 | 18.07 | 33.20 |
| mmGPT | 26.94 | 23.09 | 18.80 | 13.43 | 31.13 | 36.20 | 26.43 | 12.73 | 28.80 |
| Shikra | 2.57 | 4.75 | 5.07 | 4.33 | 9.54 | 31.13 | 29.76 | 5.93 | 27.53 |
| Lynx | 34.03 | 25.08 | 20.93 | 11.51 | 37.22 | 36.00 | 32.86 | 10.60 | 30.80 |
| Cheetor$_V$ | 52.50 | 39.01 | 53.87 | 29.73 | 56.16 | 52.27 | 40.00 | 11.20 | 28.20 |
| Cheetor$_{L_2}$ | 42.78 | 31.38 | 39.20 | 20.36 | 34.83 | 45.40 | 16.67 | 6.67 | 25.13 |
| BLIVA | 77.29 | 68.40 | 72.47 | 51.49 | 71.26 | 66.93 | **64.76** | **21.67** | **37.27** |

Table 13: Evaluation results on scene text perception.

tion yields better results than generation evaluation, especially for LLaVA, mPLUG-Owl, LA-V2, mmGPT, Shikra and Cheetor. This might be attributed to the high demand for spatial reasoning skills for this task, thereby placing a greater emphasis on the image comprehension abilities of visual backbones. Most of these models use ViT-L, which lacks robust spatial semantic understanding.

### C.1.6 RESULTS ON CROSS-MODAL INFERENCE

Table 18 provides results of the cross-modal inference capability dimension, which includes image-text matching tasks and visual entailment tasks. For the image-text matching task in MSCOCO, we consider a one-to-one setup of the naive image-text matching and a one-to-four selection setup of image-text selection. BLIP-2 and Instruct BLIP perform well in both setups under the generation

| Model | KIE | | | OCR-based VQA | | | Avg. |
|---|---|---|---|---|---|---|---|
| | SROIE | POIE | FUNSD | TextVQA | DocVQA | OCR-VQA | |
| **Generation Evaluation** | | | | | | | |
| BLIP-2$_F$ | 1.30 | 0.76 | 1.72 | 21.47 | 5.39 | 21.62 | 38.06 |
| InstructBLIP$_F$ | 0.87 | 0.44 | 2.07 | 26.76 | 4.78 | 28.07 | **41.19** |
| InstructBLIP$_V$ | 3.48 | 0.82 | 1.72 | 30.22 | 6.21 | 34.37 | 40.73 |
| LLaVA$_V$ | 0.00 | 0.44 | 1.72 | 19.02 | 3.13 | 5.74 | 18.44 |
| LLaVA$_{L_2}$ | 0.00 | 1.21 | 1.72 | 26.31 | 6.52 | 12.13 | 21.49 |
| MiniGPT4 | 0.29 | 0.85 | 2.07 | 17.29 | 3.95 | 12.49 | 33.04 |
| mPLUG-Owl | **4.06** | 1.58 | 1.72 | **30.71** | **8.40** | 37.87 | 39.79 |
| PandaGPT | 0.00 | 0.09 | 1.72 | 0.80 | 2.22 | 0.00 | 4.46 |
| IB-LLM | 0.00 | 0.06 | 1.72 | 10.09 | 3.62 | 0.91 | 8.20 |
| LA-V2 | 0.72 | **3.16** | 1.72 | 30.40 | 8.06 | 16.40 | 24.22 |
| mmGPT | 0.00 | 1.33 | 1.72 | 21.07 | 4.78 | 4.47 | 16.73 |
| Shikra | 0.00 | 0.82 | 1.72 | 1.56 | 0.19 | 0.25 | 8.34 |
| Lynx | 0.00 | 1.58 | 1.72 | 24.71 | 4.11 | 7.61 | 18.58 |
| Cheetor$_V$ | 0.14 | 0.79 | 1.72 | 13.16 | 3.62 | 7.26 | 25.90 |
| Cheetor$_{L_2}$ | 0.00 | 0.57 | 1.72 | 11.02 | 4.11 | 3.05 | 18.83 |
| BLIVA | 2.61 | 3.04 | **3.45** | 29.69 | 6.18 | 34.97 | 40.77 |

Table 14: Supplement of Table 13.

| Model | VQA | | | | | | KVQA | | | | | |
|---|---|---|---|---|---|---|---|---|---|---|---|---|
| | GQA | | VQA v2 | | Whoops | | OK-VQA | | ScienceQA | | VizWiz | |
| | Acc | Instability | Acc | Instability | Acc | Instability | Acc | Instability | Acc | Instability | Acc | Instability |
| **Generation Evaluation** | | | | | | | | | | | | |
| BLIP-2$_F$ | 62.66 | 0.18 | 69.86 | 0.19 | **81.96** | 0.10 | 68.97 | 0.23 | **63.78** | 0.27 | **82.13** | 0.12 |
| InstructBLIP$_F$ | **66.11** | 0.19 | **74.31** | 0.16 | 80.04 | 0.08 | **70.87** | 0.15 | 62.19 | 0.26 | 74.90 | 0.18 |
| InstructBLIP$_V$ | 59.09 | 0.31 | 62.59 | 0.32 | 71.96 | 0.27 | 63.73 | 0.36 | 58.01 | 0.42 | 49.37 | 0.41 |
| LLaVA$_V$ | 37.26 | 0.80 | 46.01 | 0.55 | 50.48 | 0.69 | 41.59 | 0.78 | 46.57 | 0.63 | 31.37 | 0.80 |
| LLaVA$_{L_2}$ | 52.36 | 0.43 | 51.65 | 0.40 | 57.14 | 0.43 | 54.09 | 0.57 | 57.91 | 0.49 | 40.32 | 0.54 |
| MiniGPT4 | 44.57 | 0.66 | 46.49 | 0.64 | 47.08 | 0.78 | 38.65 | 0.88 | 43.78 | 0.71 | 35.22 | 0.83 |
| mPLUG-Owl | 34.56 | 0.89 | 35.48 | 0.88 | 37.44 | 0.93 | 33.45 | 0.97 | 41.39 | 0.80 | 30.63 | 0.96 |
| PandaGPT | 38.07 | 0.67 | 37.28 | 0.68 | 24.64 | 0.85 | 29.80 | 0.90 | 44.48 | 0.69 | 24.87 | 0.89 |
| IB-LLM | 38.63 | 0.75 | 38.56 | 0.77 | 27.86 | 0.95 | 31.67 | 0.96 | 41.49 | 0.73 | 26.22 | 0.97 |
| LA-V2 | 40.21 | 0.68 | 39.27 | 0.67 | 34.17 | 0.94 | 29.52 | 1.00 | 41.59 | 0.76 | 28.63 | 0.93 |
| mmGPT | 35.12 | 0.85 | 34.47 | 0.85 | 27.20 | 1.01 | 27.34 | 1.00 | 40.10 | 0.78 | 23.67 | 0.95 |
| Shikra | 41.69 | 0.73 | 44.93 | 0.67 | 50.48 | 0.73 | 41.15 | 0.86 | 38.61 | 0.72 | 41.81 | 0.81 |
| Lynx | 55.88 | 0.47 | 60.14 | 0.43 | 67.02 | 0.41 | 55.75 | 0.55 | 53.63 | 0.49 | 49.65 | 0.61 |
| Cheetor$_V$ | 46.17 | 0.58 | 48.17 | 0.56 | 55.71 | 0.64 | 43.49 | 0.78 | 47.06 | 0.63 | 37.59 | 0.78 |
| Cheetor$_{L_2}$ | 48.39 | 0.42 | 45.62 | 0.43 | 42.26 | 0.51 | 44.64 | 0.61 | 56.12 | 0.50 | 32.76 | 0.65 |
| BLIVA | 43.40 | 0.58 | 50.06 | 0.01 | 46.31 | 0.76 | 36.75 | 0.76 | 42.09 | 0.65 | 35.64 | 0.80 |
| **Likelihood Evaluation** | | | | | | | | | | | | |
| BLIP-2$_F$ | 62.70 | 0.06 | 69.37 | 0.06 | 70.95 | 0.04 | 66.83 | 0.07 | 53.93 | 0.03 | 76.71 | 0.08 |
| InstructBLIP$_F$ | 66.65 | 0.06 | 79.67 | 0.05 | 68.99 | 0.04 | 76.35 | 0.03 | 56.32 | 0.03 | 62.92 | 0.04 |
| InstructBLIP$_V$ | **67.76** | 0.04 | **82.92** | 0.02 | 72.32 | 0.02 | **82.82** | 0.02 | 57.91 | 0.02 | 57.17 | 0.03 |
| LLaVA$_V$ | 54.14 | 0.12 | 58.94 | 0.09 | 65.36 | 0.06 | 62.18 | 0.11 | 54.03 | 0.08 | 40.28 | 0.07 |
| LLaVA$_{L_2}$ | 52.68 | 0.09 | 51.81 | 0.10 | 60.06 | 0.06 | 48.77 | 0.15 | 55.22 | 0.09 | 47.05 | 0.09 |
| MiniGPT4 | 56.10 | 0.06 | 56.04 | 0.06 | 63.15 | 0.05 | 55.08 | 0.10 | 51.74 | 0.04 | 49.00 | 0.05 |
| mPLUG-Owl | 50.95 | 0.06 | 50.53 | 0.07 | 60.89 | 0.05 | 49.80 | 0.10 | 50.25 | 0.04 | 44.32 | 0.09 |
| PandaGPT | 41.35 | 0.20 | 37.86 | 0.25 | 28.69 | 0.20 | 36.11 | 0.24 | 43.88 | 0.13 | 19.12 | 0.17 |
| IB-LLM | 46.86 | 0.03 | 47.06 | 0.03 | 48.93 | 0.04 | 45.52 | 0.05 | 54.03 | 0.03 | 32.71 | 0.05 |
| LA-V2 | 50.29 | 0.06 | 47.36 | 0.05 | 60.71 | 0.06 | 44.60 | 0.10 | 50.05 | 0.03 | 41.21 | 0.08 |
| mmGPT | 52.86 | 0.06 | 48.54 | 0.06 | 49.64 | 0.06 | 56.43 | 0.10 | 49.25 | 0.03 | 38.93 | 0.06 |
| Shikra | 57.07 | 0.09 | 64.38 | 0.07 | 65.83 | 0.06 | 59.72 | 0.08 | 51.54 | 0.06 | 42.00 | 0.04 |
| Lynx | 69.21 | 0.12 | 73.82 | 0.10 | **73.21** | 0.05 | 68.97 | 0.11 | 55.52 | 0.05 | **79.44** | 0.04 |
| Cheetor$_V$ | 58.71 | 0.09 | 59.70 | 0.09 | 62.86 | 0.08 | 58.81 | 0.11 | 48.26 | 0.08 | 44.73 | 0.13 |
| Cheetor$_{L_2}$ | 53.03 | 0.08 | 50.13 | 0.10 | 56.55 | 0.09 | 50.63 | 0.13 | 55.42 | 0.06 | 35.96 | 0.11 |
| BLIVA | 67.37 | 0.05 | 81.36 | 0.03 | 69.88 | 0.03 | 78.53 | 0.03 | **60.70** | 0.02 | 51.32 | 0.04 |

Table 15: Evaluation results on visually grounded reasoning.

evaluation. However, the performance is not satisfactory under the likelihood evaluation for image-text selection, we attribute this to the same reason that is mentioned in the analysis of referring expression selection in Appendix C.1.2. Unlike MSCOCO, WikiHow considers the scenarios to match images and abstract instructions, while Winogroud uses negative options with only minor word-level modifications. These pose significant challenges for the models, resulting in a noticeable decrease in accuracy. However, InstructBLIP maintains a lead. Regarding the visual entailment task, apart from the two models based on FlanT5, the performance of the other models is not promising. In summary, we believe that current LVLMs still have relatively weak capabilities in logical reasoning and understanding fine-grained textual details.

| Model | KVQA | | | | | | | | | | | | Avg. | |
| | ViQuAE | | K-ViQuAE | | A-OKVQA | | A-OKVQARA | | A-OKVQAR | | ImageNetVC | | | |
| | Acc | Instability | Acc | Instability | Acc | Instability | Acc | Instability | Acc | Instability | Acc | Instability | Acc | Instability |
|---|---|---|---|---|---|---|---|---|---|---|---|---|---|---|
| **Generation Evaluation** | | | | | | | | | | | | | | |
| BLIP-2$_F$ | 44.32 | 0.37 | **87.26** | 0.10 | 71.40 | 0.18 | 85.61 | 0.07 | 80.00 | 0.17 | **81.72** | 0.14 | 73.31 | 0.18 |
| InstructBLIP$_F$ | 43.68 | 0.36 | 87.10 | 0.09 | **77.02** | 0.17 | **89.82** | 0.05 | **81.23** | 0.16 | 79.07 | 0.14 | **73.86** | 0.17 |
| InstructBLIP$_V$ | **49.76** | 0.41 | 80.32 | 0.26 | 71.58 | 0.28 | 79.47 | 0.24 | 47.72 | 0.61 | 62.56 | 0.30 | 63.01 | 0.35 |
| LLaVA$_V$ | 39.52 | 0.77 | 50.00 | 0.73 | 42.56 | 0.78 | 62.11 | 0.62 | 31.58 | 0.83 | 48.94 | 0.49 | 44.00 | 0.71 |
| LLaVA$_{L_2}$ | 47.20 | 0.50 | 85.00 | 0.17 | 60.18 | 0.47 | 81.93 | 0.24 | 61.93 | 0.40 | 66.58 | 0.30 | 59.69 | 0.41 |
| MiniGPT4 | 32.48 | 0.91 | 65.81 | 0.56 | 39.65 | 0.85 | 59.30 | 0.67 | 40.18 | 0.82 | 53.81 | 0.57 | 45.59 | 0.74 |
| mPLUG-Owl | 31.04 | 0.99 | 51.29 | 0.81 | 35.26 | 1.01 | 41.58 | 0.88 | 35.61 | 0.90 | 42.65 | 0.79 | 37.53 | 0.90 |
| PandaGPT | 39.84 | 0.74 | 74.68 | 0.37 | 29.82 | 0.94 | 63.16 | 0.62 | 39.82 | 0.76 | 56.81 | 0.55 | 41.94 | 0.72 |
| IB-LLM | 29.28 | 0.97 | 46.13 | 0.86 | 29.65 | 0.96 | 54.56 | 0.76 | 32.28 | 1.01 | 44.32 | 0.70 | 36.72 | 0.87 |
| LA-V2 | 27.20 | 1.00 | 40.32 | 0.90 | 31.75 | 0.99 | 47.02 | 0.86 | 31.93 | 0.90 | 43.78 | 0.56 | 36.28 | 0.85 |
| mmGPT | 31.04 | 0.99 | 45.81 | 0.90 | 24.39 | 1.00 | 38.42 | 0.93 | 25.26 | 0.95 | 43.14 | 0.73 | 33.00 | 0.91 |
| Shikra | 29.92 | 0.95 | 38.71 | 0.86 | 41.93 | 0.83 | 40.53 | 0.86 | 37.19 | 0.78 | 47.13 | 0.65 | 41.17 | 0.79 |
| Lynx | 39.68 | 0.65 | 72.42 | 0.41 | 65.09 | 0.46 | 79.30 | 0.28 | 37.54 | 0.85 | 64.08 | 0.37 | 58.35 | 0.50 |
| Cheetor$_V$ | 40.48 | 0.80 | 70.00 | 0.48 | 41.05 | 0.76 | 63.33 | 0.59 | 48.42 | 0.68 | 56.81 | 0.51 | 49.86 | 0.65 |
| Cheetor$_{L_2}$ | 44.16 | 0.57 | 82.26 | 0.22 | 48.42 | 0.54 | 82.28 | 0.22 | 57.37 | 0.50 | 68.50 | 0.26 | 54.40 | 0.45 |
| BLIVA | 33.92 | 0.76 | 45.16 | 0.85 | 44.21 | 0.71 | 54.74 | 0.74 | 31.05 | 0.77 | 45.70 | 0.56 | 42.42 | 0.66 |
| **Likelihood Evaluation** | | | | | | | | | | | | | | |
| BLIP-2$_F$ | 38.72 | 0.10 | 87.26 | 0.01 | 64.74 | 0.08 | 81.58 | 0.03 | **84.04** | 0.01 | 80.34 | 0.07 | 69.76 | 0.05 |
| InstructBLIP$_F$ | 33.12 | 0.05 | **88.71** | 0.02 | 70.88 | 0.06 | 79.82 | 0.03 | 83.86 | 0.01 | 84.47 | 0.05 | 70.98 | 0.04 |
| InstructBLIP$_V$ | 46.88 | 0.05 | 82.26 | 0.01 | 78.25 | 0.01 | 86.67 | 0.04 | 81.40 | 0.03 | **85.70** | 0.02 | **73.51** | 0.03 |
| LLaVA$_V$ | 39.20 | 0.10 | 75.00 | 0.07 | 59.30 | 0.09 | 73.51 | 0.12 | 61.40 | 0.05 | 63.34 | 0.07 | 58.89 | 0.09 |
| LLaVA$_{L_2}$ | 35.36 | 0.17 | 80.48 | 0.06 | 48.25 | 0.14 | 64.74 | 0.09 | 68.95 | 0.01 | 67.17 | 0.10 | 56.71 | 0.10 |
| MiniGPT4 | 27.68 | 0.08 | 73.23 | 0.03 | 57.72 | 0.07 | 70.35 | 0.06 | 64.21 | 0.03 | 62.80 | 0.05 | 57.26 | 0.06 |
| mPLUG-Owl | 30.24 | 0.09 | 78.87 | 0.08 | 42.98 | 0.10 | 63.51 | 0.09 | 67.89 | 0.02 | 61.03 | 0.06 | 54.27 | 0.07 |
| PandaGPT | 24.64 | 0.26 | 77.26 | 0.07 | 31.75 | 0.25 | 59.30 | 0.14 | 61.93 | 0.05 | 57.44 | 0.17 | 43.28 | 0.18 |
| IB-LLM | 32.80 | 0.05 | 67.42 | 0.08 | 44.39 | 0.04 | 58.42 | 0.10 | 66.32 | 0.04 | 58.92 | 0.02 | 50.28 | 0.05 |
| LA-V2 | 39.20 | 0.05 | 70.00 | 0.08 | 43.51 | 0.07 | 70.88 | 0.07 | 66.49 | 0.03 | 64.28 | 0.05 | 54.05 | 0.06 |
| mmGPT | 33.44 | 0.10 | 82.58 | 0.07 | 49.47 | 0.09 | 69.47 | 0.10 | 77.54 | 0.01 | 66.19 | 0.05 | 56.20 | 0.07 |
| Shikra | 35.20 | 0.10 | 65.65 | 0.04 | 57.72 | 0.07 | 64.74 | 0.11 | 75.26 | 0.00 | 62.56 | 0.09 | 58.47 | 0.07 |
| Lynx | 41.92 | 0.15 | 80.32 | 0.05 | 64.56 | 0.10 | 77.02 | 0.16 | 78.42 | 0.01 | 76.86 | 0.12 | 69.94 | 0.09 |
| Cheetor$_V$ | 34.40 | 0.10 | 70.81 | 0.07 | 59.12 | 0.11 | 70.88 | 0.09 | 70.18 | 0.02 | 66.29 | 0.06 | 58.73 | 0.09 |
| Cheetor$_{L_2}$ | 37.12 | 0.13 | 82.26 | 0.06 | 53.33 | 0.10 | 73.51 | 0.08 | 74.91 | 0.02 | 66.78 | 0.11 | 57.47 | 0.09 |
| BLIVA | **51.84** | 0.06 | 83.87 | 0.02 | **80.53** | 0.03 | **87.72** | 0.01 | 79.12 | 0.02 | 82.95 | 0.04 | 72.93 | 0.03 |

Table 16: Supplement of Table 15.

### C.1.7 RESULTS ON VISUAL DESCRIPTION

Table 19 and Table 20 provides image captioning results of the visual description capability dimension. We choose CIDEr metric to estimate visual description capability while providing BLEU-4, METEOR and ROUGE-L results for additional references. As mentioned in previous work (Xu et al., 2023), these datasets require concise captions while most LVLMs tend to generate detailed descriptions. Therefore, the performance of most models is not satisfying enough. For this task, PandaGPT always generates a sentence starting with "the image features" and the performance is limited. At the same time, BLIP-2 dominates the task because BLIP-2 is able to provide short captions. To adapt to the development of LVLMs, there is a strong need for a benchmark for evaluating detailed description capabilities.

### C.1.8 RESULTS ON MULTI-TURN DIALOGUE

Table 21 provides results of the multi-turn Dialogue task. BLIP-2 and InstructBLIP perform the best in this task while BLIVA takes second place under likelihood evaluation. For VQA-MT, there is no inter-turn dependency, and there is no obvious common relationship between the performance and the number of dialogue turns. However, for VisDial where strong dependencies exist between the proceeding and following questions, LVLMs generally perform worse in multi-turn dialogues where models face more complex contexts. Moreover, the performance of LVLMs deteriorates with the increase in dialogue turns. We hypothesize the reason is that single-turn image-text pairs dominate the training data of LVLMs. Multi-turn data should be incorporated during training to further improve existing LVLMs. Additionally, the advantages of some models like BLIVA and Shikra are only demonstrated with the likelihood evaluation method, this phenomenon has been discussed in Section 4.3.4

### C.2 EFFECT OF IN-CONTEXT SAMPLE

Table 22 provides the complete results for Section 4.3.3. The models that are not previously mentioned already possess strong abilities to follow single-choice instructions, either through the introduction of language-only instruction-following data during training (ImageBind-LLM) or through the inherent abilities of the frozen backbones like Vicuna (Chiang et al., 2023), LLaMA2-Chat (Touvron et al., 2023b), and OpenFlamingo. These models are able to generate option marks for most

| Model | Space-based Perception | | Spatial Relation Judgment | | | | Avg. | |
|---|---|---|---|---|---|---|---|---|
| | CLEVR | | VSR | | MP3D-Spatial | | | |
| | Acc | Instability | Acc | Instability | Acc | Instability | Acc | Instability |
| **Generation Evaluation** | | | | | | | | |
| BLIP-2$_F$ | 42.67 | 0.28 | 46.95 | 0.21 | 39.87 | 0.32 | 43.16 | 0.27 |
| InstructBLIP$_F$ | 44.84 | 0.39 | 52.37 | 0.25 | **41.01** | 0.37 | **46.07** | 0.34 |
| InstructBLIP$_V$ | **46.32** | 0.51 | 52.37 | 0.49 | 34.59 | 0.50 | 44.43 | 0.50 |
| LLaVA$_V$ | 19.01 | 1.24 | 40.00 | 0.88 | 27.19 | 1.13 | 28.73 | 1.08 |
| LLaVA$_{L_2}$ | 36.52 | 0.61 | **52.54** | 0.21 | 34.67 | 0.64 | 41.24 | 0.49 |
| MiniGPT4 | 33.74 | 0.84 | 36.44 | 0.81 | 33.62 | 0.84 | 34.60 | 0.83 |
| mPLUG-Owl | 27.48 | 1.01 | 28.81 | 0.97 | 24.23 | 1.04 | 26.84 | 1.01 |
| PandaGPT | 29.65 | 0.90 | 35.76 | 0.86 | 34.50 | 0.80 | 33.30 | 0.85 |
| IB-LLM | 31.45 | 0.96 | 40.00 | 0.94 | 35.22 | 0.83 | 35.56 | 0.91 |
| LA-V2 | 21.39 | 1.05 | 23.05 | 1.04 | 27.06 | 1.01 | 23.83 | 1.03 |
| mmGPT | 22.26 | 1.13 | 28.98 | 1.01 | 29.30 | 0.98 | 26.85 | 1.04 |
| Shikra | 23.82 | 0.77 | 46.27 | 0.60 | 29.77 | 0.84 | 33.29 | 0.74 |
| Lynx | 40.58 | 0.68 | 45.76 | 0.66 | 34.38 | 0.78 | 40.24 | 0.71 |
| Cheetor$_V$ | 24.72 | 1.03 | 35.76 | 0.77 | 31.21 | 0.88 | 30.56 | 0.89 |
| Cheetor$_{L_2}$ | 29.10 | 0.77 | 40.85 | 0.69 | 33.53 | 0.73 | 34.49 | 0.73 |
| BLIVA | 30.64 | 0.85 | 35.25 | 0.61 | 34.12 | 0.59 | 33.34 | 0.68 |
| **Likelihood Evaluation** | | | | | | | | |
| BLIP-2$_F$ | 48.78 | 0.05 | 61.36 | 0.11 | 43.21 | 0.13 | 51.12 | 0.10 |
| InstructBLIP$_F$ | 48.29 | 0.08 | 60.51 | 0.17 | 44.82 | 0.12 | 51.21 | 0.12 |
| InstructBLIP$_V$ | **53.19** | 0.06 | 59.15 | 0.19 | 44.40 | 0.16 | 52.25 | 0.14 |
| LLaVA$_V$ | 38.96 | 0.24 | 52.54 | 0.21 | 35.81 | 0.31 | 42.44 | 0.25 |
| LLaVA$_{L_2}$ | 45.73 | 0.22 | 59.66 | 0.16 | 36.66 | 0.22 | 47.35 | 0.20 |
| MiniGPT4 | 49.37 | 0.39 | 57.12 | 0.17 | 41.18 | 0.21 | 49.22 | 0.26 |
| mPLUG-Owl | 46.14 | 0.18 | 59.15 | 0.17 | 40.59 | 0.22 | 48.63 | 0.19 |
| PandaGPT | 36.67 | 0.31 | 52.03 | 0.29 | 29.60 | 0.33 | 39.43 | 0.31 |
| IB-LLM | 43.39 | 0.20 | 54.07 | 0.16 | 40.89 | 0.20 | 46.12 | 0.19 |
| LA-V2 | 42.92 | 0.14 | 60.85 | 0.15 | 42.16 | 0.18 | 48.64 | 0.16 |
| mmGPT | 49.91 | 0.15 | 50.85 | 0.23 | 40.85 | 0.20 | 47.20 | 0.19 |
| Shikra | 42.72 | 0.11 | 57.12 | 0.25 | 36.62 | 0.23 | 45.49 | 0.20 |
| Lynx | 51.77 | 0.12 | **66.27** | 0.13 | 43.63 | 0.25 | **53.89** | 0.17 |
| Cheetor$_V$ | 48.61 | 0.20 | 60.00 | 0.19 | 36.49 | 0.33 | 48.37 | 0.24 |
| Cheetor$_{L_2}$ | 47.33 | 0.20 | 58.31 | 0.18 | 40.34 | 0.20 | 48.66 | 0.19 |
| BLIVA | 46.52 | 0.05 | 63.39 | 0.20 | **45.20** | 0.18 | 51.70 | 0.14 |

Table 17: Evaluation results on spatial understanding.

questions, however, the in-context samples still assist in addressing issues on the remaining samples. For qualitative analysis, we provide some examples in Figure 11, the questions are from VQA v2.

## C.3 EFFECT OF MODEL PARAMETERS

We also examine the effect of model parameters on different evaluations, and these model parameters are categorized into four types, including number of total parameters (#oP), number of trainable parameters (#oTP), number of trainable LLM parameters (#oLTP), and number of visual tokens (#oVT). According to evaluation results, we consider the score of each model in a single task as an individual sample. These samples are partitioned into multiple groups based on different parameter scales and we generate boxplots for each group, shown in Figure 8 and Figure 9.

For the impact of total parameters on evaluation results, we round the parameters into three groups, including "4B", "7B", and "8B". BLIP-2 and InstructBLIP achieve superior results with fewer parameters, utilizing FlanT5-XL LLM. Except that, in the range from 7B to 8B, the score exhibits an increasing tendency with growth in total parameters.

For the effect of trainable parameters, it is observed that trainable parameters range of 100M to 1B may be an ideal scenario. Further, by analyzing different scales of trainable parameters from 0 to 1B, we conclude that: (1) For models with trainable parameters scales of 0-10M, primarily linear layers are trained but still achieve competitive results on these evaluation tasks. (2) For models with

| Model | ITM | | | | | | | | VE | | | | Avg. | |
|---|---|---|---|---|---|---|---|---|---|---|---|---|---|---|
| | MSCOCO$_{itm}$ | | MSCOCO$_{its}$ | | WikiHow | | Winoground | | SNLI-VE | | MOCHEG | | | |
| | Acc | Instability | Acc | Instability | Acc | Instability | Acc | Instability | Acc | Instability | Acc | Instability | Acc | Instability |
| **Generation Evaluation** | | | | | | | | | | | | | | |
| BLIP-2$_F$ | 96.40 | 0.04 | 97.53 | 0.02 | 33.67 | 0.26 | 55.25 | 0.32 | 72.20 | 0.15 | **46.33** | 0.02 | 66.90 | 0.14 |
| InstructBLIP$_F$ | **97.63** | 0.02 | **98.27** | 0.01 | **46.87** | 0.20 | **64.50** | 0.25 | **74.80** | 0.11 | 46.09 | 0.24 | **71.36** | 0.14 |
| InstructBLIP$_V$ | 63.73 | 0.24 | 96.33 | 0.04 | 33.13 | 0.17 | 55.00 | 0.51 | 32.33 | 0.00 | 42.07 | 0.19 | 53.76 | 0.19 |
| LLaVA$_V$ | 51.40 | 0.10 | 77.67 | 0.23 | 29.47 | 0.47 | 46.00 | 0.55 | 35.87 | 0.33 | 43.37 | 0.31 | 47.30 | 0.33 |
| LLaVA$_{L_2}$ | 52.17 | 0.01 | 87.20 | 0.11 | 34.00 | 0.29 | 49.00 | 0.42 | 33.13 | 0.07 | 44.02 | 0.56 | 49.91 | 0.24 |
| MiniGPT4 | 52.20 | 0.00 | 61.87 | 0.26 | 27.87 | 0.36 | 53.00 | 0.57 | 37.33 | 0.29 | 40.36 | 0.65 | 45.44 | 0.36 |
| mPLUG-Owl | 51.63 | 0.19 | 42.00 | 0.68 | 26.20 | 0.79 | 49.50 | 0.56 | 36.60 | 0.62 | 36.33 | 0.65 | 40.38 | 0.58 |
| PandaGPT | 52.10 | 0.00 | 22.47 | 0.36 | 23.47 | 0.32 | 51.50 | 0.57 | 34.40 | 0.06 | 38.46 | 0.64 | 37.06 | 0.33 |
| IB-LLM | 51.87 | 0.13 | 29.20 | 0.43 | 21.53 | 0.54 | 48.75 | 0.54 | 32.07 | 0.54 | 35.92 | 0.66 | 36.56 | 0.47 |
| LA-V2 | 52.20 | 0.00 | 42.00 | 0.73 | 25.93 | 0.83 | 48.00 | 0.54 | 37.00 | 0.69 | 41.24 | 0.14 | 41.06 | 0.49 |
| mmGPT | 51.73 | 0.29 | 32.00 | 0.87 | 24.47 | 0.86 | 50.25 | 0.59 | 32.33 | 0.59 | 38.34 | 0.57 | 38.19 | 0.63 |
| Shikra | 30.20 | 0.13 | 81.40 | 0.26 | 37.07 | 0.23 | 51.75 | 0.58 | 34.47 | 0.20 | 32.35 | 0.73 | 44.54 | 0.36 |
| Lynx | 51.13 | 0.08 | 91.33 | 0.12 | 33.27 | 0.43 | 55.25 | 0.51 | 50.07 | 0.41 | 36.92 | 0.67 | 53.00 | 0.37 |
| Cheetor$_V$ | 53.60 | 0.08 | 71.47 | 0.33 | 31.40 | 0.62 | 53.00 | 0.52 | 35.13 | 0.49 | 39.88 | 0.63 | 47.41 | 0.45 |
| Cheetor$_{L_2}$ | 52.27 | 0.02 | 52.67 | 0.32 | 30.40 | 0.31 | 50.50 | 0.49 | 32.80 | 0.01 | 45.44 | 0.50 | 44.01 | 0.28 |
| BLIVA | 50.50 | 0.55 | 67.27 | 0.29 | 30.47 | 0.33 | 47.00 | 0.58 | 33.80 | 0.22 | 42.25 | 0.47 | 45.22 | 0.41 |
| **Likelihood Evaluation** | | | | | | | | | | | | | | |
| BLIP-2$_F$ | **96.37** | 0.04 | 62.07 | 0.14 | 32.47 | 0.06 | 58.50 | 0.04 | **57.73** | 0.08 | **46.33** | 0.09 | 58.91 | 0.08 |
| InstructBLIP$_F$ | 90.97 | 0.10 | 50.00 | 0.09 | 30.80 | 0.10 | 62.75 | 0.08 | 54.57 | 0.12 | 43.91 | 0.23 | 55.50 | 0.12 |
| InstructBLIP$_V$ | 87.37 | 0.19 | 61.33 | 0.10 | 29.67 | 0.13 | **68.00** | 0.04 | 49.73 | 0.39 | 36.27 | 0.13 | 55.40 | 0.16 |
| LLaVA$_V$ | 48.30 | 0.01 | 72.40 | 0.09 | 30.47 | 0.17 | 63.75 | 0.06 | 39.13 | 0.39 | 33.96 | 0.16 | 48.00 | 0.15 |
| LLaVA$_{L_2}$ | 64.13 | 0.17 | 66.67 | 0.07 | 31.60 | 0.13 | 58.00 | 0.03 | 37.60 | 0.07 | 39.88 | 0.21 | 49.65 | 0.11 |
| MiniGPT4 | 78.27 | 0.18 | 60.13 | 0.10 | 30.27 | 0.11 | 65.00 | 0.01 | 40.73 | 0.47 | 31.18 | 0.41 | 50.93 | 0.21 |
| mPLUG-Owl | 53.50 | 0.02 | 68.53 | 0.07 | 31.13 | 0.09 | 65.75 | 0.03 | 36.87 | 0.12 | 42.78 | 0.34 | 49.76 | 0.11 |
| PandaGPT | 49.13 | 0.47 | 26.27 | 0.15 | 26.40 | 0.21 | 47.25 | 0.13 | 33.80 | 0.52 | 38.88 | 0.40 | 36.96 | 0.31 |
| IB-LLM | 52.13 | 0.00 | 61.87 | 0.07 | 29.53 | 0.11 | 55.00 | 0.03 | 33.60 | 0.03 | 41.42 | 0.04 | 45.59 | 0.05 |
| LA-V2 | 64.50 | 0.26 | 60.20 | 0.07 | 29.80 | 0.11 | 64.00 | 0.03 | 39.27 | 0.42 | 41.36 | 0.02 | 49.86 | 0.15 |
| mmGPT | 52.17 | 0.00 | 51.93 | 0.08 | 28.53 | 0.09 | 58.25 | 0.10 | 32.33 | 0.00 | 41.54 | 0.00 | 44.13 | 0.05 |
| Shikra | 90.63 | 0.14 | **78.13** | 0.08 | 31.87 | 0.08 | 64.25 | 0.03 | 49.40 | 0.10 | 41.36 | 0.00 | **59.27** | 0.07 |
| Lynx | 96.23 | 0.04 | 72.73 | 0.06 | **33.00** | 0.09 | 63.75 | 0.05 | 43.00 | 0.15 | 35.98 | 0.17 | 57.44 | 0.09 |
| Cheetor$_V$ | 79.07 | 0.26 | 58.07 | 0.17 | 29.93 | 0.21 | 62.00 | 0.05 | 40.67 | 0.54 | 34.08 | 0.10 | 50.64 | 0.22 |
| Cheetor$_{L_2}$ | 56.13 | 0.08 | 63.33 | 0.10 | 29.80 | 0.16 | 58.50 | 0.06 | 34.40 | 0.05 | 41.12 | 0.17 | 47.21 | 0.10 |
| BLIVA | 83.30 | 0.27 | 59.33 | 0.14 | 31.40 | 0.12 | 63.75 | 0.10 | 42.40 | 0.19 | 42.25 | 0.25 | 53.74 | 0.18 |

Table 18: Evaluation results on cross-modal inference.

| Model | Image Captioning | | | | Avg. |
|---|---|---|---|---|---|
| | COCO | TextCaps | NoCaps | Flickr30K | |
| BLIP-2$_F$ | **97.48** | **41.56** | **83.57** | **74.63** | **74.31** |
| InstructBLIP$_F$ | 54.79 | 16.38 | 45.31 | 58.63 | 43.78 |
| InstructBLIP$_V$ | 30.97 | 17.16 | 30.18 | 30.77 | 27.27 |
| LLaVA$_V$ | 47.16 | 21.79 | 42.43 | 35.78 | 36.79 |
| LLaVA$_{L_2}$ | 50.74 | 24.49 | 45.44 | 37.45 | 39.53 |
| MiniGPT4 | 57.20 | 29.19 | 58.71 | 44.71 | 47.45 |
| mPLUG-Owl | 59.36 | 24.25 | 48.43 | 46.61 | 44.66 |
| PandaGPT | 2.24 | 0.95 | 1.12 | 1.93 | 1.56 |
| IB-LLM | 38.15 | 16.45 | 32.83 | 23.14 | 27.64 |
| LA-V2 | 44.60 | 22.10 | 41.06 | 36.08 | 35.96 |
| mmGPT | 35.50 | 18.68 | 33.20 | 23.45 | 27.71 |
| Shikra | 41.01 | 19.76 | 37.42 | 28.91 | 31.78 |
| Lynx | 80.04 | 34.43 | 77.31 | 51.04 | 60.71 |
| Cheetor$_V$ | 86.90 | 32.70 | 73.99 | 52.88 | 61.62 |
| Cheetor$_{L_2}$ | 72.80 | 21.64 | 44.39 | 36.63 | 43.86 |
| BLIVA | 62.23 | 36.72 | 64.21 | 46.90 | 52.51 |

Table 19: Evaluation results on visual description based on CIDEr.

10-100M trainable parameters, all of them implement LoRA or bias-tuning, and these parameter-efficient fine-tuning approaches even have negative impacts on generation and likelihood tasks. (3) For models with 100M-1B trainable parameters, Q-Former or Perceiver are trained and applied as connection modules, which enhance the capacity of models across various tasks.

In the context of LLM trainable parameters, fine-tuning all the parameters of LLMs or adding an additional small amount of trainable parameters to the LLMs does not contribute to the overall performance of LVLMs. However, adding a sufficient and appropriate number of trainable parameters to the LLM may have the potential to improve models' performances on visual language tasks.

With regard to the number of visual tokens shown in Figure 9, the model's performance shows a decline with a visual token count of 1 and 10, performs optimally at 32 but deteriorates when further increased to 64 or 256, which still outperforms models with only 1 or 10 visual tokens. This indicates that a marginal increase in the number of visual tokens may be beneficial for enhancing model performance. However, the observed outcomes that poor performance exhibited by models with 1-10 VT are attributed to the joint influence of two factors: the use of ImageBind visual encoder and

| Model | Image Captioning | | | | | | | | | | | |
| --- | --- | --- | --- | --- | --- | --- | --- | --- | --- | --- | --- | --- |
| | COCO | | | TextCaps | | | NoCaps | | | Flickr30K | | |
| | BLEU-4 | METEOR | ROUGE-L | BLEU-4 | METEOR | ROUGE-L | BLEU-4 | METEOR | ROUGE-L | BLEU-4 | METEOR | ROUGE-L |
| BLIP-2$_F$ | 30.14 | 26.71 | 49.57 | 16.84 | 18.95 | 34.26 | 37.18 | 27.09 | 49.34 | 23.63 | 23.37 | 44.24 |
| InstructBLIP$_F$ | 14.24 | 18.85 | 34.98 | 2.74 | 10.63 | 18.58 | 15.59 | 17.60 | 32.94 | 19.07 | 20.51 | 38.34 |
| InstructBLIP$_V$ | 5.91 | 13.09 | 23.96 | 3.30 | 10.51 | 18.19 | 7.71 | 13.48 | 25.13 | 6.26 | 12.09 | 24.90 |
| LLaVA$_V$ | 14.48 | 20.47 | 34.87 | 6.86 | 15.34 | 25.99 | 17.39 | 21.67 | 36.27 | 11.76 | 21.87 | 33.10 |
| LLaVA$_{L_2}$ | 14.85 | 21.26 | 37.98 | 8.25 | 16.32 | 28.33 | 18.31 | 22.50 | 39.15 | 13.93 | 22.22 | 35.90 |
| MiniGPT4 | 18.31 | 22.47 | 37.65 | 9.66 | 17.34 | 29.22 | 22.84 | 24.99 | 40.83 | 13.82 | 22.28 | 34.48 |
| mPLUG-Owl | 18.30 | 21.55 | 40.19 | 7.32 | 15.56 | 27.21 | 17.35 | 21.68 | 37.66 | 16.57 | 23.34 | 39.93 |
| PandaGPT | 1.31 | 8.77 | 21.34 | 1.38 | 7.85 | 20.89 | 1.74 | 8.98 | 22.38 | 0.00 | 7.44 | 17.62 |
| IB-LLM | 11.27 | 18.76 | 32.30 | 5.14 | 13.79 | 24.66 | 12.53 | 18.79 | 32.56 | 6.91 | 17.10 | 27.73 |
| LA-V2 | 13.06 | 19.78 | 33.51 | 6.82 | 15.47 | 25.59 | 15.45 | 21.32 | 35.61 | 10.93 | 21.82 | 32.29 |
| mmGPT | 9.08 | 16.89 | 29.89 | 4.66 | 13.74 | 23.89 | 10.90 | 18.29 | 31.44 | 7.61 | 18.19 | 27.91 |
| Shikra | 12.47 | 19.30 | 31.65 | 6.25 | 14.92 | 23.35 | 13.68 | 20.51 | 32.70 | 9.47 | 20.77 | 28.40 |
| Lynx | 24.00 | 24.80 | 44.37 | 13.08 | 18.25 | 33.45 | 30.52 | 26.26 | 47.30 | 18.96 | 22.31 | 37.72 |
| Cheetor$_V$ | 28.12 | 26.01 | 50.55 | 12.51 | 17.88 | 33.34 | 31.96 | 26.66 | 49.99 | 22.71 | 25.23 | 43.47 |
| Cheetor$_{L_2}$ | 23.03 | 23.44 | 46.47 | 8.53 | 15.13 | 28.66 | 17.87 | 20.34 | 39.27 | 14.33 | 21.45 | 39.13 |
| BLIVA | 11.57 | 20.76 | 35.67 | 12.04 | 18.73 | 30.67 | 21.89 | 23.06 | 40.63 | 8.40 | 19.32 | 33.10 |

Table 20: Evaluation results on visual description based on BLEU-4, METEOR and ROUGE-L.

| Model | VQA-MT | | | VisDial | | | Avg. | |
| --- | --- | --- | --- | --- | --- | --- | --- | --- |
| | Acc | Instability | Corr | Acc | Instability | Corr | Acc | Instability |
| Generation Evaluation | | | | | | | | |
| BLIP-2$_F$ | 67.97 | 0.20 | -0.26 | 55.53 | 0.24 | -0.93 | 61.75 | 0.22 |
| InstructBLIP$_F$ | 68.67 | 0.19 | -0.57 | 52.51 | 0.24 | -0.84 | 60.59 | 0.22 |
| InstructBLIP$_V$ | 56.58 | 0.48 | -0.65 | 40.68 | 0.61 | -0.97 | 48.63 | 0.55 |
| LLaVA$_V$ | 40.06 | 0.76 | 0.16 | 31.15 | 0.77 | -0.92 | 35.61 | 0.77 |
| LLaVA$_{L_2}$ | 50.47 | 0.54 | -0.68 | 42.11 | 0.46 | -0.82 | 46.29 | 0.50 |
| MiniGPT4 | 43.98 | 0.23 | -0.31 | 35.05 | 0.66 | -0.89 | 39.52 | 0.45 |
| mPLUG-Owl | 38.66 | 0.77 | 0.71 | 31.85 | 0.80 | -0.83 | 35.23 | 0.79 |
| PandaGPT | 34.73 | 0.64 | 0.22 | 33.44 | 0.63 | -0.68 | 34.09 | 0.64 |
| IB-LLM | 37.24 | 0.66 | 0.14 | 33.26 | 0.69 | -0.78 | 35.25 | 0.68 |
| LA-V2 | 38.88 | 0.72 | -0.04 | 32.00 | 0.76 | -0.85 | 35.44 | 0.74 |
| mmGPT | 34.92 | 0.80 | 0.86 | 28.75 | 0.90 | -0.57 | 31.84 | 0.85 |
| Shikra | 43.33 | 0.67 | 0.68 | 27.12 | 0.91 | -0.93 | 35.23 | 0.79 |
| Lynx | 54.59 | 0.38 | -0.71 | 39.43 | 0.60 | -0.84 | 47.01 | 0.49 |
| Cheetor$_V$ | 44.40 | 0.55 | -0.57 | 36.14 | 0.59 | -0.89 | 40.27 | 0.57 |
| Cheetor$_{L_2}$ | 41.36 | 0.49 | -0.72 | 39.80 | 0.39 | -0.74 | 40.58 | 0.44 |
| BLIVA | 48.83 | 0.57 | -0.08 | 30.80 | 0.75 | -0.95 | 39.82 | 0.66 |
| Likelihood Evaluation | | | | | | | | |
| BLIP-2$_F$ | 71.52 | 0.04 | -0.10 | 53.62 | 0.04 | -0.06 | 62.57 | 0.04 |
| InstructBLIP$_F$ | 77.06 | 0.06 | 0.23 | 57.34 | 0.04 | -0.67 | 67.20 | 0.05 |
| InstructBLIP$_V$ | 78.06 | 0.04 | -0.21 | 59.30 | 0.04 | -0.66 | 68.68 | 0.04 |
| LLaVA$_V$ | 61.32 | 0.05 | 0.47 | 43.24 | 0.04 | -0.83 | 52.28 | 0.05 |
| LLaVA$_{L_2}$ | 56.24 | 0.06 | 0.65 | 40.97 | 0.03 | -0.46 | 48.61 | 0.05 |
| MiniGPT4 | 63.97 | 0.06 | 0.67 | 44.14 | 0.05 | -0.54 | 54.06 | 0.06 |
| mPLUG-Owl | 52.38 | 0.04 | 0.68 | 38.57 | 0.03 | -0.69 | 45.48 | 0.04 |
| PandaGPT | 43.71 | 0.18 | 0.91 | 39.21 | 0.09 | 0.42 | 41.46 | 0.14 |
| IB-LLM | 43.11 | 0.03 | 0.50 | 35.86 | 0.02 | -0.61 | 39.49 | 0.03 |
| LA-V2 | 47.29 | 0.08 | 0.55 | 39.49 | 0.04 | -0.30 | 43.39 | 0.06 |
| mmGPT | 47.52 | 0.06 | -0.55 | 38.57 | 0.03 | -0.69 | 43.05 | 0.05 |
| Shikra | 69.15 | 0.06 | 0.29 | 49.76 | 0.03 | -0.65 | 59.46 | 0.05 |
| Lynx | 67.75 | 0.08 | 0.29 | 52.22 | 0.05 | -0.70 | 59.99 | 0.07 |
| Cheetor$_V$ | 66.01 | 0.06 | 0.53 | 49.22 | 0.06 | 0.29 | 57.62 | 0.06 |
| Cheetor$_{L_2}$ | 51.86 | 0.10 | -0.69 | 41.82 | 0.05 | -0.78 | 46.84 | 0.08 |
| BLIVA | 77.92 | 0.05 | 0.18 | 58.24 | 0.04 | -0.65 | 68.08 | 0.05 |

Table 21: Evaluation results on multi-turn Dialogue. "Corr" represents the correlation coefficient between the model performance and the number of dialogue turns.

the restricted number of visual tokens. Since there are only three models in this range and two of them utilize the ImageBind encoder, distinguishing whether the observed degradation is primarily attributable to the suboptimal performance of the ImageBind encoder or limited visual tokens remains uncertain. Moreover, increasing the number of visual tokens from 32 does not inherently lead to augmented model capability.

## C.4 EFFECT OF DATASET

**High-Quality Pre-training Dataset** To quantitatively assess the impact of the COCO dataset during the pre-training stage, we conduct an evaluation using two groups of models, one group includes three models (BLIP2, mPlug-Owl, and Lynx), while the other group consists of two models (LLaVA

| Backbone | FlanT5-xl | | LLaMA-7B | | | Vicuna-7B | | |
|---|---|---|---|---|---|---|---|---|
| Model | BLIP-2 | InstructBLIP | ImageBind-LLM | LA-V2 | mPLUG-Owl | Cheetor | BLIVA | InstructBLIP |
| Hit Rate | 100.0 | 99.99 | 99.94 | 85.14 | 62.86 | 99.97 | 99.77 | 99.99 |
| Hit Rate+ | 100.0 | 100.0 | 100.0 | 100 | 100 | 100.0 | 100.0 | 100.0 |

| Backbone | Vicuna-7B | Vicuna-7B+ | | Vicuna-7B+Δ | | LLaMA2-7B-Chat | | OpenFlamingo |
|---|---|---|---|---|---|---|---|---|
| Model | MiniGPT-4 | LLaVA | Shikra | Lynx | PandaGPT | LLaVA | Cheetor | mmGPT |
| Hit Rate | 100.0 | 85.32 | 65.42 | 94.41 | 99.41 | 100.0 | 99.31 | 95.71 |
| Hit Rate+ | 100.0 | 100.0 | 97.72 | 100.0 | 99.97 | 100.0 | 100.0 | 99.97 |

Table 22: Complete results for Table 3. "+Δ" represents delta tuning with parameter-efficient modules like LoRA and adapters.

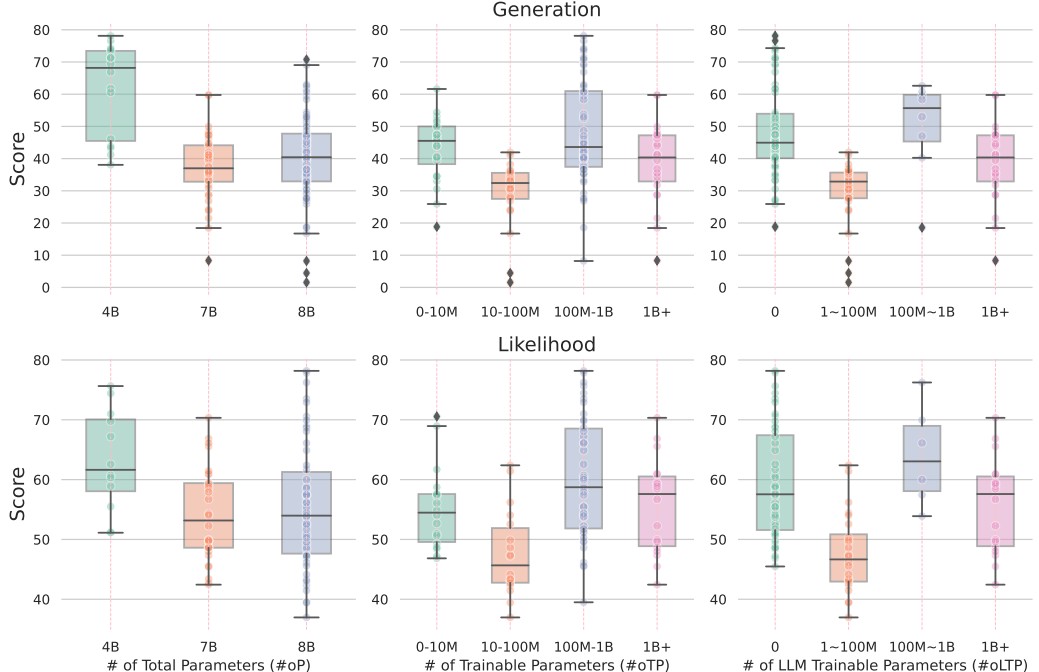

Figure 8: Performance under different numbers of total parameters (#oP), trainable parameters (#oTP), and LLM trainable parameters (#oLTP). Note that, regarding the "# of Total Parameters (#oP)", models around 4B parameters include only BLIP2-FlanT5-XL and InstructBLIP-FlanT5-XL.

and MiniGPT4) not trained with COCO. For fair evaluation, we choose 6 in-domain tasks (containing images from COCO) and 6 out-domain tasks (not containing images from COCO) across perception and cognition capacity. To mitigate the influence of score fluctuations across tasks, we utilize the average rank within each group as a metric to evaluate their collective performance on particular tasks. The selected tasks and the average rank for each group are listed in Table 23.

**Scaling Up Pre-Training Dataset**   To explore the influence of high-quality pre-training data, we select two groups of models: one group (including LLaVA$_V$, MiniGPT4, ImageBind-LLM, and mPLUG-Owl) utilizes data filtered based on rules or CLIP like CC (Sharma et al., 2018) and LAION (Schuhmann et al., 2021), while the other group (including LA-V2, Shikra, BLIP2, and Lynx) is pre-trained using relatively high-quality data like COCO (Chen et al., 2015) and BlipCap-Filt (Li et al., 2022) as indicated in Table 8.

**Instruct-Tuning Dataset**   Based on the data in Table 8, we calculate the number of instruction-tuning samples for each model and plot the relationship between the number of instruction-tuning samples and the average rank.

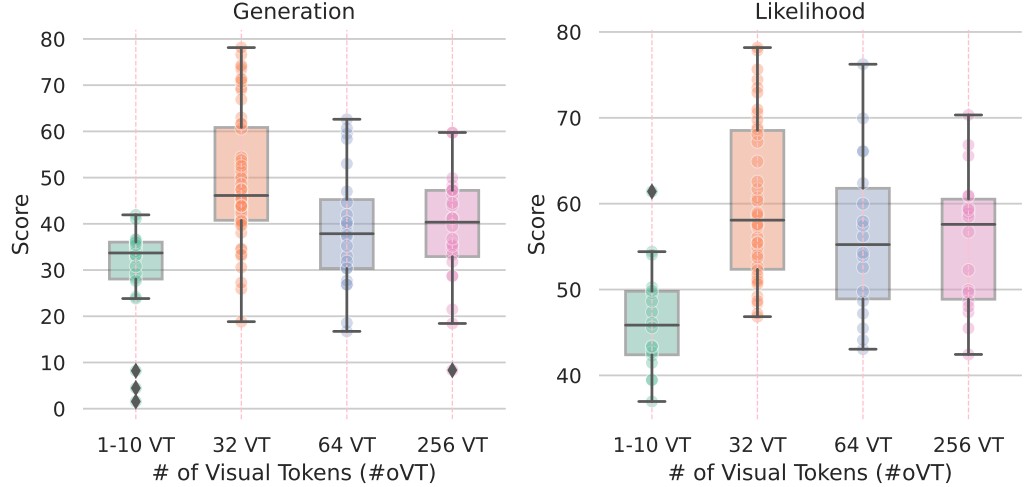

Figure 9: Performance under different numbers of visual tokens (#oVT). It is noteworthy that the group of 1-10VT consists of only two models with 1 visual token (1VT) and one model with 10 visual tokens (10VT). Specifically, 1VT models encompass PandaGPT and IB-LLM, where both of them apply an ImageBind visual encoder.

| Model Group | In-Domain Task | | | | | |
| --- | --- | --- | --- | --- | --- | --- |
| | $MSCOCO_{mci}$ | $MSCOCO_{goi}$ | COCO-Caption | Flickr30K | $MSCOCO_{itm}$ | $MSCOCO_{its}$ |
| Pre-Training w. COCO | **6** | 5.67 | **8** | 7.33 | **6.33** | **5** |
| Pre-Training w.o. COCO | 6.5 | **5.5** | 8.5 | 9 | 12 | 7 |
| Model Group | Out-Domain Task | | | | | |
| | Pets37 | Flowers102 | Visdial | NoCaps | Wikihow | Winoground |
| Pre-Training w. COCO | **3.67** | **3.33** | **7.33** | 7.67 | **6.67** | **3.33** |
| Pre-Training w.o. COCO | 8 | 7 | 10.5 | **7.5** | 8.5 | 4.5 |

Table 23: Average rank of model groups in in-domain and out-domain tasks.

## C.5 INSTABILITY

Table 24 provides the complete results of models' instability caused by different perturbations. Under the generation evaluation, all models are most sensitive to the order of options, followed by the option marks, and lastly, random instructions. FlanT5 models are the most stable models under the generation evaluation, showing that FlanT5 can well comprehend the multiple-choice questions. For likelihood evaluation, all models are stable since the evaluation strategy directly utilizes the characteristics of generative models.

To further perceive the influence of instruction perturbation on the answer accuracy, we analyze the above instruction perturbation results. As we employ different instructions to describe the same task, the accuracy of samples that follow each instruction can be calculated. For the accuracy of each instruction, we adopt the difference between the maximum and minimum accuracies to represent the model's instability level towards the instruction. The results are shown in Table 25. We discover that all models exhibit some fluctuations in accuracy, illustrating that LVLMs are sensitive to designed prompts. However, the fluctuations in accuracy under generation and likelihood evaluation of most LVLMs are both within an acceptable range. There are still models exhibiting fluctuations in accuracy exceeding $10\%$, indicating the restricted instruction-following capabilities of LVLMs. In general, LVLMs require further improvements to enhance its ability to understand and follow diverse instructions.

| Model | Generation | | | Likelihood |
|---|---|---|---|---|
| | Instruct | Option Order | Option Mark | Instruct |
| $BLIP2_F$ | 0.029 | 0.276 | 0.107 | 0.037 |
| $InstructBLIP_F$ | 0.028 | 0.242 | 0.105 | 0.028 |
| $InstructBLIP_V$ | 0.038 | 0.414 | 0.182 | 0.018 |
| $LLaVA_V$ | 0.197 | 0.606 | 0.299 | 0.105 |
| $LLaVA_{L_2}$ | 0.141 | 0.464 | 0.178 | 0.090 |
| MiniGPT4 | 0.113 | 0.647 | 0.194 | 0.043 |
| mPLUG-Owl | 0.330 | 0.706 | 0.406 | 0.046 |
| PandaGPT | 0.125 | 0.592 | 0.198 | 0.117 |
| IB-LLM | 0.159 | 0.702 | 0.498 | 0.024 |
| LA-V2 | 0.382 | 0.682 | 0.518 | 0.032 |
| mmGPT | 0.578 | 0.763 | 0.601 | 0.030 |
| Shikra | 0.028 | 0.617 | 0.206 | 0.054 |
| Lynx | 0.069 | 0.375 | 0.195 | 0.052 |
| $Cheetor_V$ | 0.177 | 0.666 | 0.356 | 0.076 |
| $Cheetor_{L_2}$ | 0.051 | 0.476 | 0.163 | 0.058 |
| BLIVA | 0.128 | 0.610 | 0.204 | 0.023 |
| **Average** | **0.161** | **0.552** | **0.276** | **0.049** |

Table 24: Instability of models caused by different random perturbations.

| Model | Generation | Likelihood |
|---|---|---|
| $BLIP2_F$ | 6.47 | 6.96 |
| $InstructBLIP_F$ | 3.48 | 5.97 |
| $InstructBLIP_V$ | 4.48 | 5.97 |
| $LLaVA_V$ | 3.48 | 6.47 |
| $LLaVA_{L_2}$ | 1.99 | 7.46 |
| MiniGPT4 | 3.48 | 5.97 |
| mPLUG-Owl | 4.98 | 6.47 |
| PandaGPT | 5.47 | 6.47 |
| IB-LLM | 7.46 | 3.98 |
| LA-V2 | 3.48 | 3.98 |
| mmGPT | 10.45 | 4.48 |
| Shikra | 12.94 | 2.99 |
| Lynx | 6.97 | 9.95 |
| $Cheetor_V$ | 3.48 | 5.97 |
| $Cheetor_{L_2}$ | 6.47 | 7.46 |
| BLIVA | 1.99 | 9.95 |
| **Average** | **5.44** | **6.28** |

Table 25: The difference between the maximum and minimum accuracies of all instruction groups.

## C.6 OPTION PREFERENCE

Option preference is a phenomenon in our benchmark that when uncertain about the answer, LVLMs prefer a particular option regardless of options' content. We verify the option preference inside the LVLMs in Figure 10. It has been observed that ImageBind-LLM, Shikra, and BLIVA exhibit a preference for option "A" when confronted with uncertainty. MiniGPT4, mPLUG-Owl, PandaGPT, LA-V2 and mmGPT show a strong preference for option "B". Other LVLMs show no obvious preference in this task. It's worth noting that predicted choice distribution under the likelihood evaluation method has no preference, as all options are considered in an unordered state.

The phenomenon of option preference contributes to the instability from random option order but reduces that from random instruction and option mark (as mentioned in Section 4.3.5). Concretely, when LVLMs are uncertain about answers, they select the ceratin option repeatedly for the essentially identical questions. As the option contents have been shuffled in random option mark mode, the LVLMs are regarded as selecting distinct answers. Regarding random instruction and option mark situations, LVLMs are firm in their answers regardless variation of question form.

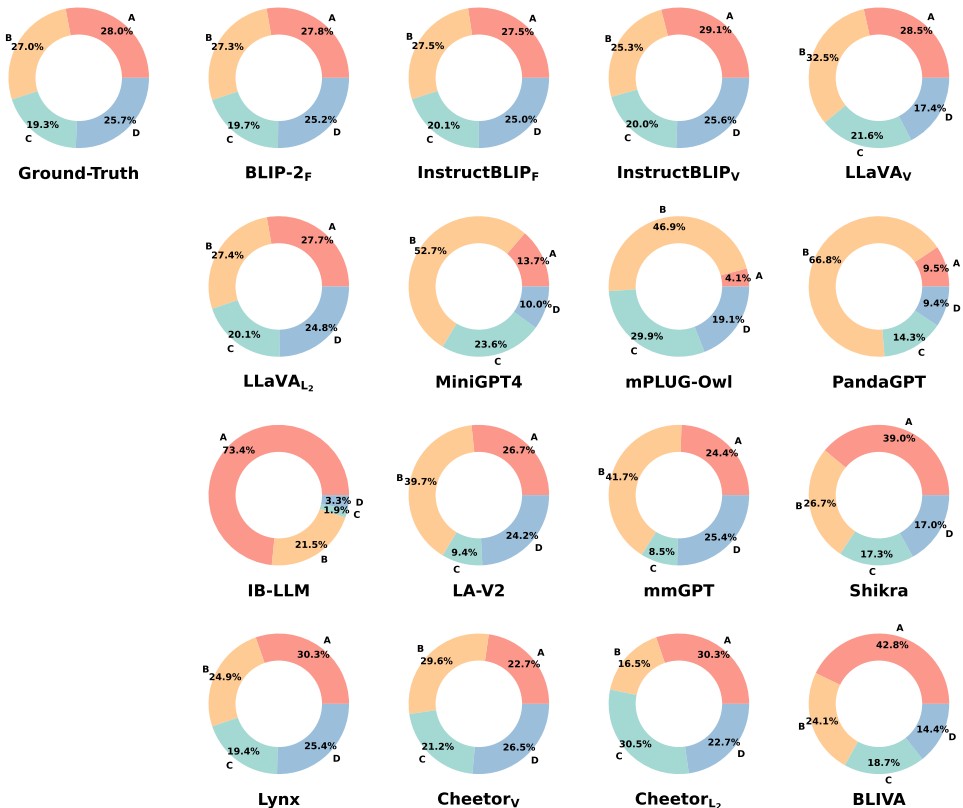

Figure 10: The choice distribution of ground-truth answers and prediction of all LVLMs in COCO image text selection task under generation method.

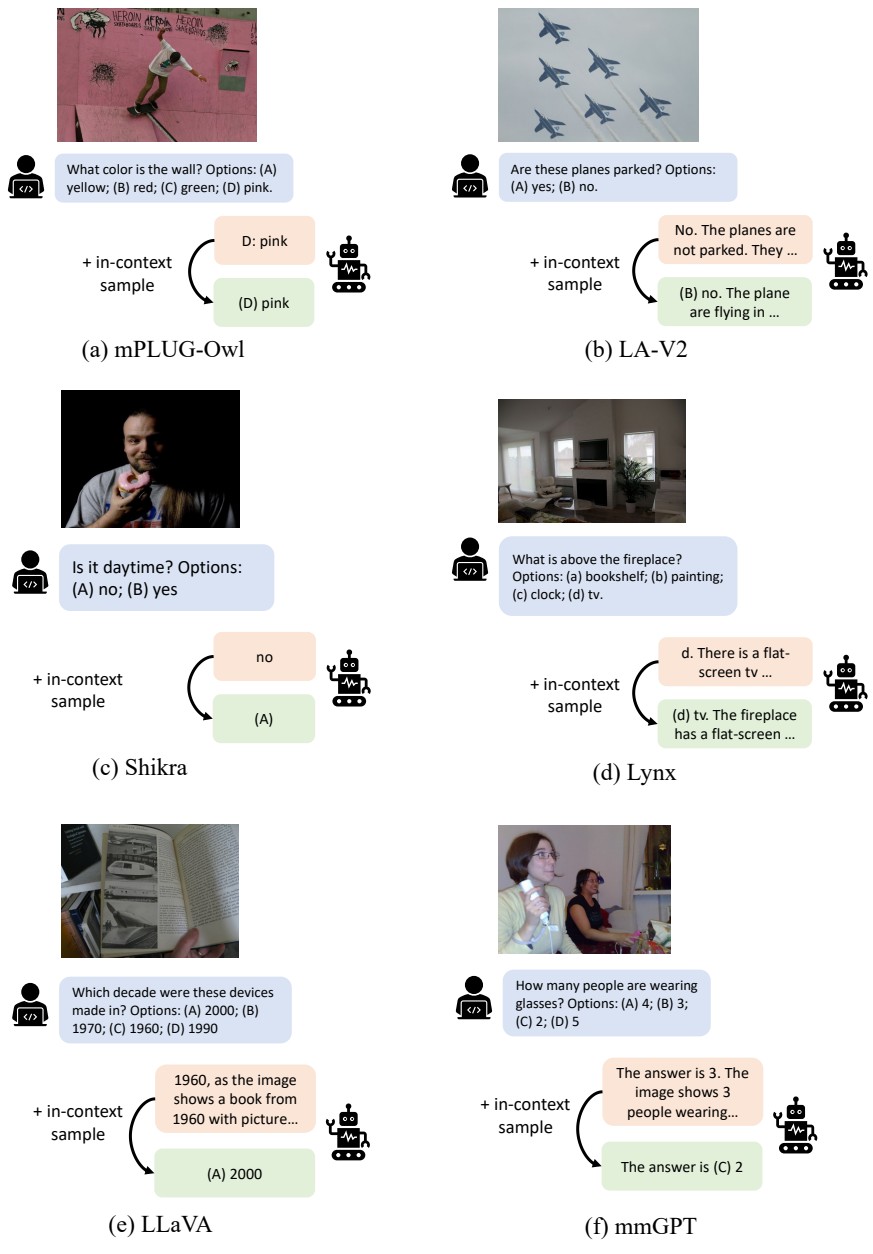

Figure 11: Case study of the effect of in-context samples, the instruction used is "Take a close look at the image and question, and then choose the correct option" which is omitted for simplicity.

| Model | Generation Evaluation | | | | | | | | | Likelihood Evaluation | | | | | | |
| | Perception | | | Cognition | | | | | $R$ | Perception | | Cognition | | | | $R$ |
| | CG | FG | STP | Spatial | VGR | Dialog | CMI | Desc | | CG | FG | Spatial | VGR | Dialog | CMI | |
| BLIP-2$_F$ | 74.6 | 67.1 | 43.4 | 43.2 | 71.8 | 55.5 | 45.1 | 83.6 | 2 | 62.0 | 64.7 | 51.1 | 66.2 | 53.6 | 45.8 | 5 |
| InstructBLIP$_F$ | 76.4 | 68.2 | 46.2 | 46.1 | 70.4 | 52.5 | 52.5 | 45.3 | 2 | 59.8 | 64.7 | 51.2 | 66.3 | 57.3 | 45.8 | 4 |
| InstructBLIP$_V$ | 73.0 | 62.9 | 44.9 | 44.4 | 64.5 | 40.7 | 43.4 | 30.2 | 4 | 57.2 | 70.9 | 52.3 | 69.0 | 59.3 | 44.7 | 3 |
| LLaVA$_V$ | 27.7 | 31.9 | 19.9 | 28.7 | 47.1 | 31.2 | 39.6 | 42.4 | 12 | 61.7 | 67.4 | 42.4 | 59.4 | 43.2 | 42.7 | 8 |
| LLaVA$_{L_2}$ | 42.2 | 50.1 | 22.6 | 41.2 | 62.8 | 42.1 | 42.3 | 45.4 | 5 | 47.1 | 60.3 | 47.4 | 59.7 | 41.0 | 43.2 | 10 |
| MiniGPT4 | 41.6 | 47.4 | 38.4 | 34.6 | 48.6 | 35.1 | 40.4 | 58.7 | 7 | 50.7 | 66.1 | 49.2 | 55.7 | 44.1 | 42.2 | 9 |
| mPLUG-Owl | 35.9 | 31.7 | 43.0 | 26.8 | 40.8 | 31.9 | 37.3 | 48.4 | 11 | 64.5 | 58.4 | 48.6 | 56.3 | 38.6 | 46.6 | 9 |
| PandaGPT | 26.2 | 28.7 | 4.8 | 33.3 | 48.1 | 33.4 | 37.8 | 1.1 | 13 | 41.2 | 41.8 | 39.4 | 46.4 | 39.2 | 37.5 | 16 |
| IB-LLM | 23.6 | 32.9 | 8.9 | 35.6 | 37.8 | 33.3 | 35.4 | 32.8 | 13 | 51.9 | 55.3 | 46.1 | 52.4 | 35.9 | 42.0 | 14 |
| LA-V2 | 25.4 | 28.0 | 25.2 | 23.8 | 37.4 | 32.0 | 38.4 | 41.1 | 13 | 41.2 | 56.3 | 48.6 | 56.9 | 39.5 | 45.1 | 11 |
| mmGPT | 23.7 | 30.4 | 18.0 | 26.9 | 37.5 | 28.8 | 37.7 | 33.2 | 14 | 53.5 | 56.5 | 47.2 | 56.2 | 38.6 | 42.8 | 12 |
| Shikra | 42.2 | 45.8 | 9.0 | 33.3 | 41.0 | 27.1 | 40.4 | 37.4 | 11 | 59.1 | 59.5 | 45.5 | 56.2 | 49.8 | 45.8 | 9 |
| Lynx | 60.5 | 56.9 | 20.0 | 40.2 | 59.4 | 39.4 | 41.8 | 77.3 | 5 | 69.6 | 69.1 | 53.9 | 65.6 | 52.2 | 44.2 | 4 |
| Cheetor$_V$ | 50.4 | 40.8 | 30.0 | 30.6 | 54.0 | 36.1 | 41.4 | 74.0 | 7 | 54.6 | 62.8 | 48.4 | 56.5 | 49.2 | 42.0 | 10 |
| Cheetor$_{L_2}$ | 43.7 | 41.6 | 22.2 | 34.5 | 58.7 | 39.8 | 42.1 | 44.4 | 7 | 54.9 | 59.9 | 48.7 | 59.6 | 41.8 | 43.1 | 9 |
| BLIVA | 33.7 | 37.5 | 44.6 | 33.3 | 42.6 | 30.8 | 39.9 | 64.2 | 9 | 67.1 | 71.8 | 51.7 | 69.9 | 58.4 | 45.8 | 2 |

Table 26: General zero-shot evaluation results of LVLMs across capability dimensions. "CG", "FG", "CMI", and "Desc" are respectively short for coarse-grained perception, fine-grained perception, cross-modal inference, and description. "$\bar{R}$" represents the average rank across dimensions.

# D ASSESSMENT OF ZERO-SHOT CAPABILITIES

## D.1 SUB-BENCHMARK CONSTRUCTION

In addition to evaluating with the complete data in ReForm-Eval, we introduce two data selection strategies to construct sub-benchmarks from the complete data. This allows users to assess LVLMs from specific perspectives to meet different evaluation requirements.

The first strategy is ***model-oriented***, designed to assess the zero-shot capabilities of models. Specifically, given the models and their corresponding training datasets, we filter out the overlapping parts in ReForm-Eval, retaining only the commonly held-out benchmark datasets for assessment.

The second strategy is ***user-oriented***. Users can select one or more of the 61 benchmarks included in ReForm-Eval to construct a specific sub-benchmark based on the dimensions of abilities, tasks, or other requirements. For example, a biologist might select Flowers102 and Pets37 to construct a sub-benchmark for animal and plant recognition.

## D.2 EXPERIMENTAL SETUP

In order to assess the zero-shot capabilities of LVLMs, we ***follow the first strategy*** introduced in Appendix D.1 to automatically construct a sub-benchmark based on ReForm-Eval, excluding held-in datasets used in the training processes of the 16 compared models. Specifically, we exclude the following benchmarks: ImageNet, TDIUC, TextVQA, SNLI-VE, GQA, VizWiz, and Flickr30K (used by Lynx); refCOCO (used by Shikra); OK-VQA, VQA v2, OK-VQA, A-OKVQA, A-OKVQRA, A-OKVQAR, VQA-MT, and TextCaps (used by InstructBLIP, Lynx, and BLIVA); COCO image captioning, COCO-ITM, and COCO-ITS that are based on the caption data in MS-COCO (used for pre-training most LVLMs). Benefiting from the richness and ample data in ReForm-Eval, the constructed sub-benchmark still provides sufficient evaluation data for each dimension.

## D.3 GENERAL PERFORMANCE

Table 26 presents the comprehensive zero-shot performance of LVLMs. compared to results provided in Table 1, the average ranks of models show little difference and similar trends can be observed. (1) In the black-box generation evaluation, InstructBLIP and BLIP-2 show clear advantage and hold the top-2 positions in most dimensions, followed by Lynx and LLaVA$_{L_2}$, which also perform well. In the white-box likelihood evaluation, BLIVA excels in many dimensions and the average rank, while InstructBLIP, BLIP-2, and Lynx emerge as strong competitors. (2) Some models exhibit performance differences under the two evaluation strategies, particularly noticeable for LLaVA$_{L_2}$ and BLIVA. We also investigate the phenomenon from the perspective of the models' instruction-following abilities in Appendix D.4. (3) Compared to models based on CLIP visual encoders, PandaGPT and IB-LLM, which are based on the ImageBind encoder, exhibit relatively poorer performance in image-text tasks. Meanwhile, most top-performing models utilize Vicuna

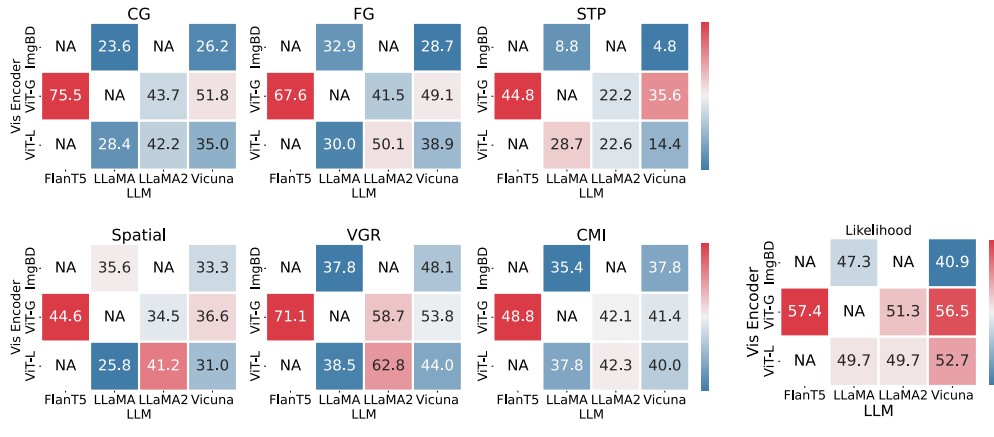

(a) Generation        (b) Likelihood

Figure 12: The influence of different language and visual backbones. For generation evaluation, we average the results of various models based on the backbone used. To better visualize the results, we selected heatmaps across six dimensions (dialog and desc are omitted). For likelihood evaluation, we further compute the average score across dimensions since the performance trend is consistent. Note that "ImgBD" is short for ImageBind in this figure.

| Visual Backbone | | ImageBind | | ViT-G | | ViT-L | | |
|---|---|---|---|---|---|---|---|---|
| Connection Arch | | BindNet+Gate | Linear | Perceiver | Q-Former | Adapter | Linear | Perceiver |
| **Generation** | Perception | 21.8 | 19.9 | 45.8 | 49.0 | 26.2 | 32.4 | 30.5 |
| | Cognition | 35.0 | 30.8 | 51.6 | 47.8 | 34.5 | 40.1 | 34.9 |
| **Likelihood** | Perception | 28.2 | 27.4 | 58.7 | 54.2 | 26.7 | 40.0 | 30.4 |
| | Cognition | 35.5 | 38.2 | 45.2 | 45.5 | 32.9 | 39.7 | 33.4 |

Table 27: Average evaluation performance categorized by connection modules (see Table 7 for more details) and visual backbones under generation and likelihood strategy.

and FlanT5 as the backbone. (4) Apart from the architecture, a common characteristic among BLIP-2, InstructBLIP, Lynx, and BLIVA is the use of relatively high-quality data during pre-training.

## D.4 COMPREHENSIVE ANALYSIS

**Model Architecture**  Since the performance of most models on the held-out datasets is consistent with that on the complete datasets, the conclusions in this part do not change. Figure 12 and Table 27 demonstrate that, language backbones are supposed to possess strong instruction-following capabilities. As for visual backbones, it's advisable to choose ViT-G and carefully select a connection module compatible with the corresponding visual backbone.

**Effect of Dataset**  Figure 13 analyzes the influence of datasets in the pre-training and instruct-tuning stages. We conclude that the usage of high-quality pre-training data like human-annotated MSCOCO (Lin et al., 2014) and synthetic captions from BLIP (Li et al., 2022) benefits the performance. Moreover, the more instruction-tuning data used, the better the model performance is.

**Generation v.s. Likelihood Evaluation**  As shown in Figure 14, likelihood evaluation yields better results than generation evaluation in most cases, even when LVLMs are guided through in-context learning. This indicates that most LVLMs have limited instruction-following capability, further hindering downstream performance. To address the issue, LVLMs should leverage stronger backbones or introduce sufficiently diverse data for instruct tuning, as done in FlanT5. Besides, the comparison between Vicuna and Vicuna+ demonstrates that multi-modal instruct tuning the backbone currently can not improve the instruction-following capability of LVLMs.

## D.5 COMPLETE *v.s* HELD-OUT DATA

To directly investigate the differences in model performance evaluated with the complete ReForm-Eval benchmark and with only commonly held-out datasets, we present the average scores and ranks of each model in Table 28 under both scenarios. Firstly, we observe that after removing the

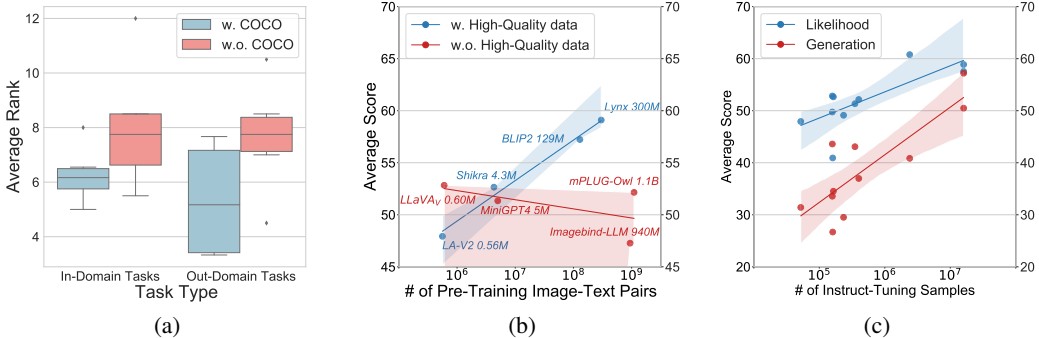

Figure 13: The influence of datasets in the pre-training and instruct-tuning stages. (a) compares the average rank of models pre-trained with and without the MSCOCO dataset. (b) shows the relationship between the scale of pre-training data and the average performance score of models grouped by data quality. (c) shows the relations between the number of instruct-tuning samples and the average score. The shaded area represents the 95% confidence interval.

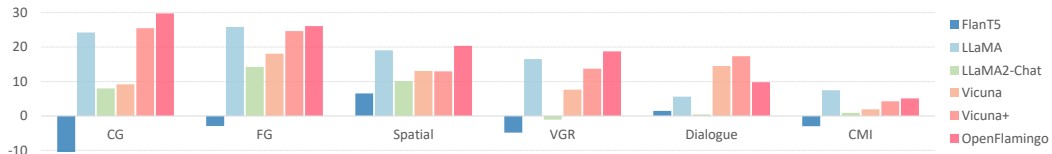

Figure 14: Performance gap of models under different evaluation strategies, grouped and averaged based on the language backbone. The vertical axis indicates how much the likelihood evaluation surpasses the generation evaluation, truncated for simplicity. "+" indicates fine-tuned backbones.

held-in dataset, the average scores of all models have decreased. This overall trend indicates that the commonly held-out datasets tend to be more challenging. For a relative comparison between LVLMs, we find that the average ranks of each model do not change significantly. Models like BLIVA, Lynx, and InstructBLIP have slightly higher ranks with the complete dataset, reflecting the impact of instruct tuning, but the differences are not substantial. The relative ranking trends of the compared models are not significantly affected. In conclusion, we believe that the evaluation results with the complete data and zero-shot evaluation show strong consistency. We attribute this to (1) data in ReForm-Eval has been re-formulated into different formats, weakening the correlation between evaluation data and instruct tuning data; (2) the rich and diverse datasets covered by ReForm-Eval, with a sufficiently large volume of data, preventing the results from being heavily influenced by individual parts. Hence, Reform-Eval benchmark is versatile and fair for evaluating LVLMs.

## E    IN-DEPTH ANALYSIS OF REFORM-EVAL BENCHMARK

### E.1    INFLUENCE OF SAMPLING METHODS

Here we clarify the sampling method used in this paper. Since the Reform-Eval covers 61 benchmarks and contains more than 500,000 evaluation samples, we employ *a balanced sampling strategy* to ensure the sampled subsets maintain similar characteristics of original benchmarks, thereby enhancing the robustness of the evaluation. Within each benchmark, we perform a balanced sampling based on the distribution of the original data, at a rate of 10% except for three cases: (1) when the size of the original benchmark is less than 1000, we keep the whole benchmark for a stable evaluation; (2) when the original benchmark has more than 10,000 evaluation samples, we filter the data and then conduct the sampling process; (3) for benchmarks used in Multi-turn Dialogue dimension, we retain all evaluation samples as the total sample volume is moderate in this dimension ($\sim$3000). It is worth noting that our calculation method is to first compute the scores of the models on each evaluation benchmark and then average across the benchmarks to obtain the final score, rather than mixing all the evaluation benchmarks together and then computing the overall score on the mixed dataset. Therefore, such sampling methods guarantee that the results on each evaluation benchmark are stable and reliable, leading to relative fairness and balance across all benchmarks.

| Model | Held-Out Data | | Complete Data | |
|---|---|---|---|---|
| | Avg. Rank | Avg. Score | Avg. Rank | Avg. Score |
| BLIP-2$_F$ | 2.00 | 56.10 | 2.25 | 62.94 |
| InstructBLIP$_F$ | 2.13 | 57.20 | 2.00 | 60.77 |
| InstructBLIP$_V$ | 4.38 | 50.48 | 4.38 | 52.20 |
| LLaVA$_V$ | 11.63 | 33.56 | 11.13 | 34.24 |
| LLaVA$_{L_2}$ | 5.38 | 43.60 | 5.88 | 45.78 |
| MiniGPT4 | 7.13 | 43.08 | 7.25 | 43.12 |
| mPLUG-Owl | 11.13 | 36.98 | 10.63 | 37.95 |
| PandaGPT | 12.63 | 26.68 | 13.88 | 26.84 |
| IB-LLM | 12.75 | 30.03 | 13.00 | 30.24 |
| LA-V2 | 13.00 | 31.41 | 12.50 | 32.60 |
| mmGPT | 14.13 | 29.51 | 14.38 | 29.38 |
| Shikra | 11.13 | 34.52 | 11.00 | 36.14 |
| Lynx | 5.38 | 49.44 | 5.00 | 50.00 |
| Cheetor$_V$ | 7.13 | 44.67 | 6.75 | 44.74 |
| Cheetor$_{L_2}$ | 7.13 | 40.86 | 7.88 | 41.75 |
| BLIVA | 9.00 | 40.84 | 7.88 | 42.40 |

Table 28: General black-box evaluation results of LVLMs with different data. Ranks and scores are averaged across all evaluation dimensions.

| Dataset | Metric | Generation | | | | | Likelihood | | | | |
|---|---|---|---|---|---|---|---|---|---|---|---|
| | | 1% | 2% | 10% | 20% | 100% | 1% | 2% | 10% | 20% | 100% |
| VQAv2 | $\rho$ | 0.9861 | 0.9948 | 0.9989 | 0.9996 | 1 | 0.9689 | 0.9857 | 0.9970 | 0.9991 | 1 |
| | $\bar{d}$ | 3.15 | 1.71 | 0.56 | 0.37 | 0 | 2.65 | 2.12 | 0.87 | 0.50 | 0 |
| Flowers102 | $\rho$ | 0.9575 | 0.9559 | 0.9794 | 0.9336 | 1 | 0.7984 | 0.7861 | 0.9131 | 0.9727 | 1 |
| | $\bar{d}$ | 8.57 | 9.03 | 4.38 | 1.70 | 0 | 12.18 | 10.69 | 3.25 | 2.93 | 0 |

Table 29: Evaluation results under different sampling ratios on the VQAv2 and Flowers102 benchmark. We derive the results of different models on test sub-benchmarks under each sampling ratio, and calculate the correlation coefficient $\rho$ and average absolute deviation $\bar{d}$ of these results compared to the results on the complete test benchmark.

We further analyze whether the shrink or expansion of the dataset size will change the evaluation results. We conduct several experiments on the VQAv2 benchmark and Flowers102 benchmark (the evaluation sample size is 21441 and 818, respectively) in the Coarse-Grained Perception and Visually Grounded Reasoning dimensions. Table 29 demonstrates that the more data sampled, the better the stability of the results, and the more consistent they are with the evaluation on the complete dataset. A 10% sampling ratio can achieve a good balance between evaluation efficiency and consistency.

Moreover, for larger datasets, the sampling ratio has little impact on the results; for smaller datasets, the sampling ratio greatly affects the results (see Table 30). Therefore, we generally perform balanced distribution sampling for larger datasets and retain the entire dataset for smaller datasets.

## E.2 INFLUENCE OF NEGATIVE OPTIONS

As we mentioned in Section 3.1, how to efficiently and effectively construct negative options is the key aspect in creating reasonable multiple-choice questions. In ReForm-Eval, different approaches are adopted for classification tasks and open-ended QA tasks. In this section, we elaborate on the rationale behind each construction method for these two types of tasks.

### E.2.1 THE NUMBER OF NEGATIVE OPTIONS

As for classification tasks, each dataset provides a fixed set of labels, restricting the output space. In order to maintain consistency with the original formulation, we do not generate new candidate options. Instead, for each question in datasets with an excess of candidate options, we apply hard negative sampling to select $N$ categories most semantically similar to the correct label as distractors. We take Flowers102 as an example to investigate the impact of the $N$ value on the evaluation results. The results are illustrated in Figure 15.

| Dataset | Size | Generation | | | | | Likelihood | | | | |
|---|---|---|---|---|---|---|---|---|---|---|---|
| | | 1% | 2% | 10% | 20% | 100% | 1% | 2% | 10% | 20% | 100% |
| VQAv2 | 21441 | 5.22 | 2.90 | 0.44 | 0.19 | 0 | 8.63 | 5.06 | 0.81 | 0.26 | 0 |
| Flowers102 | 818 | 169.79 | 85.60 | 18.83 | 6.62 | 0 | 243.01 | 174.59 | 17.79 | 10.66 | 0 |

Table 30: **Variance of accuracy(%) under different sampling ratios.** We repeatedly sample a certain percentage of the data for 10 times. We derive the accuracy each time and then compute the variance for each model. The final variance value is averaged across the models.

| Benchmark | Size | Annotation | | Scalability | Evaluation | | Instability | | | Instability |
|---|---|---|---|---|---|---|---|---|---|---|
| | | Human | ChatGPT | | ChatGPT | Unified Form | Instruction | Option Mark | Option Order | Measure |
| LAMM | 186,000 | | ✓ | high | ✓ | | | | | None |
| MME | 2,374 | ✓ | | low | | ✓ | | | | None |
| LVLM-eHub | 1,242,830 | | ✓ | high | ✓ | | ✓ | | | None |
| MMBench | 2,974 | ✓ | | low | ✓ | ✓ | | | ✓ | $\Delta acc$ |
| ReForm-Eval | 521,712 | | ✓ | high | | ✓ | ✓ | ✓ | ✓ | entropy |

Table 32: **Comparison of evaluation benchmarks.** The term "unified form" denotes a standardized evaluation format. In MMBench Liu et al. (2023c), the option order instability is measured by the difference between the accuracy $\Delta acc$ from CircularEval and VanillaEval. While in Reform-Eval, we propose to measure the instability by the entropy of the prediction distribution (see Section 3.3.2).

It can be observed that as the number of options increases, the questions become more challenging. $N = 4$ is a turning point, where the impact of increasing $N$ diminishes. Additionally, the computational cost rises with the increase of $N$, and the four-option question is the most common format for multiple-choice questions, facilitating comprehension by language backbones. Therefore, we set $N$ to 4.

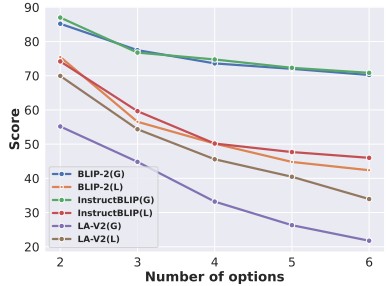

Figure 15: Performance of models with different numbers of options. "G" and "L" are respectively short for generation evaluation and likelihood evaluation.

### E.2.2 SOURCE OF DISTRACTORS

In terms of open-ended QA tasks, it is necessary to automatically generate negative options. We considered two sources of distractors: answers from other questions in the dataset and ChatGPT. In Table 31, we construct distractors in VisDial through 3 methods: randomly selecting from the answers in the dataset, hard negative sampling from the answers in the dataset based on text similarities, and generating from ChatGPT. We estimate the false-negative rate among the distractors based on human evaluation and measure the impact on evaluation results with the average score of all models.

The results indicate that the main issue with selecting distractors from the answer pool within the dataset is the high likelihood of generating false negatives. For instance, when the correct answer is "yes, it is," "yes" might also be selected as a distractor, especially in the case of hard negative sampling, resulting in a false-negative rate of 37%. Therefore, we did not evaluate models with those unreliable data. In contrast, distractors generated by ChatGPT are more reasonable, as they are less likely to be false negative and more challenging than randomly selected distractors. Overall, we believe that for open-ended QA tasks, ChatGPT is the most suitable source for distractors.

| Distractors | FN Rate | Avg. Score |
|---|---|---|
| Random in Dataset | 0.09 | 45.3 |
| HN in Dataset | 0.37 | * |
| From ChatGPT | 0.01 | 36.9 |

Table 31: Comparison between different sources of distractors in VisDial. "FN" and "HN" are respectively short for false negative and hard negative. "*" indicates that the corresponding experiment is omitted due to the false-negative issues.

### E.3    COMPARISON WITH OTHER BENCHMARKS

We compare our ReForm-Eval with several previously proposed evaluation benchmarks from the following five perspectives: (1) The dataset size; (2) During the data annotation stage, whether manual annotation is required and whether external ChatGPT-like services are needed; (3) During the evaluation phase, whether external ChatGPT-like services are required to assist the scoring, whether a unified evaluation format is used; (4) Whether the benchmark considers the instability of models with regard to prompts, and if so, whether a quantitative analysis of instability is conducted; (5) Scalability: the ability to expand the size of the dataset at a low cost. As illustrated in Table 32, ReForm-Eval has a large scale and wide coverage of data; it requires no manual efforts during the annotation stage and has strong scalability; based on the unified evaluation format, it does not require external services to assist the evaluation, making it simple and easy to use; it considers the instability of models to prompts and provides quantitative measurements, making it comprehensive and reliable. We believe that ReForm-Eval has clear superiority over previous benchmarks.

