# OpenReview forum: "ReForm-Eval: Evaluating Large Vision Language Models via Unified Re-Formulation of Task-Oriented Benchmarks"
_ICLR.cc/2024/Conference — Submitted to ICLR 2024_

### Official Review · Reviewer_J51n · 2023-10-25

**Soundness:** 3 good
**Presentation:** 3 good
**Contribution:** 3 good
**Rating:** 6
**Confidence:** 5

**Summary:**

This paper contributes a new benchmark for evaluating the large vision-language models (LVLMs) comprehensively. The benchmark re-formulates 61 benchmark datasets based on existing data resources and evaluate the models with both black-box and white-box methods. The authors also conduct extensive experiments to analyze the strengths and weaknesses of existing LVLMs.

**Strengths:**

1. The scale of benchmark is large, including 61 datasets and is 100 times the size of MMBench.
2. The evaluation dimensions are comprehensive, including many perception and cognition sub-tasks.
3. Some insightful conclusions are required from the experiments on proposed benchmark.

**Weaknesses:**

Limitations in novelty to some extent: the formulation of multiple-choice is widely used in previous work like MMBench [1]; the proposed generation and likelihood evaluation method are also used in previous benchmarks, like VisualGPTScore [2] proposed using likelihood of generating references conditioned on images and prompts to do multiple-choices tasks.

[1] Liu, Yuan, et al. "MMBench: Is Your Multi-modal Model an All-around Player?." arXiv preprint arXiv:2307.06281 (2023).
[2] Lin, Zhiqiu, et al. "VisualGPTScore: Visio-Linguistic Reasoning with Multimodal Generative Pre-Training Scores." arXiv preprint arXiv:2306.01879 (2023).

**Questions:**

1. Are the formulation of current datasets used in the benchmark modified manually?
2. Should we use generation metric, likelihood metric, or both of them when using the benchmark?
3. Are there any human validation or other validations to show the superiority over other benchmarks?

**Details Of Ethics Concerns:**

Considering that this benchmark contains 61 existing datasets, are there any copyright issues involved in re-formulating them?

---

> ### Author Response · Authors · 2023-11-17
> **Response to Official Review of Submission3580 by Reviewer J51n (Part 1 of 2)**
>
> Thank you for your valuable advice. Here is our response to your concerns and questions:
>
> **Weakness 1: Limitation in novelty. The format of multiple-choice questions and the likelihood-based evaluation method have been explored.**
>
> Response:  **The core contribution of ReForm-Eval is a strategy for re-formulating existing task-oriented datasets to suit the evaluation of LVLMs**. The reason for choosing multiple-choice questions is that this format is in line with the evaluation needs of LVLMs to output in free-form texts. At the same time, based on the findings that current LVLMs have weak capability in following multiple-choice instructions, we further assist the evaluation through both black-box (text-only in-context samples) and white-box (likelihood) methods. Thus, **ReForm-Eval exhibits a versatile and efficient way to evaluate LVLMs by combining these individual components in a reasonable way.**
>
> ------------
>
> **Q1: Are the formulation of current datasets used in the benchmark modified manually?**
>
> Response:  In ReForm-Eval, we propose a re-formulation framework for existing benchmark datasets. For each dataset, we conduct thorough evaluation and analysis, based on which the target problem format of re-formulation is determined. Subsequently, the data is transformed with the corresponding **automatic pipeline** **where no human annotation is required**.
>
> -----------
>
> **Q2: Should we use the generation metric, likelihood metric, or both of them when using the benchmark?**
>
> Response:  If possible, **we recommend using both approaches for comparison.** The **black-box strategy is more versatile** but requires higher adherence to instructions by models; some models (like BLIVA) require the **white-box method to better reflect their multimodal capabilities**, and it offers higher stability and computational efficiency. At the same time, comparing the performance differences between the two strategies can also reveal the instruction-following abilities between models.

---

> ### Author Response · Authors · 2023-11-17
> **Response to Official Review of Submission3580 by Reviewer J51n (Part 2 of 2)**
>
> **Q3: Are there any human validation or other validations to show the superiority over other benchmarks?**
>
> Response:
>
> Thanks for your advice. **We compare ReForm-Eval with other LVLM benchmarks from five perspectives: (1) Dataset size; (2) Annotation; (3) Evaluation; (4) Instability and the measurement; (5) Scalability.** As demonstrated in Rebuttal Table 7, ReForm-Eval has a large scale and wide coverage of data; it requires no manual efforts during the annotation stage and has strong scalability; based on the unified evaluation format, it does not require external services to assist the evaluation, making it simple and easy to use; it considers the instability of models to prompts and provides quantitative measurements, making it comprehensive and reliable. **We believe that ReForm-Eval has clear superiority over previous benchmarks.**
>
> Rebuttal Table 7: Comparison between benchmarks for LVLMs.
>
> | Benchmark   | Size  | Annotation   |              | Scalability | Evaluation   |                  | Instability     |                 |                  | Instability Measure  |
> | ----------- | ----- | ------------ | ------------ | ----------- | ------------ | ---------------- | --------------- | --------------- | ---------------- | ------------ |
> |             |       | **Human**    | **ChatGPT**  |             | **ChatGPT**  | **Unified Form** | **Instruction** | **Option Mark** | **Option Order** |  |
> | LAMM        | 186k  |              | $\checkmark$ | high        | $\checkmark$ |                  |                 |                 |                  | None         |
> | MME         | 2.4k  | $\checkmark$ |              | low         |              | $\checkmark$     |                 |                 |                  | None         |
> | LVLM-ehub   | 1243k |              |              | high        | $\checkmark$ |                  | $\checkmark$    |                 |                  | None         |
> | MMBench     | 3.0k  | $\checkmark$ |              | low         | $\checkmark$ | $\checkmark$     |                 |                 | $\checkmark$     | $\Delta acc$ |
> | ReForm-Eval | 521k  |              | $\checkmark$ | high        |              | $\checkmark$     | $\checkmark$    | $\checkmark$    | $\checkmark$     | entropy      |
>
>
> Furthermore, we show that **an obvious advantage of Reform-Eval is its large volume of data, providing a more stable estimation of model capabilities.** We perform bootstrap sampling on the data and measure the stability in assessing model capabilities by calculating the variance of the bootstrap estimator. As presented in Rebuttal Table 8, ReForm-Eval provides a more stable evaluation.
>
> Rebuttal Table 8: Variance of bootstrap estimators on VQAv2 dataset from ReForm-Eval and on fine-grained perception (single-instance) dataset from MMBench. For each, we repeatedly draw 20% of the data with replacement for 10 times.
> | **Benchmark**   | **BLIP-2** | **LLaVA_L2** | **mPLUG-Owl** | **PandaGPT** | **ImageBindLLM** | **mmGPT** | **Shikra** | **Cheetor_V** |
> | --------------- | ---------- | ------------ | ------------- | ------------ | ---------------- | --------- | ---------- | ------------- |
> | **MMBench**     | 6.7        | 7.1          | 3.28          | 8.08         | 4.79             | 8.72      | 14.25      | 5.94          |
> | **Reform-Eval** | 0.3303     | 0.2316       | 0.0658        | 0.1403       | 0.3683           | 0.1587    | 0.2341     | 0.292         |
>
> -----------
>
> **Q4: Will there be copyright issues?**
>
> Response:
>
> During the construction of ReForm-Eval benchmark, **we only use the open-source datasets. The usage of these data for academic purposes is permitted.** The purpose of ReForm-Eval is to assist in the academic research of LVLMs, so we believe there should be no related issues. Furthermore, we will supplement the copyright requirements of the corresponding datasets in the usage instructions of ReForm-Eval.

---

> > ### Comment · Reviewer_J51n · 2023-11-22
> > **Response to authors**
> >
> > Thank the authors for response. The response has addressed my questions to some extent. After consideration, I decide to raise the score to 6.

---

### Official Review · Reviewer_3n83 · 2023-10-28

**Soundness:** 3 good
**Presentation:** 3 good
**Contribution:** 4 excellent
**Rating:** 6
**Confidence:** 4

**Summary:**

This paper proposes a new strategy to benchmark large vision language models.
It reformulates 61 existing benchmarks into multiple-choice problems or specialized text generation problems,
test existing LVLMs with the ReForm-Eval, and report the accuracy and CIDEr for two types of problems, respectively.
With ReForm-Eval, the authors benchmarked multiple LVLMs and studied the effect of model backbone, connection module, pre-training data, instruction-tuning data.
Furthermore, the paper also discussed the effect of in-context sample, the difference between generation and likelihood based evaluation, and the instability during evaluation.

**Strengths:**

This paper provides a practical approach to unify existing computer vision benchmarks under a unified formulation. It converts 61 existing datasets to multi-choice problems and specialized text generation problems and provides extensive evaluation results.

**Weaknesses:**

1. Some findings presented in this paper are not original findings. For example, the effect of in-context examples and the instability of existing LVLMs have already been discussed in MMBench.
2. The core contribution of this work is to propose an approach to convert existing benchmarks to a unified formulation. However, the authors used many pages to present and discuss the evaluation results, rather than delving deeper into the reformulation methodologies. In fact, many aspects can be explored during the reformulation:
    1. In general, one need to use some distractors as the negative options when building multi-choice problems. There exists multiple ways to obtain these distractors (as mentioned in this paper): 1. find the negative classes with the highest confidence; 2. find some semantically related but not synonymous answers; 3. LLM-based hard-negative generation. Besides, another baseline is to randomly pick incorrect class labels as negative options. How can the use of those distractors quantitatively affect the evaluation results?
    2. Fine-grained recognition, which is a substantial component of Fine-grained perception,  is not included in the fine-grained perception tasks.

**Questions:**

1. For Figure 10, why the proportions of 'A' and 'B' in Ground-truth Option Distribution are not the same? Does that mean there exists questions with only one option?
2. Typo in Page.8 Line 1, MSOCO
3. It would be better if the authors can provide more ablation study for the reformulation process, on factors including: 1. the methods to add the distractors, 2. the number of options.
4. Reform-Eval is a large dataset contains over 500,000 evaluation instances, when doing the sub-sampling, do the authors do it uniformly or sample evaluation instances from each benchmark with different probability to improves the data balance? Besides, have the authors studied if the shrink of the dataset size will change the evaluation results? Can we use a even smaller subset for evaluation?

---

> ### Author Response · Authors · 2023-11-17
> **Response to Official Review of Submission3580 by Reviewer 3n83 (Part 1 of 2)**
>
> Thank you for your valuable advice. Here is our response to your concerns:
>
> **Weakness 1: Some findings presented in this paper are not original findings, namely the effect of in-context samples and the instability of LVLMs.**
>
> Response:
>
> 1. In-context samples have been widely used during the model training stage to enhance the instruction following ability of models, i.e. MIMIC-IT dataset used in Otter. **To the best of our knowledge,** **ReForm-Eval is the first attempt to utilize in-context samples to** **assist the** **evaluation of LVLMs****.** In this paper, **our novel findings is that** complete image-text pairs are not necessary as in-context samples; **text-only in-context samples can provide effective guidance**, helping LVLMs to answer multiple-choice questions in the correct format.
> 2. The analysis of instability in MMBench is limited to the impact of option order perturbations. **In ReForm-Eval, we comprehensively consider various perturbations****:** not only shuffling option order, but also applying different instructions and option markers. Furthermore, we **quantitatively measure the instability of models in the form of entropy**. Above are the original contributions of ReForm-Eval in terms of instability.
>
> -------------
>
> **Weakness 2.1: Lacking analysis and experiments on the re-formulation process, such as the quantity and sources of distractors.**
>
> Response: Please refer to our response to Q3 in the common response.
>
> -------------
>
> **Weakness 2.2: Fine-grained recognition tasks are not included in ReForm-Eval.**
>
> Response:
>
> We believe this is a misunderstanding due to the different interpretations of term "fine-grained". **In ReForm-Eval, fine-grained perception refers to the perception of semantic information at the level of local objects.** On the other hand, fine-grained image classification tasks (**also known as fine-grained recognition**) focus on distinguishing fine-grained categories, which involve perception at the image level. Therefore, in ReForm-Eval, **such tasks are categorized as image classification tasks in coarse-grained perception**, such as Flowers102 and Pets37.

---

> ### Author Response · Authors · 2023-11-17
> **Response to Official Review of Submission3580 by Reviewer 3n83 (Part 2 of 2)**
>
> Here is our response to your questions:
>
> **Q1: For Figure 10, why the proportions of 'A' and 'B' in Ground-truth Option Distribution are not the same?**
>
> Response:  In this dataset, there are no questions with only one option. The distribution of correct answers has a certain level of randomness due to the shuffle, so the proportions of A and B are not exactly the same.
>
> ---------
>
> **Q2: Typo in Page.8 Line 1**
>
> Response: Thank you for pointing out the error. We have corrected this typo.
>
> ---------
>
> **Q3: It would be better if the authors could provide more ablation studies for the reformulation process**
>
> Response: Please refer to our response to Q3 in the common response.
>
> ---------
>
> **Q4: Explanation and analysis of the sampling method.**
>
> Response:
>
> - **Detailed explanation of our sampling method:** Since the Reform-Eval covers 61 benchmarks and contains more than 500,000 evaluation samples, we employ ***a balanced sampling strategy*** to enhance the robustness of the evaluation. Within each benchmark, we sample 10% data based on the distribution of the original data **except for three cases**: (1) when the size of the original benchmark is less than 1000, we keep the whole benchmark for a stable evaluation; (2) when the original benchmark has more than 10,000 evaluation samples, we filter the data and then conduct the sampling process; (3) for benchmarks used in Multi-turn Dialogue dimension, we retain all evaluation samples as only two datasets are included and the total sample volume is moderate in this dimension (~3000).
>
> - **Why we sample at a rate** **of** **10%**: We analyze whether the shrink or expansion of the dataset size will change the evaluation results, by conducting several experiments on the VQAv2 benchmark and Flowers102 benchmark (the evaluation sample size is 21441 and 818, respectively). Rebuttal Table 5 demonstrates that the more data sampled, the more consistent they are with the evaluation on the complete dataset. **A 10% sampling ratio can achieve a good balance between the evaluation efficiency and consistency.**
>
> - **Rationality of our sampling method**: As shown in Rebuttal Table 6, for larger datasets, the sampling ratio has less impact on the results; for smaller datasets, different sampling ratios cause the results to fluctuate  dramatically. **To ensure the stability of the evaluation, we generally perform balanced sampling for larger datasets and retain the entire dataset for smaller datasets.**
>
> Rebuttal Table 5: Evaluation results under different sampling ratios on the VQAv2 and Flowers102 benchmark. We derive the results of different models on test sub-benchmarks under each sampling ratio, and calculate the correlation coefficient $\rho$ and average absolute deviation $\bar{d}$ of these results compared with the results on the complete test benchmark.
>
> | **Dataset**    | **Metric**  | **Generation** |        |         |         |          | **Likelihood** |         |         |         |          |
> | -------------- | ----------- | -------------- | ------ | ------- | ------- | -------- | -------------- | ------- | ------- | ------- | -------- |
> |                |             | **1%**         | **2%** | **10%** | **20%** | **100%** | **1%**         | **2%**  | **10%** | **20%** | **100%** |
> | **VQAv2**      | $\rho$    | 0.9861         | 0.9948 | 0.9989  | 0.9996  | 1        | 0.9689         | 0.9857  | 0.9970  | 0.9991  | 1        |
> |                | $\bar{d}$ | 3.1483         | 1.7075 | 0.5550  | 0.3658  | 0        | 2.6450         | 2.1167  | 0.8725  | 0.4958  | 0        |
> | **Flowers102** | $\rho$    | 0.9575         | 0.9559 | 0.9794  | 0.9336  | 1        | 0.7984         | 0.7861  | 0.9131  | 0.9727  | 1        |
> |                | $\bar{d}$ | 8.5738         | 9.0256 | 4.3775  | 1.6994  | 0        | 12.1756        | 10.6850 | 3.2488  | 2.9275  | 0        |
>
> Rebuttal Table 6: Variance of accuracy(%) under different sampling ratios. We repeatedly sample a certain percentage of the data for 10 times. We derive the accuracy each time and then compute the variance for each model. The final variance value is averaged across the models.
>
> | **Dataset**    | **Size** | **Generation** |        |         |         |          | **Likelihood** |        |         |         |          |
> | -------------- | -------- | -------------- | ------ | ------- | ------- | -------- | -------------- | ------ | ------- | ------- | -------- |
> |                |          | **1%**         | **2%** | **10%** | **20%** | **100%** | **1%**         | **2%** | **10%** | **20%** | **100%** |
> | **VQAv2**      | 21441    | 5.22           | 2.90   | 0.44    | 0.19    | 0        | 8.63           | 5.06   | 0.81    | 0.26    | 0        |
> | **Flowers102** | 818      | 169.79         | 85.60  | 18.83   | 6.62    | 0        | 243.01         | 174.59 | 17.79   | 10.66   | 0        |

---

### Official Review · Reviewer_mogP · 2023-10-30

**Soundness:** 2 fair
**Presentation:** 3 good
**Contribution:** 2 fair
**Rating:** 5
**Confidence:** 4

**Summary:**

This work introduces a novel benchmark called ReForm-Eval for assessing large vision-language models. The underlying approach of ReForm-Eval involves transforming several publicly available VQA datasets into a multiple-choice format.

**Strengths:**

1. This work proposes a novel benchmark, namely ReForm-Eval. Reformatting the current VQA dataset into multiple-choice questions partially alleviates a problem in using the current metric in the VQA dataset to evaluate generative VLMs. The problem is the exact matching between the prediction and the reference target, which leads to potential limitations.
2. Some insights are good, such as "FlanT5-based models" performing well on the multiple-choice tasks, which aligns with findings in the NLP domain.

**Weaknesses:**

1. The very impressive ability lies in the large language models (GPT-4, ChatGPT, Llama, etc.) and the popular vision-language model (CLIP) is their powerful zero-shot learning ability. One important way to evaluate the zero-shot learning of models is to ensure there is no dataset overlap between the evaluating data and the training data, such as the evaluating strategy in CLIP[1]. However, ReForm-Eval includes many datasets that are trained in the evaluated VLM. This might incur two issues: (1) it is unfair to compare models that were trained by datasets evaluated in ReForm-Eval with models that have not been trained on any datasets in ReForm-Eval. (2) Ultimately, ReForm-Eval can only evaluate the "supervised learning" ability instead of the "zero-shot learning."

2. Some insights might not be solid. For example, (1) when discussing which connection module is more suitable for which visual backbone, this work should ensure that other influential factors are the same between compared models, such as training data and language model. (2) The grouping of high-quality data and without high-quality data might be cherry-picked, as some models (Lynx) in the high-quality group also use "data filtered on rules or CLIP." (3) The variance in Fig. 4 (c) is so large that it is difficult to conclude that "more instructional data leads to better performance."

3. Using CIDEr to evaluate visual descriptions is not optimal, especially for models that intend not to generate concise descriptions, such as LLaVA, and it benefits models that are tuned by dataset, such as coco-caption, whose ground truth descriptions are in a shorter format.

[1] Radford, Alec, et al. "Learning transferable visual models from natural language supervision." ICML 2021.

**Questions:**

N/A

---

> ### Author Response · Authors · 2023-11-17
> **Response to Official Review of Submission3580 by Reviewer mogP**
>
> Thanks for your valuable advice. Here is our response to your concerns:
>
> **Q1: Some models might have been trained on datasets included in ReForm-Eval, potentially leading to unfair comparisons and challenges in measuring zero-shot capabilities of the models.**
>
> Response: Please refer to our response to Q1 in the common response.
>
> --------
>
> **Q2: Questions about the soundness of insights gleaned in this paper. (1) Issues regarding controlling variables; (2) Criteria for grouping models in Figure 4 (b); (3) Excessive variance in the curve fitted in Figure 4 (c).**
>
> Response: Please refer to our response to Q2 in the common response.
>
> --------
>
> **Q3: CIDEr may not be suitable as a metric for visual description tasks.**
>
> Response:
>
> We chose CIDEr as an automated evaluation metric following BLIP-2. In the updated version of our paper (see Table 20), we have supplemented other generation metrics including BLEU-4, Meteor and Rouge-L for a comprehensive evaluation. Here is a quick look:
>
> Rebuttal Table 4: Evaluation results on visual description based on CIDEr, BLEU-4, Meteor and Rouge-L. The evaluation dataset is NoCaps for zero-shot evaluation.
> | **Model**          | **CIDEr** | **BLEU-4** | **Meteor** | **Rouge-L** |
> | ------------------ | --------- | ---------- | ---------- | ----------- |
> | **BLIP-2**         | 83.57     | 37.18      | 27.09      | 49.34       |
> | **Lynx**           | 77.31     | 30.52      | 26.26      | 47.30       |
> | **MiniGPT4**       | 58.71     | 22.84      | 24.99      | 40.83       |
> | **mPLUG-Owl**      | 48.43     | 17.35      | 21.68      | 37.66       |
> | **LLaVA_V**        | 42.43     | 17.39      | 21.67      | 36.27       |
> | **InstructBLIP_V** | 30.18     | 7.71       | 13.48      | 25.13       |

---

> > ### Comment · Reviewer_mogP · 2023-11-21
> >
> > Thank you for your response.
> >
> > However, the primary concern remains: Given that current LVLM training tends to utilize all available data, my concern is that ReForm-Eval may ultimately assess only 'supervised learning' capabilities, rather than 'zero-shot learning.' This concern could significantly influence the long-term contribution of this work to the community.
> >
> > Furthermore, I acknowledge the difficulty in fairly evaluating current LMMs. Therefore, I suggest presenting only results that can be scientifically validated.

---

> > > ### Author Response · Authors · 2023-11-21
> > > **Response to the long-term concern by Reviewer mogP**
> > >
> > > Thanks for your feedback! Here is our response to your concern:
> > > 1. From a short-term perspective: Our constructed benchmark does provide ample data to assess the zero-shot capabilities of existing models.
> > >
> > > 2. From a long-term perspective, ReForm-Eval is also meaningful:
> > >
> > >    * The assumption that "LVLM training tends to utilize **all** available data" is unrealistic due to **significant computational costs**. Furthermore, following the development trend of LLMs, we anticipate that LVLM training will converge to paradigms following Flan-T5, partitioning held-in and held-out datasets in a reasonable way (as done in InstructBLIP).
> > >
> > >    * Moreover, ReForm-Eval includes two types of held-out data suitable for long-term evaluation:
> > >
> > >      - Datasets like CUTE80, Winoground, and Whoops, which are typically not used in training **as they only have test sets**;
> > >
> > >      - Datasets like MP3D-Spatial, originally proposed in this paper, **built upon embodied visual navigation simulators**.
> > >
> > >    * ReForm-Eval is not limited to the current 61 datasets. We will continue to maintain and expand ReForm-Eval, leveraging its efficient scalability to **encompass more types of tasks and data**, ensuring comprehensive coverage.
> > >
> > > Lastly, thank you for your suggestion. We will take zero-shot results as the main results in the next version of our paper.

---

### Official Review · Reviewer_cerW · 2023-11-04

**Soundness:** 2 fair
**Presentation:** 3 good
**Contribution:** 1 poor
**Rating:** 3
**Confidence:** 5

**Summary:**

This paper claims that the capabilities of LVLMs have not been comprehensively and quantitatively evaluated. Accordingly, it proposes a ReForm-Eval benchmark, which re-formulates existing task-oriented benchmarks into unified LVLM-compatible formats. Based on ReForm-Eval, it conducts extensive experiments, thoroughly analyzes the strengths and weaknesses of existing LVLMs, and try to reveal insights behind LVLMs.

**Strengths:**

1. ReForm-Eval benchmark re-formulates 61 benchmark datasets based on existing data resources, including visual perception to high-level visual reasoning and dialog.
2. ReForm-Eval has a large scale.

**Weaknesses:**

1. ReForm-Eval is not suitable for a fair comparison of capability dimensions among different LVLMs. It is composed of 61 existing datasets. A number of these datasets are widely used in training data for LVLMs, e.g., VQA, VQAv2, GQA, OK-VQA, TextVQA, OCR-VQA, Text caps, Flickr30K, and so on. Different LVLMs will choose different training datasets but only cover some of them. However, ReForm-Eval merges both trained and reserved datasets into ability dimensions, leading to unfair comparison.
2. This paper tries to reveal insights into model architecture and training datasets. However, since different LVLMs have different model architectures (Vision Encoders, connection modules, LLMs) and training datasets (pretraining and instruction tuning), the summarized insights in this paper are not convincing.
3. The methods to generate appropriate negative options in visually grounded reasoning and multi-turn dialogue may not be reliable. The information of question and answer may not be sufficient to generate reasonable negative options with ChatGPT.

**Questions:**

See weakness

---

> ### Author Response · Authors · 2023-11-17
> **Response to Official Review of Submission3580 by Reviewer cerW**
>
> Thank you for your valuable advice. Here is our response to your questions:
>
> **Q1: Some models might have been trained on datasets included in ReForm-Eval, potentially leading to unfair comparisons.**
>
> Response:  Please refer to our response to Q1 in the common response.
>
> ---------
>
> **Q2: Questions about the soundness of insights gleaned in this paper, namely the issues regarding controlling variables.**
>
> Response:  Please refer to our response to Q2.1 in the common response.
>
> ---------
>
> **Q3: Lack of rational analysis of using ChatGPT to generate distractors based on textual QA pairs.**
>
> Response:  Thank you for the feedback. Here, we assess the rationality of distractors on two levels:
>
> * **Reliability**: whether distractors face false-negative issues;
> * **Difficulty**: whether negative options are challenging and not overly naive.
>
> According to the Rebuttal Table 3 in the common response, **ChatGPT-generated distractors have a low false-negative rate in terms of reliability**. Regarding difficulty, we find that the performance of all models is not satisfactory (the best model achieves an accuracy of 55%), indicating that **ChatGPT-generated distractors are challenging**. Introducing additional image-related information, such as captions and objects, would require extra information or the use of detection models, incurring higher costs. Therefore, we believe the current approach is relatively reliable and efficient.

---

> ### Author Response · Authors · 2023-11-23
> **Hope to receive feedback from reviewer cerW**
>
> Dear Reviewer cerW:
>
> Tonight is the rebuttal deadline. We really want to receive your further feedback, as in response to your questions, we have supplemented relevant experimental results and provided explanations.
> We hope our responses can address your concerns. We will be very happy to clarify any further questions.
>
> Best regards,
>
> Authors

---

### Author Response · Authors · 2023-11-17
**Common Response to Official Reviews (Part 1 of 3)**

We appreciate the valuable feedback posed by reviewers. In response to the raised concerns and questions, we have updated relevant experimental results and discussions in Appendix D,E of the paper. We kindly ask the reviewers to refer to it.

Subsequently, we summarize and respond to each question one by one.

**Q1: Some models might have been trained on datasets included in ReForm-Eval, potentially leading to unfair comparisons and challenges in measuring zero-shot capabilities of the models.**

Response:

1. With regard to fairness, we introduce two **adaptive sub-benchmark construction** methods based on Reform-Eval: (1) *Model-oriented*: **Selecting** **the** **held-out** **datasets** **common to all compared models for zero-shot evaluation**; (2) *User-oriented*: Allowing users to choose and combine benchmarks based on their own requirements. The coverage and flexibility of ReForm-Eval inherently support these sub-benchmark construction methods. Ample data can be provided for zero-shot evaluation of many models.  To facilitate the construction of adaptive benchmarks, we will provide **an interactive interface** where users can select any number of models and any number of evaluation datasets to obtain a benchmark in-need. We will release the tool with this paper.

2. Based on the sub-benchmark construction method, we conduct the zero-shot evaluation and the results have been updated in Appendix D of the paper. Overall, we find that the **zero-shot evaluation results and the corresponding findings are highly consistent with those obtained from the full dataset.** Rebuttal Table 1 presents the comprehensive performance of the models evaluated with different data. There is little variation in the relative ranking trends of the compared models. We attribute this to (1) **data in ReForm-Eval has been re-formulated into different formats, weakening the correlation between evaluation data and instruct tuning data**; (2) **the rich and diverse datasets covered by ReForm-Eval, preventing the results from being heavily influenced by individual parts**. Hence, ReForm-Eval benchmark is versatile and fair for evaluating LVLMs. We plan to use the zero-shot evaluation as the main experimental results in the next version of our paper.

Rebuttal Table 1: Comparison between black-box evaluation results of LVLMs with different data. Ranks and scores are averaged across all evaluation dimensions.

| **Metric**    | **Benchmark** | **BLIP-2** | **InstructBLIP_F** | **InstructBLIP_V** | **Lynx** | **MiniGPT4** | **mmGPT** |
| ------------- | ------------- | ---------- | ------------------ | ------------------ | -------- | ------------ | --------- |
| **Avg-Rank**  | Full          | 2.25       | 2.00               | 4.38               | 5.00     | 7.25         | 14.38     |
|               | Sub           | 2.00       | 2.13               | 4.38               | 5.38     | 7.13         | 14.13     |
| **Avg-Score** | Full          | 62.94      | 60.77              | 52.20              | 50.00    | 43.12        | 29.38     |
|               | Sub           | 56.10      | 57.20              | 50.48              | 49.44    | 43.08        | 29.51     |

---

> ### Author Response · Authors · 2023-11-17
> **Common Response to Official Reviews (Part 2 of 3)**
>
> **Q2: Questions about the soundness of insights gleaned in this paper.**
>
> **Q2.1**  **Issues regarding controlling variables;**
>
> Response:
>
> This issue is difficult to resolve because **current LVLMs are not ample enough to enable precise control of variables**. For instance, there are few LVLMs with identical encoder architectures or training datasets. Therefore, we group models based on specific factors. **We believe that comparing the average performance of models within different groups can to some extent reflect the impact of that factor**. We will continue to maintain and update ReForm-Eval, adding more models for evaluation to better reflect the impact of different factors.
>
> **Q2.2**  **Criteria for grouping models in Figure 4 (b);**
>
> Response:
>
> Regarding Figure 4(b), models like Lynx also used data from CC3M, CC12M, but this data only accounted for a small proportion of training samples. Here we provide the criteria for distinguishing between the two types of models in this figure: We consider CC3M, CC12M, and LAION to be rule-based or CLIP-based filtered, and thus of relatively lower quality. Thus **we distinguish the red and blue models by the proportion of low-quality data in their total training data**. Rebuttal Table 2 reflects the proportion of low-quality data for these models.
>
> Rebuttal Table 2: Illustration of the quality of pre-training data used by LVLMs.
>
> | **Model**              | **LLaVA** | **MiniGPT4** | **Imagebind-LLM** | **mPLUG-Owl**   | **LA-V2** | **Shikra** | **BLIP2** | **Lynx** |
> | ---------------------- | --------- | ------------ | ----------------- | --------------- | --------- | ---------- | --------- | -------- |
> | **High quality data**  | 0         | 0            | 0.5M(COCO)        | 0.5M(COCO)      | 567K      | 4.3M       | ~115M     | ~288M    |
> | **Total data**         | 595K      | 5M           | 940M              | ~1.1B           | 0         | 4.3M       | 129M      | ~300M    |
> | **High quality Ratio** | 0         | 0            | $5.31 * 10^{-4}$  | $4.55* 10^{-4}$ | 1         | 1          | 0.89      | 0.96     |
> | **Low quality Ratio**  | 100%      | 100%         | ~100%             | ~100%           | 0%        | 0%         | 11%       | 4%       |
>
> **Q2.3  Excessive variance in the curve fitted in Figure 4 (c).**
>
> Response:
>
> In response to the issue of high variance, we re-analyzed the data. We found significant differences in sample sizes across different instruction-tuning datasets. Therefore, we replace the horizontal axis to be the sample size, so as to describe the amount of information learned by different models at this stage. Please refer to the revised version of this figure in Appendix D.4. **As the data volume increases, the performance of the models shows a clear upward trend, and the variance of the fitted curve is acceptable.**

---

> ### Author Response · Authors · 2023-11-17
> **Common Response to Official Reviews (Part 3 of 3)**
>
> **Q3: Lacking analysis and experiments on the re-formulation process, such as the quantity and sources of distractors.**
>
> Response:
>
> We appreciate the valuable advice. The corresponding results and discussions have been updated in Appendix E.2 of the revised paper. Here is a brief summary:
>
> 1. **The quantity of distractors**: For classification tasks, the fixed label sets restrict the feasible output space. To maintain consistency, we merely reduce the number of options to N. According to Figure 15, as the number of options increases, the questions become more challenging. **N=4 is a turning point of slope from sharp to mild, and considering the trade-off where more options incur greater computational cost, we set N=4**. Questions with 4 options are also the most common form of multiple-choice questions.
>
> 2. **The sources of distractors**: For open-ended QA tasks, we compare 3 sources of distractors: randomly selecting from answers to other questions, further conducting hard negative sampling, and generating from ChatGPT. We manually check the sampled distractors to estimate the false-negative rate and calculate the average scores of all models with different types of distractors. According to Rebuttal Table 3, **negative options selected from answers to other questions are likely to be false negative**, especially with hard negative sampling (like "yes" and "yes, it is"). **Distractors generated by ChatGPT are challenging with the lowest false-negative rate**. Therefore, we believe that ChatGPT is the most suitable source in such cases.
>
> Rebuttal Table 3: Comparison between different sources of distractors in VisDial. “*” indicates that the corresponding experiment is omitted due to low confidence in false-negative issues.
>
> | **Source of Distractors**     | **False Negative Rate** | **Avg. Score of All Models** |
> | ----------------------------- | ----------------------- | ---------------------------- |
> | **Random answers in Dataset** | 0.087                   | 45.3                         |
> | **Hard Negatives in Dataset** | 0.37                    | *                            |
> | **Generated from ChatGPT**    | 0.01                    | 36.9                         |

---

### Author Response · Authors · 2023-11-20
**Looking forward to further feedbacks**

Dear Reviewers:

Thanks again for your great efforts and valuable comments. In response to your questions, we have supplemented relevant experimental results and provided explanations. We hope our responses can address your concerns. As the discussion period is coming to a close, we are looking forward to receiving further feedback. We will be very happy to clarify further concerns (if any).

Best regards,

Authors

---

### Meta-Review · Area_Chair_XrPt · 2023-12-09

**Metareview:**

The meta-reviewer has carefully read the paper, reviews, rebuttals, and discussions between authors and reviewers. The meta-reviewer agrees with the reviewers that this is a borderline submission to ICLR (a bit below the bar).  The paper introduces a benchmark for evaluating large vision-language models (LVLMs) by reformulating 61 existing datasets into formats suitable for LVLMs, using an evaluation system called ReForm-Eval. It measures LVLM performance using accuracy and CIDEr metrics for multiple-choice and text-generation tasks. The study investigates the effects of various factors on LVLM performance, including the model architecture, pre-training data, and instruction tuning, while also examining evaluation methods and the issue of stability in assessments. The meta-reviewer agrees with the reviewers that several aspects of the paper need to be polished, such as presenting zero-shot/few-shot/supervised results, and copyright issues of the datasets used. The meta-reviewer believes this can be a strong submission in the next venue with related issues addressed.

**Justification For Why Not Higher Score:**

N/A

**Justification For Why Not Lower Score:**

N/A

---

### Decision · Program_Chairs · 2024-01-16

Reject